# THEORETICAL CHARACTERIZATION OF THE GENERALIZATION PERFORMANCE OF OVERFITTED META-LEARNING

**Peizhong Ju**
Department of ECE
The Ohio State University
Columbus, OH 43210, USA
`ju.171@osu.edu`

**Yingbin Liang**
Department of ECE
The Ohio State University
Columbus, OH 43210, USA
`liang.889@osu.edu`

**Ness B. Shroff**
Department of ECE & CSE
The Ohio State University
Columbus, OH 43210, USA
`shroff.11@osu.edu`

## ABSTRACT

Meta-learning has arisen as a successful method for improving training performance by training over many similar tasks, especially with deep neural networks (DNNs). However, the theoretical understanding of when and why overparameterized models such as DNNs can generalize well in meta-learning is still limited. As an initial step towards addressing this challenge, this paper studies the generalization performance of overfitted meta-learning under a linear regression model with Gaussian features. In contrast to a few recent studies along the same line, our framework allows the number of model parameters to be arbitrarily larger than the number of features in the ground truth signal, and hence naturally captures the overparameterized regime in practical deep meta-learning. We show that the overfitted min $\ell_2$-norm solution of model-agnostic meta-learning (MAML) can be beneficial, which is similar to the recent remarkable findings on "benign overfitting" and "double descent" phenomenon in the classical (single-task) linear regression. However, due to the uniqueness of meta-learning such as task-specific gradient descent inner training and the diversity/fluctuation of the ground-truth signals among training tasks, we find new and interesting properties that do not exist in single-task linear regression. We first provide a high-probability upper bound (under reasonable tightness) on the generalization error, where certain terms decrease when the number of features increases. Our analysis suggests that benign overfitting is more significant and easier to observe when the noise and the diversity/fluctuation of the ground truth of each training task are large. Under this circumstance, we show that the overfitted min $\ell_2$-norm solution can achieve an even lower generalization error than the underparameterized solution.

## 1 INTRODUCTION

Meta-learning is designed to learn a task by utilizing the training samples of many similar tasks, i.e., learning to learn (Thrun & Pratt, 1998). With deep neural networks (DNNs), the success of meta-learning has been shown by many works using experiments, e.g., (Antoniou et al., 2018; Finn et al., 2017). However, theoretical results on why DNNs have a good generalization performance in meta-learning are still limited. Although DNNs have so many parameters that can completely fit all training samples from all tasks, it is unclear why such an overfitted solution can still generalize well, which seems to defy the classical knowledge bias-variance-tradeoff (Bishop, 2006; Hastie et al., 2009; Stein, 1956; James & Stein, 1992; LeCun et al., 1991; Tikhonov, 1943).

The recent studies on the "benign overfitting" and "double-descent" phenomena in classical (single-task) linear regression have brought new insights on the generalization performance of overfitted solutions. Specifically, "benign overfitting" and "double descent" describe the phenomenon that the test error descends again in the overparameterized regime in linear regression setup (Belkin et al., 2018; 2019; Bartlett et al., 2020; Hastie et al., 2019; Muthukumar et al., 2019; Ju et al., 2020; Mei & Montanari, 2019). Depending on different settings, the shape and the properties of the descent curve of the test error can differ dramatically. For example, Ju et al. (2020) showed that the min $\ell_1$-

norm overfitted solution has a very different descent curve compared with the min $\ell_2$-norm overfitted solution. A more detailed review of this line of work can be found in Appendix A.

Compared to the classical (single-task) linear regression, model-agnostic meta-learning (MAML) (Finn et al., 2017; Finn, 2018), which is a popular algorithm for meta-learning, differs in many aspects. First, the training process of MAML involves task-specific gradient descent inner training and outer training for all tasks. Second, there are some new parameters to consider in meta-training, such as the number of tasks and the diversity/fluctuation of the ground truth of each training task. These distinct parts imply that we cannot directly apply the existing analysis of "benign overfitting" and "double-descent" on single-task linear regression in meta-learning. Thus, it is still unclear whether meta-learning also has a similar "double-descent" phenomenon, and if yes, how the shape of the descent curve of overfitted meta-learning is affected by system parameters.

A few recent works have studied the generalization performance of meta-learning. In Bernacchia (2021), the expected value of the test error for the overfitted min $\ell_2$-norm solution of MAML is provided for the asymptotic regime where the number of features and the number of training samples go to infinity. In Chen et al. (2022), a high probability upper bound on the test error of a similar overfitted min $\ell_2$-norm solution is given in the non-asymptotic regime. However, the eigenvalues of weight matrices that appeared in the bound of Chen et al. (2022) are coupled with the system parameters such as the number of tasks, samples, and features, so that their bound is not fully expressed in terms of the scaling orders of those system parameters. This makes it hard to explicitly characterize the shape of the double-descent or analyze the tightness of the bound. In Huang et al. (2022), the authors focus on the generalization error during the SGD process when the training error is not zero, which is different from our focus on the overfitted solutions that make the training error equal to zero (i.e., the interpolators). (A more comprehensive introduction on related works can be found in Appendix A.) All of these works let the number of model features in meta-learning equal to the number of true features, which cannot be used to analyze the shape of the double-descent curve that requires the number of features used in the learning model to change freely without affecting the ground truth (just like the setup used in many works on single-task linear regression, e.g., Belkin et al. (2020); Ju et al. (2020)).

To fill the gap, we study the generalization performance of overfitted meta-learning, especially in quantifying how the test error changes with the number of features. As the initial step towards the DNNs' setup, we consider the overfitted min $\ell_2$-norm solution of MAML using a linear model with Gaussian features. We first quantify the error caused by the one-step gradient adaption for the test task, with which we provide useful insights on 1) practically choosing the step size of the test task and quantifying the gap with the optimal (but not practical) choice, and 2) how overparameterization can affect the noise error and task diversity error in the test task. We then provide an explicit high-probability upper bound (under reasonable tightness) on the error caused by the meta-training for training tasks (we call this part "model error") in the non-asymptotic regime where all parameters are finite. With this upper bound and simulation results, we confirm the benign overfitting in meta-learning by comparing the model error of the overfitted solution with the underfitted solution. We further characterize some interesting properties of the descent curve. For example, we show that the descent is easier to observe when the noise and task diversity are large, and sometimes has a descent floor. Compared with the classical (single-task) linear regression where the double-descent phenomenon critically depends on non-zero noise, we show that meta-learning can still have the double-descent phenomenon even under zero noise as long as the task diversity is non-zero.

## 2 System Model

In this section, we introduce the system model of meta-learning along with the related symbols. For the ease of reading, we also summarize our notations in Table 2 in Appendix B.

### 2.1 Data Generation Model

We adopt a meta-learning setup with linear tasks first studied in Bernacchia (2021) as well as a few recent follow-up works Chen et al. (2022); Huang et al. (2022). However, in their formulation, the number of features and true model parameters are the same. Such a setup fails to capture the prominent effect of overparameterized models in practice where the number of model parameters

is much larger than the number of actual feature parameters. Thus, in our setup, we introduce an additional parameter $s$ to denote the number of true features and allow $s$ to be different and much smaller than the number $p$ of model parameters to capture the effect of overparameterization. In this way, we can fix $s$ and investigate how $p$ affects the generalization performance. We believe this setup is closer to reality because determining the number of features to learn is controllable, while the number of actual features of the ground truth is fixed and not controllable.

We consider $m$ training tasks. For the $i$-th training task (where $i = 1, 2, \cdots, m$), the ground truth is a linear model represented by $y = \boldsymbol{x}_s^T \boldsymbol{w}_s^{(i)} + \epsilon$, where $s$ denotes the number of features, $\boldsymbol{x}_s \in \mathbb{R}^s$ is the underlying features, $\epsilon \in \mathbb{R}$ denotes the noise, and $y \in \mathbb{R}$ denotes the output. When we collect the features of data, since we do not know what are the true features, we usually choose more features than $s$, i.e., choose $p$ features where $p \geq s$ to make sure that these $p$ features include all $s$ true features. For analysis, without loss of generality, we let the first $s$ features of all $p$ features be the true features. Therefore, although the ground truth model only has $s$ features, we can alternatively expressed it with $p$ features as $y = \boldsymbol{x}^T \boldsymbol{w}^{(i)} + \epsilon$ where the first $s$ elements of $\boldsymbol{x} \in \mathbb{R}^p$ is $\boldsymbol{x}_s$ and $\boldsymbol{w}^{(i)} := \begin{bmatrix} \boldsymbol{w}_s^{(i)} \\ \boldsymbol{0} \end{bmatrix} \in \mathbb{R}^p$. The collected data are split into two parts with $n_t$ training data and $n_v$ validation data. With those notations, we write the data generation model into matrix equations as

$$\boldsymbol{y}^{t(i)} = \mathbf{X}^{t(i)^T} \boldsymbol{w}^{(i)} + \boldsymbol{\epsilon}^{t(i)}, \quad \boldsymbol{y}^{v(i)} = \mathbf{X}^{v(i)^T} \boldsymbol{w}^{(i)} + \boldsymbol{\epsilon}^{v(i)}, \tag{1}$$

where each column of $\mathbf{X}^{t(i)} \in \mathbb{R}^{p \times n_t}$ corresponds to the input of each training sample, each column of $\mathbf{X}^{v(i)} \in \mathbb{R}^{p \times n_v}$ corresponds to the input of each validation sample, $\boldsymbol{y}^{t(i)} \in \mathbb{R}^{n_t}$ denotes the output of all training samples, $\boldsymbol{y}^{v(i)} \in \mathbb{R}^{n_v}$ denotes the output of all validation samples, $\boldsymbol{\epsilon}^{t(i)} \in \mathbb{R}^{n_t}$ denotes the noise in training samples, and $\boldsymbol{\epsilon}^{v(i)}$ denotes the noise invalidation samples.

Similarly to the training tasks, we denote the ground truth of the test task by $\boldsymbol{w}_s^r \in \mathbb{R}^s$, and thus $y = \boldsymbol{x}_s^T \boldsymbol{w}_s^r$. Let $\boldsymbol{w}^r = \begin{bmatrix} \boldsymbol{w}_s^r \\ \boldsymbol{0} \end{bmatrix} \in \mathbb{R}^p$. Let $n_r$ denote the number of training samples for the test task, and let each column of $\mathbf{X}^r \in \mathbb{R}^{p \times n_r}$ denote the input of each training sample. Similar to Eq. (1), we then have

$$\boldsymbol{y}^r = (\mathbf{X}^r)^T \boldsymbol{w}^r + \boldsymbol{\epsilon}^r, \tag{2}$$

where $\boldsymbol{\epsilon}^r \in \mathbb{R}^{n_r}$ denotes the noise and each element of $\boldsymbol{y}^r \in \mathbb{R}^{n_r}$ corresponds to the output of each training sample.

In order to simplify the theoretical analysis, we adopt the following two assumptions. Assumption 1 is commonly taken in the theoretical study of the generalization performance, e.g., Ju et al. (2020); Bernacchia (2021). Assumption 2 is less restrictive (no requirement to be any specific distribution) and a similar one is also used in Bernacchia (2021).

**Assumption 1** (Gaussian features and noise). *We adopt* i.i.d. *Gaussian features* $\boldsymbol{x} \sim \mathcal{N}(0, \mathbf{I}_p)$ *and assume* i.i.d. *Gaussian noise. We use* $\sigma$ *and* $\sigma_r$ *to denote the standard deviation of the noise for training tasks and the test task, respectively. In other words,* $\boldsymbol{\epsilon}^{t(i)} \sim \mathcal{N}(\mathbf{0}, \sigma^2 \mathbf{I}_{n_t})$, $\boldsymbol{\epsilon}^{v(i)} \sim \mathcal{N}(\mathbf{0}, \sigma^2 \mathbf{I}_{n_v})$ *for all* $i = 1, \cdots, m$, *and* $\boldsymbol{\epsilon}^r \sim \mathcal{N}(\mathbf{0}, \sigma_r^2 \mathbf{I}_{n_r})$.

**Assumption 2** (Diversity/fluctuation of unbiased ground truth). *The ground truth* $\boldsymbol{w}_s^r$ *and* $\boldsymbol{w}_s^{(i)}$ *for all* $i = 1, 2, \cdots, m$ *share the same mean* $\boldsymbol{w}_s^0$, *i.e.,* $\mathbb{E}[\boldsymbol{w}_s^{(i)}] = \boldsymbol{w}_s^0 = \mathbb{E}[\boldsymbol{w}_s^r]$. *For the* $i$-th training task, elements of the true model parameter $\boldsymbol{w}_s^{(i)} \in \mathbb{R}^s$ are independent, i.e.,

$$\mathbb{E}[(\boldsymbol{w}_s^{(i)} - \boldsymbol{w}_s^0)^T (\boldsymbol{w}_s^{(i)} - \boldsymbol{w}_s^0)] = \boldsymbol{\Lambda}_{(i)} := \mathrm{diag}\left( (\nu_{(i),1})^2, (\nu_{(i),2})^2, \cdots, (\nu_{(i),s})^2 \right).$$

*Let* $\nu_{(i)} := \sqrt{\mathrm{Tr}(\boldsymbol{\Lambda}_{(i)})}$, $\nu := \sqrt{\sum_{i=1}^m \nu_{(i)}^2 / m}$, $\nu_r := \sqrt{\mathbb{E}\|\boldsymbol{w}_s^r - \boldsymbol{w}_0\|_2^2}$, *and* $\boldsymbol{w}_0 := \begin{bmatrix} \boldsymbol{w}_s^0 \\ \boldsymbol{0} \end{bmatrix} \in \mathbb{R}^p$.

## 2.2 MAML PROCESS

We consider the MAML algorithm (Finn et al., 2017; Finn, 2018), the objective of which is to train a good initial model parameter among many tasks, which can adapt quickly to reach a desirable model parameter for a target task. MAML generally consists of inner-loop training for each individual task and outer-loop training across multiple tasks. To differentiate the ground truth parameters $\boldsymbol{w}^{(i)}$, we use $(\hat{\cdot})$ (e.g., $\hat{\boldsymbol{w}}^{(i)}$) to indicate that the parameter is the training result.

In the inner-loop training, the model parameter for every individual task is updated from a common meta parameter $\hat{\boldsymbol{w}}$. Specifically, for the $i$-th training task ($i = 1, \ldots, m$), its model parameter $\hat{\boldsymbol{w}}^{(i)}$ is updated via a one-step gradient descent of the loss function based on its training data $\mathbf{X}^{t(i)}$:

$$\hat{\boldsymbol{w}}^{(i)} := \hat{\boldsymbol{w}} - \frac{\alpha_t}{n_t} \frac{\partial \mathcal{L}_{\text{inner}}^{(i)}}{\partial \hat{\boldsymbol{w}}}, \quad \mathcal{L}_{\text{inner}}^{(i)} := \frac{1}{2} \left\| \boldsymbol{y}^{t(i)} - \mathbf{X}^{t(i)T} \hat{\boldsymbol{w}} \right\|_2^2. \tag{3}$$

where $\alpha_t \geq 0$ denotes the step size.

In the outer-loop training, the meta loss $\mathcal{L}^{\text{meta}}$ is calculated based on the validation samples of all training tasks as follows:

$$\mathcal{L}^{\text{meta}} := \frac{1}{mn_v} \sum_{i=1}^{m} \mathcal{L}_{\text{outer}}^{(i)}, \quad \text{where} \quad \mathcal{L}_{\text{outer}}^{(i)} := \frac{1}{2} \left\| \boldsymbol{y}^{v(i)} - \mathbf{X}^{v(i)T} \hat{\boldsymbol{w}}^{(i)} \right\|_2^2. \tag{4}$$

The common (i.e., meta) parameter $\hat{\boldsymbol{w}}$ is then trained to minimize the meta loss $\mathcal{L}^{\text{meta}}$.

At the test stage, we use the test loss $\mathcal{L}_{\text{inner}}^r := \frac{1}{2} \left\| \boldsymbol{y}^r - \mathbf{X}^{rT} \hat{\boldsymbol{w}} \right\|_2^2$ and adapt the meta parameter $\hat{\boldsymbol{w}}$ via a single gradient descent step to obtain a desirable model parameter $\hat{\boldsymbol{w}}^r$, i.e., $\hat{\boldsymbol{w}}^r := \hat{\boldsymbol{w}} - \frac{\alpha_r}{n_r} \frac{\partial \mathcal{L}_{\text{inner}}^r}{\partial \hat{\boldsymbol{w}}}$ where $\alpha_r \geq 0$ denotes the step size. The squared test error for any input $\boldsymbol{x}$ is given by

$$\mathcal{L}_{\text{test}}(\boldsymbol{x}, \boldsymbol{w}^r; \hat{\boldsymbol{w}}^r) := \left\| \boldsymbol{x}^T \boldsymbol{w}^r - \boldsymbol{x}^T \hat{\boldsymbol{w}}^r \right\|_2^2. \tag{5}$$

## 2.3 SOLUTIONS OF MINIMIZING META LOSS

The meta loss in Eq. (4) depends on the meta parameter $\hat{\boldsymbol{w}}$ via the inner-loop loss in Eq. (3). It can be shown (see the details in Appendix C) that $\mathcal{L}^{\text{meta}}$ can be expressed as:

$$\mathcal{L}^{\text{meta}} = \frac{1}{2mn_v} \left\| \gamma - \mathbf{B} \hat{\boldsymbol{w}} \right\|_2^2, \tag{6}$$

where $\gamma \in \mathbb{R}^{(mn_v) \times 1}$ and $\mathbf{B} \in \mathbb{R}^{(mn_v) \times p}$ are respectively stacks of $m$ vectors and matrices given by

$$\gamma := \begin{bmatrix} \boldsymbol{y}^{v(1)} - \frac{\alpha_t}{n_t} \mathbf{X}^{v(1)T} \mathbf{X}^{t(1)} \boldsymbol{y}^{t(1)} \\ \boldsymbol{y}^{v(2)} - \frac{\alpha_t}{n_t} \mathbf{X}^{v(2)T} \mathbf{X}^{t(2)} \boldsymbol{y}^{t(2)} \\ \vdots \\ \boldsymbol{y}^{v(m)} - \frac{\alpha_t}{n_t} \mathbf{X}^{v(m)T} \mathbf{X}^{t(m)} \boldsymbol{y}^{t(m)} \end{bmatrix}, \quad \mathbf{B} := \begin{bmatrix} \mathbf{X}^{v(1)T} \left( \mathbf{I}_p - \frac{\alpha_t}{n_t} \mathbf{X}^{t(1)} \mathbf{X}^{t(1)T} \right) \\ \mathbf{X}^{v(2)T} \left( \mathbf{I}_p - \frac{\alpha_t}{n_t} \mathbf{X}^{t(2)} \mathbf{X}^{t(2)T} \right) \\ \vdots \\ \mathbf{X}^{v(m)T} \left( \mathbf{I}_p - \frac{\alpha_t}{n_t} \mathbf{X}^{t(m)} \mathbf{X}^{t(m)T} \right) \end{bmatrix}. \tag{7}$$

By observing Eq. (6) and the structure of $\mathbf{B}$, we know that $\min_{\hat{\boldsymbol{w}}} \mathcal{L}^{\text{meta}}$ has a unique solution almost surely when the learning model is *underparameterized*, i.e., $p \leq mn_v$. However, in the real-world application of meta learning, an *overparameterized* model is of more interest due to the success of the DNNs. Therefore, in the rest of this paper, we mainly focus on the *overparameterized* situation, i.e., when $p > mn_v$ (so the meta training loss can decrease to zero). In this case, there exist (almost surely) numerous $\hat{\boldsymbol{w}}$ that make the meta loss become zero, i.e., interpolators of training samples.

Among all overfitted solutions, we are particularly interested in the min $\ell_2$-norm solution, since it corresponds to the solution of gradient descent that starts at zero in a linear model. Specially, the min $\ell_2$-norm overfitted solution $\hat{\boldsymbol{w}}_{\ell_2}$ is defined as

$$\hat{\boldsymbol{w}}_{\ell_2} := \arg\min_{\hat{\boldsymbol{w}}} \left\| \hat{\boldsymbol{w}} \right\|_2 \quad \text{subject to} \quad \mathbf{B}\hat{\boldsymbol{w}} = \gamma. \tag{8}$$

In this paper, we focus on quantifying the generalization performance of this min $\ell_2$-norm solution with the metric in Eq. (5).

## 3 MAIN RESULTS

To analyze the generalization performance, we first decouple the overall test error into two parts: i) the error caused by the one-step gradient adaption for the test task, and ii) the error caused by the meta training for the training tasks. The following lemma quantifies such decomposition.

**Lemma 1.** *With Assumptions 1 and 2, for any learning result $\hat{w}$ (i.e., regardless how we train $\hat{w}$), the expected squared test error is*

$$\mathop{\mathbb{E}}_{\boldsymbol{x},\mathbf{X}^r,\boldsymbol{\epsilon}^r,\boldsymbol{w}^r} \mathcal{L}_{test}(\boldsymbol{x},\boldsymbol{w}^r;\hat{\boldsymbol{w}}^r) = f_{test}\left(\|\hat{\boldsymbol{w}} - \boldsymbol{w}_0\|_2^2\right),$$

*where $f_{test}(\zeta) := \left((1-\alpha_r)^2 + \frac{p+1}{n_r}\alpha_r^2\right)\left(\zeta + \nu_r^2\right) + \frac{\alpha_r^2 p}{n_r}\sigma_r^2.$*

Notice that in the meta-learning phase, the ideal situation is that the learned meta parameter $\hat{\boldsymbol{w}}$ perfectly matches the mean $\boldsymbol{w}_0$ of true parameters. Thus, the term $\|\hat{\boldsymbol{w}} - \boldsymbol{w}_0\|_2^2$ characterizes how well the meta-training goes. The rest of the terms in the Lemma 1 then characterize the effect of the one-step training for the test task. The proof of Lemma 1 is in Appendix H. Note that the expression of $f_{test}(\zeta)$ coincides with Eq. (65) of Bernacchia (2021). However, Bernacchia (2021) uses a different setup and does not analyze its implication as what we will do in the rest of this section.

### 3.1 UNDERSTANDING THE TEST ERROR

**Proposition 1.** *We have*

$$\tfrac{p+1}{n_r+p+1}(\|\hat{\boldsymbol{w}} - \boldsymbol{w}_0\|_2^2 + \nu_r^2) \le \min_{\alpha_r} \mathbb{E}_{\boldsymbol{x},\mathbf{X}^r,\boldsymbol{\epsilon}^r,\boldsymbol{w}^r}[\mathcal{L}_{test}(\boldsymbol{x},\boldsymbol{w}^r;\hat{\boldsymbol{w}}^r)] \le \|\hat{\boldsymbol{w}} - \boldsymbol{w}_0\|_2^2 + \nu_r^2.$$

*Further, by letting $\alpha_r = \frac{n_r}{n_r+p+1}$ (which is optimal when $\sigma_r = 0$), we have*

$$\mathbb{E}_{\boldsymbol{x},\mathbf{X}^r,\boldsymbol{\epsilon}^r,\boldsymbol{w}^r}[\mathcal{L}_{test}(\boldsymbol{x},\boldsymbol{w}^r;\hat{\boldsymbol{w}}^r)] = \tfrac{p+1}{n_r+p+1}(\|\hat{\boldsymbol{w}} - \boldsymbol{w}_0\|_2^2 + \nu_r^2) + \tfrac{n_r p}{(n_r+p+1)^2}\sigma_r^2.$$

The derivation of Proposition 1 is in Appendix H.1. Some insights from Proposition 1 are as follows.

**1) Optimal $\alpha_r$ does not help much when overparameterized.** For meta-learning, the number of training samples $n_r$ for the test task is usually small (otherwise there is no need to do meta-learning). Therefore, in Proposition 1, when overparameterized (i.e., $p$ is relatively large), the coefficient $\frac{p+1}{n_r+p+1}$ of $(\|\hat{\boldsymbol{w}} - \boldsymbol{w}_0\|_2^2 + \nu_r^2)$ is close to 1. On the other hand, the upper bound $\|\hat{\boldsymbol{w}} - \boldsymbol{w}_0\|_2^2 + \nu_r^2$ can be achieved by letting $\alpha_r = 0$, which implies that the effect of optimally choosing $\alpha_r$ is limited under this circumstance. Further, calculating the optimal $\alpha_r$ requires the precise values of $\|\hat{\boldsymbol{w}} - \boldsymbol{w}_0\|_2^2$, $\nu_r^2$, and $\sigma_r^2$. However, those values are usually impossible/hard to get beforehand. Hence, we next investigate how to choose an easy-to-obtain $\alpha_r$.

**2) Choosing $\alpha_r = n_r/(n_r + p + 1)$ is practical and good enough when overparameterized.** Choosing $\alpha_r = n_r/(n_r + p + 1)$ is practical since $n_r$ and $p$ are known. By Proposition 1, the gap between choosing $\alpha_r = n_r/(n_r + p + 1)$ and choosing optimal $\alpha_r$ is at most $\frac{n_r p}{(n_r+p+1)^2}\sigma_r^2 \le \frac{n_r}{p}\sigma_r^2$. When $p$ increases, this gap will decrease to zero. In other words, when heavily overparameterized, choosing $\alpha_r = n_r/(n_r + p + 1)$ is good enough.

**3) Overparameterization can reduce the noise error to zero but cannot diminish the task diversity error to zero.** In the expression of $f_{test}\left(\|\hat{\boldsymbol{w}} - \boldsymbol{w}_0\|_2^2\right)$ in Lemma 1, there are two parts related to the test task: the noise error (the term for $\sigma_r$) and the task diversity error (the term for $\nu_r$). By Proposition 1, even if we choose the optimal $\alpha_r$, the term of $\nu_r^2$ will not diminish to zero when $p$ increases. In contrast, by letting $\alpha_r = n_r/(n_r + p + 1)$, the noise term $\frac{n_r p}{(n_r+p+1)^2}\sigma_r^2 \le \frac{n_r}{p}\sigma_r^2$ will diminish to zero when $p$ increases to infinity.

### 3.2 CHARACTERIZATION OF MODEL ERROR

Since we already have Lemma 1, to estimate the generalization error, it only remains to estimate $\|\hat{\boldsymbol{w}} - \boldsymbol{w}_0\|_2^2$, which we refer as *model error*. The following Theorem 1 gives a high probability upper bound on the model error.

**Theorem 1.** *Under Assumptions 1 and 2, when $\min\{p, n_t\} \ge 256$, we must have*

$$\Pr_{\mathbf{X}^{t(1:m)},\mathbf{X}^{v(1:m)}}\left\{\mathbb{E}_{\boldsymbol{w}^{(1:m)},\boldsymbol{\epsilon}^{t(1:m)},\boldsymbol{\epsilon}^{v(1:m)}}\|\hat{\boldsymbol{w}}_{\ell_2} - \boldsymbol{w}_0\|_2^2 \le b_w\right\} \ge 1 - \eta,$$

*where $b_w := b_{\boldsymbol{w}_0} + b_w^{ideal}$ and $\eta := \frac{27m^2 n_v^2}{\min\{p,n_t\}^{0.4}}.$*

The value of $b_{\boldsymbol{w}_0}$ and $b_w^{\text{ideal}}$ are completely determined by the finite (i.e., non-asymptotic region) system parameters $p, s, m, n_t, n_v, \|\boldsymbol{w}_0\|_2^2, \nu^2, \sigma^2$, and $\alpha_t$. The precise expression will be given in Section 4, along with the proof sketch of Theorem 1. Notice that although $b_w$ is only an upper bound, we will show in Section 4 that each component of this upper bound is relatively tight.

Theorem 1 differs from the result in Bernacchia (2021) in two main aspects. First, Theorem 1 works in the non-asymptotic region where $p$ and $n_t$ are both finite, whereas their result holds only in the asymptotic regime where $p, n_t \to \infty$. In general, a non-asymptotic result is more powerful than an asymptotic result in order to understand and characterize how the generalization performance changes as the model becomes more overparameterized. Second, the nature of our bound is in high probability with respect to the training and validation data, which is much stronger than their result in expectation. Due to the above key differences, our derivation of the bound is very different and much harder.

As we will show in Section 4, the detailed expressions of $b_{\boldsymbol{w}_0}$ and $b_w^{\text{ideal}}$ are complicated. In order to derive some useful interpretations, we provide a simpler form by approximation[1] for an overparameterized regime, where $\alpha_t p \ll 1$ and $\min\{p, n_t\} \gg m n_v$. In such a regime, we have

$$b_{\boldsymbol{w}_0} \approx \frac{p - m n_v}{p} \|\boldsymbol{w}_0\|_2^2, \quad b_w^{\text{ideal}} \approx \frac{b_\delta}{p - C_4 m n_v}, \tag{9}$$

where $b_\delta \approx m n_v \left( (1 + \frac{C_1}{n_t})\sigma^2 + C_2(1 + \frac{C_3}{n_t})\nu^2 \right)$, and $C_1$ to $C_4$ are some constants. It turns out that a number of interesting insights can be obtained from the simplified bounds above.

**1) Heavier overparameterization reduces the negative effects of noise and task diversity.** From Eq. (9), we know that when $p$ increases, $b_w^{\text{ideal}}$ decreases to zero. Notice that $b_w^{\text{ideal}}$ is the only term that is related to $\sigma^2$ and $\nu^2$, i.e., $b_w^{\text{ideal}}$ corresponds to the negative effect of noise and task diversity/fluctuation. Therefore, we can conclude that using more features/parameters can reduce the negative effect of noise and task diversity/fluctuation.

Interestingly, $b_w^{\text{ideal}}$ can be interpreted as the model error for the *ideal interpolator* $\hat{\boldsymbol{w}}_{\text{ideal}}$ defined as

$$\hat{\boldsymbol{w}}_{\text{ideal}} := \arg\min_{\hat{\boldsymbol{w}}} \|\hat{\boldsymbol{w}} - \boldsymbol{w}_0\|_2^2 \quad \text{subject to} \quad \mathbf{B}\hat{\boldsymbol{w}} = \gamma.$$

Differently from the min $\ell_2$-norm overfitted solution in Eq. (8) that minimizes the norm of $\hat{\boldsymbol{w}}$, the ideal interpolator minimizes the distance between $\hat{\boldsymbol{w}}$ and $\boldsymbol{w}_0$, i.e., the model error (this is why we define it as the ideal interpolator). The following proposition states that $b_w^{\text{ideal}}$ corresponds to the model error of $\hat{\boldsymbol{w}}_{\text{ideal}}$.

**Proposition 2.** *When* $\min\{p, n_t\} \geq 256$, *we must have*

$$\Pr_{\mathbf{X}^{v(1:m)}, \mathbf{X}^{t(1:m)}} \left\{ \mathbb{E}_{\boldsymbol{w}^{(1:m)}, \boldsymbol{\epsilon}^{t(1:m)}, \boldsymbol{\epsilon}^{v(1:m)}} \|\hat{\boldsymbol{w}}_{ideal} - \boldsymbol{w}_0\|_2^2 \leq b_w^{ideal} \right\} \geq 1 - \frac{26 m^2 n_v^2}{\min\{p, n_t\}^{0.4}},$$

**2) Overfitting is beneficial to reduce model error for the ideal interpolator.** Although calculating $\hat{\boldsymbol{w}}_{\text{ideal}}$ is not practical since it needs to know the value of $\boldsymbol{w}_0$, we can still use it as a benchmark that describes the best performance among all overfitted solutions. From the previous analysis, we have already shown that $b_w^{\text{ideal}} \to 0$ when $p \to \infty$, i.e., the model error of the ideal interpolator decreases to 0 when the number of features grows. Thus, we can conclude that overfitting is beneficial to reduce the model error for the ideal interpolator. This can be viewed as evidence that overfitting itself should not always be viewed negatively, which is consistent with the success of DNNs in meta-learning.

**3) The descent curve is easier to observe under large noise and task diversity, and the curve sometimes has a descent floor.** In Eq. (9), when $p$ increases, $b_{\boldsymbol{w}_0}$ increases but $b_w^{\text{ideal}}$ decreases. When $\sigma$ and $\nu$ becomes larger, $b_w^{\text{ideal}}$ becomes larger while $b_{\boldsymbol{w}_0}$ does not change, so that the descent of $b_w^{\text{ideal}}$ contributes more to the trend of $b_w = b_{\boldsymbol{w}_0} + b_w^{\text{ideal}}$ (i.e., the overall model error). By further calculating the derivative of $b_w$ with respect to $p$ (see details in Appendix D), we observe that, if $g \geq 1$, then $b_w$ always decreases for $p > C_4 m n_v$. If $g := \frac{b_\delta}{m n_v \|\boldsymbol{w}_0\|_2^2} < 1$, then $b_w$ decreases

---

[1]The approximation considers only the dominating terms and treats logarithm terms as constants (since they change slowly). Notice that our approximation here is different from an asymptotic result, since the precision of such approximation in the finite regime can be precisely quantified, whereas an asymptotic result can only estimate the precision in the order of magnitude in the infinite regime as typically denoted by $O(\cdot)$ notations.

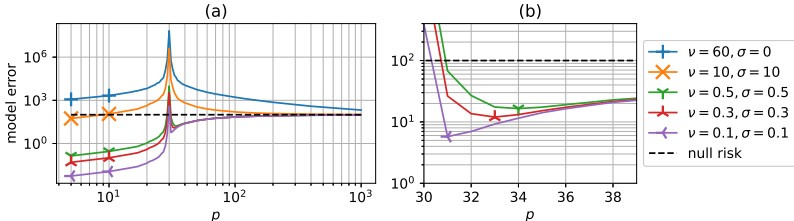

Figure 1: The model error w.r.t. different values of $\nu$ and $\sigma$, where $m = 10$, $n_t = 50$, $n_v = 3$, $s = 5$, $\|\boldsymbol{w}_0\|_2^2 = 100$, and $\alpha_t = \frac{0.02}{p}$. Subfigure (b) is a copy of subfigure (a) that zooms in the descent floor. Every point is the average of 100 random simulations. The markers in subfigure (b) indicate the descent floor for each curve.

when $p \in (C_4 m n_v, \frac{C_4 m n_v}{1-\sqrt{g}})$, and increases when $p > \frac{C_4 m n_v}{1-\sqrt{g}}$. Thus, the minimal value (i.e., the floor value) of $b_w$ is equal to $\|\boldsymbol{w}_0\|_2^2 \left(1 - \frac{(1-\sqrt{g})^2}{C_4}\right)$, which is achieved at $p = \frac{C_4 m n_v}{1-\sqrt{g}}$. This implies that the descent curve of the model error has a floor only when $b_\delta$ is small, i.e., when the noise and task diversity are small. Notice that the threshold $\frac{C_4 m n_v}{1-\sqrt{g}}$ and the floor value $\|\boldsymbol{w}_0\|_2^2 \left(1 - \frac{(1-\sqrt{g})^2}{C_4}\right)$ increase as $g$ increases. Therefore, we anticipate that as $\nu$ and $\sigma$ increase, the descent floor value and its location both increase.

In Fig. 1, we draw the curve of the model error with respect to $p$ for the min $\ell_2$-norm solution. In subfigure (a), the blue curve (with the marker "+") and the yellow curve (with the marker "×") have relatively large $\nu$ and/or $\sigma$. These two curves always decrease in the overparameterized region $p > m n_v$ and have no descent floor. In contrast, the rest three curves (purple, red, green) in subfigure (a) have descent floor since they have relatively small $\nu$ and $\sigma$. Subfigure (b) shows the location and the value of the descent floor. As we can see, when $\nu$ and $\sigma$ increase, the descent floor becomes higher and locates at larger $p$. These observations are consistent with our theoretical analysis.

In Appendix E.2, we provide a further experiment where we train a two-layer fully connected neural network over the MNIST data set. We observe that a descent floor still occurs. Readers can find more details about the experiment in Appendix E.2.

**4) Task diversity yields double descent under zero noise.** For a single-task classical linear regression model $y = \boldsymbol{x}^T \boldsymbol{w}_0 + \epsilon$, the authors of Belkin et al. (2020) study the overfitted min $\ell_2$-norm solutions $\boldsymbol{w}_{\ell_2,\text{single}}$ learned by interpolating $n$ training samples with $p \geq n + 2$ *i.i.d.* Gaussian features. The result in Belkin et al. (2020) shows that its expected model error is

$$\mathbb{E} \|\boldsymbol{w}_{\ell_2,\text{single}} - \boldsymbol{w}_0\|_2^2 = \frac{p - n}{p} \|\boldsymbol{w}_0\|_2^2 + \frac{n}{p - n - 1} \sigma^2.$$

We find that both meta-learning and the single-task regression have similar bias terms $\frac{p - m n_v}{p} \|\boldsymbol{w}_0\|_2^2$ and $\frac{p - n}{p} \|\boldsymbol{w}_0\|_2^2$, respectively. When $p \to \infty$, these bias terms increase to $\|\boldsymbol{w}_0\|_2^2$, which corresponds to the *null risk* (the error of a model that always predicts zero). For the single-task regression, the remaining term $\frac{n}{p-n-1} \sigma^2 \approx \frac{n}{p} \sigma^2$ when $p \gg n$, which contributes to the descent of generalization error when $p$ increases. On the other hand, if there is no noise ($\sigma = 0$), then the benign overfitting/double descent will disappear for the single-task regression. In contrast, for meta-learning, the term that contributes to benign overfitting is $b_w^{\text{ideal}}$. As we can see in the expression of $b_w^{\text{ideal}}$ in Eq. (9), even if the noise is zero, as long as there exists task diversity/fluctuation (i.e., $\nu > 0$), the descent of the model error with respect to $p$ should still exist. This is also confirmed by Fig. 1(a) with the descent blue curve (with marker "+") of $\nu = 60$ and $\sigma = 0$.

**5) Overfitted solution can generalize better than underfitted solution.** Let $\hat{\boldsymbol{w}}_{\ell_2}^{(p=1)}$ denote the solution when $p = s = 1$. We have

$$\left\|\hat{\boldsymbol{w}}_{\ell_2}^{(p=1)} - \boldsymbol{w}_0\right\|_2^2 \approx \frac{\nu^2}{m} + \frac{\sigma^2 \alpha_t^2}{m} + \frac{\sigma^2}{(1-\alpha_t)^2 m n_v}.$$

The derivation is in Appendix O. Notice that $p = s$ means that all features are true features, which is ideal for an underparameterized solution. Compared to the model error of overfitted solution, there

is no bias term of $\boldsymbol{w}_0$, but the terms of $\nu^2$ and $\sigma^2$ are only discounted by $m$ and $n_v$. In contrast, for the overfitted solution when $p \gg mn_v$, the terms of $\nu^2$ and $\sigma^2$ are discounted by $p$. Since the value of $m$ and the value of $n_v$ are usually fixed or limited, while $p$ can be chosen freely and made arbitrarily large, the overfitted solution can do much better to mitigate the negative effect caused by noise and task divergence. This provides us a new insight that when $\sigma$, $\nu$ are large and $\|\boldsymbol{w}_0\|_2$ is small, the overfitted $\ell_2$-norm solution can have a much better overall generalization performance than the underparameterized solution. This is further verified in Fig. 1(a) by the descent blue curve (with marker "+"), where the first point of this curve with $p = 5 = s$ (in the underparameterized regime) has a larger test error than its last point with $p = 1000$ (in the overparameterized regime).

## 4 TIGHTNESS OF THE BOUND AND ITS PROOF SKETCH

We now present the main ideas of proving Theorem 1 and theoretically explain why the bound in Theorem 1 is reasonably tight by showing each part of the bound is tight. (Numerical verification of the tightness is provided in Appendix E.1.) The expressions of $b_{\boldsymbol{w}_0}$ and $b_w^{\text{ideal}}$ (along with other related quantities) in Theorem 1 will be defined progressively as we sketch our proof. Readers can also refer to the beginning of Appendix I for a full list of the definitions of these quantities.

We first re-write $\|\boldsymbol{w}_0 - \hat{\boldsymbol{w}}_{\ell_2}\|_2^2$ into terms related to $\mathbf{B}$. When $\mathbf{B}$ is full row-rank (which holds almost surely when $p > mn_v$), we have

$$\hat{\boldsymbol{w}}_{\ell_2} = \mathbf{B}^T \left( \mathbf{B}\mathbf{B}^T \right)^{-1} \gamma. \tag{10}$$

Define $\delta\gamma$ as

$$\delta\gamma := \gamma - \mathbf{B}\boldsymbol{w}_0. \tag{11}$$

Doing some algebra transformations (details in Appendix F), we have

$$\|\boldsymbol{w}_0 - \hat{\boldsymbol{w}}_{\ell_2}\|_2^2 = \underbrace{\left\| \left( \mathbf{I}_p - \mathbf{B}^T(\mathbf{B}\mathbf{B}^T)^{-1}\mathbf{B} \right) \boldsymbol{w}_0 \right\|_2^2}_{\text{Term 1}} + \underbrace{\left\| \mathbf{B}^T(\mathbf{B}\mathbf{B}^T)^{-1}\delta\gamma \right\|_2^2}_{\text{Term 2}} \tag{12}$$

To estimate the model error, we take a divide-and-conquer strategy and provide a set of propositions as follows to estimate Term 1 and Term 2 in Eq. (12).

**Proposition 3.** *Define*

$$b_{\boldsymbol{w}_0} := \frac{(p - mn_v) + 2\sqrt{(p - mn_v)\ln p} + 2\ln p}{p - 2\sqrt{p\ln p}} \|\boldsymbol{w}_0\|_2^2,$$

$$\tilde{b}_{\boldsymbol{w}_0} := \frac{(p - mn_v) - 2\sqrt{(p - mn_v)\ln p}}{p + 2\sqrt{p\ln p} + 2\ln p} \|\boldsymbol{w}_0\|_2^2.$$

*When $p \geq mn_v$ and $p \geq 16$, We then have*

$$\Pr[\textit{Term 1 of Eq. (12)} \leq b_{\boldsymbol{w}_0}] \geq 1 - 2/p, \quad \Pr[\textit{Term 1 of Eq. (12)} \geq \tilde{b}_{\boldsymbol{w}_0}] \geq 1 - 2/p,$$

$$\mathop{\mathbb{E}}_{\mathbf{X}^{t(1:m)}, \mathbf{X}^{v(1:m)}} [\textit{Term 1 of Eq. (12)}] = \frac{p - mn_v}{p} \|\boldsymbol{w}_0\|_2^2.$$

Proposition 3 gives three estimates on Term 1 of Eq. (12): the upper bound $b_{\boldsymbol{w}_0}$, lower bound $\tilde{b}_{\boldsymbol{w}_0}$, and the mean value. If we omit all logarithm terms, then these three estimates are the same, which implies that our estimation on Term 1 of Eq. (12) is fairly precise. (Proof of Proposition 3 is in Appendix J.) It then remains to estimate Term 2 in Eq. (12). Indeed, this term is also the model error of the ideal interpolator. To see this, since $\mathbf{B}\hat{\boldsymbol{w}} = \gamma$, we have $\mathbf{B}(\hat{\boldsymbol{w}} - \boldsymbol{w}_0) = \gamma - \mathbf{B}\boldsymbol{w}_0 = \delta\gamma$. Thus, to get $\min \|\hat{\boldsymbol{w}} - \boldsymbol{w}_0\|_2^2$ as the ideal interpolator, we have $\hat{\boldsymbol{w}}_{\text{ideal}} - \boldsymbol{w}_0 = \mathbf{B}^T(\mathbf{B}\mathbf{B}^T)^{-1}\delta\gamma$. Therefore, we have

$$\|\hat{\boldsymbol{w}}_{\text{ideal}} - \boldsymbol{w}_0\|_2^2 = \left\| \mathbf{B}^T(\mathbf{B}\mathbf{B}^T)^{-1}\delta\gamma \right\|_2^2. \tag{13}$$

Now we focus on $\left\| \mathbf{B}^T(\mathbf{B}\mathbf{B}^T)^{-1}\delta\gamma \right\|_2^2$. By Lemma 6 in Appendix G.2, we have

$$\frac{\|\delta\gamma\|_2^2}{\lambda_{\max}(\mathbf{B}\mathbf{B}^T)} \leq \left\| \mathbf{B}^T(\mathbf{B}\mathbf{B}^T)^{-1}\delta\gamma \right\|_2^2 \leq \frac{\|\delta\gamma\|_2^2}{\lambda_{\min}(\mathbf{B}\mathbf{B}^T)}. \tag{14}$$

We have the following results about the eigenvalues of $\mathbf{B}\mathbf{B}^T$.

**Proposition 4.** *Define* $\alpha_t' := \frac{\alpha_t}{n_t}\left(\sqrt{p} + \sqrt{n_t} + \ln\sqrt{n_t}\right)^2$ *and*

$$b_{\text{eig,min}} := p + \left(\max\{0, 1 - \alpha_t'\}^2 - 1\right)n_t - \left((n_v + 1)\max\{\alpha_t', 1 - \alpha_t'\}^2 + 6mn_v\right)\sqrt{p}\ln p,$$

$$b_{\text{eig,max}} := p + \left(\max\{\alpha_t', 1 - \alpha_t'\}^2 - 1\right)n_t + \left((n_v + 1)\max\{\alpha_t', 1 - \alpha_t'\}^2 + 6mn_v\right)\sqrt{p}\ln p.$$

*When $p \geq n_t \geq 256$, we must have*

$$\Pr\left\{b_{\text{eig,min}} \leq \lambda_{\min}(\mathbf{B}\mathbf{B}^T) \leq \lambda_{\max}(\mathbf{B}\mathbf{B}^T) \leq b_{\text{eig,max}}\right\} \geq 1 - 23m^2 n_v^2/n_t^{0.4}.$$

**Proposition 5.** *Define*

$$c_{\text{eig,min}} := \max\{0, 1 - \alpha_t'\}^2 p - 2mn_v \max\{\alpha_t', 1 - \alpha_t'\}^2\sqrt{p\ln p},$$

$$c_{\text{eig,max}} := \max\{\alpha_t', 1 - \alpha_t'\}^2\left(p + (2mn_v + 1)\sqrt{p\ln p}\right).$$

*When $n_t \geq p \geq 256$, we have*

$$\Pr\left\{c_{\text{eig,min}} \leq \lambda_{\min}(\mathbf{B}\mathbf{B}^T) \leq \lambda_{\max}(\mathbf{B}\mathbf{B}^T) \leq c_{\text{eig,max}}\right\} \geq 1 - 16m^2 n_v^2/p^{0.4}.$$

To see how the upper and lower bounds of the eigenvalues of $\mathbf{B}\mathbf{B}^T$ match, consider $\alpha_t p \ll 1$, which implies $\alpha_t' \ll 1$, and the fact that $\sqrt{p}$ and $\ln p$ are lower order terms than $p$, then each of $b_{\text{eig,min}}, b_{\text{eig,max}}, c_{\text{eig,min}}, c_{\text{eig,max}}$ can be approximated by $p \pm \tilde{C}mn_v$ for some constant $\tilde{C}$. Further, when $p \gg mn_v$, all $b_{\text{eig,min}}, b_{\text{eig,max}}, c_{\text{eig,min}}, c_{\text{eig,max}}$ can be approximated by $p$, i.e., the upper and lower bounds of the eigenvalues of $\mathbf{B}\mathbf{B}^T$ match. Therefore, our estimation on $\lambda_{\max}(\mathbf{B}\mathbf{B}^T)$ and $\lambda_{\min}(\mathbf{B}\mathbf{B}^T)$ in Proposition 4 and Proposition 5 are fairly tight. (Proposition 4 is proved in Appendix L, and Proposition 5 is proved in Appendix M.) From Eq. (14), it remains to estimate $\|\delta\gamma\|_2^2$.

**Proposition 6.** *Define*

$$D := \left(\max\left\{\left|1 - \alpha_t\frac{n_t + 2\sqrt{n_t\ln(sn_t)} + 2\ln(sn_t)}{n_t}\right|, \left|1 - \alpha_t\frac{n_t - 2\sqrt{n_t\ln(sn_t)}}{n_t}\right|\right\}\right)^2,$$

$$b_\delta := mn_v\sigma^2\left(1 + \frac{\alpha_t^2 p(\ln n_t)^2\ln p}{n_t}\right) + mn_v\nu^2 \cdot 2\ln(sn_t)\cdot\left(D + \frac{\alpha_t^2(p-1)}{n_t}6.25(\ln(spn_t))^2\right).$$

*When $\min\{p, n_t\} \geq 256$, we must have*

$$\Pr_{\mathbf{X}^{t(1:m)}, \mathbf{X}^{v(1:m)}}\left\{\mathbb{E}_{\boldsymbol{w}^{(1:m)}, \boldsymbol{\epsilon}^{t(1:m)}, \boldsymbol{\epsilon}^{v(1:m)}}\|\delta\gamma\|_2^2 \leq b_\delta\right\} \geq 1 - \frac{5mn_v}{n_t} - \frac{2mn_v}{p^{0.4}}.$$

*We also have* $\mathbb{E}\|\delta\gamma\|_2^2 = mn_v\sigma^2\left(1 + \frac{\alpha_t^2 p}{n_t}\right) + \nu^2 mn_v\left((1 - \alpha_t)^2 + \frac{\alpha_t^2(p+1)}{n_t}\right)$, *where the expectation is on all random variables.*

Proposition 6 provides an upper bound $b_\delta$ on $\|\delta\gamma\|_2^2$ and an explicit form for $\mathbb{E}\|\delta\gamma\|_2^2$. By comparing $b_\delta$ and $\mathbb{E}\|\delta\gamma\|_2^2$, the differences are only some coefficients and logarithm terms. Thus, the estimation on $\|\delta\gamma\|_2^2$ in Proposition 6 is fairly tight. Proposition 6 is proved in Appendix N.

Combining Eq. (13), Eq. (14), Proposition 4, Proposition 5, and Proposition 6, we can get the result of Proposition 2 by the union bound, where $b_w^{\text{ideal}} := \frac{b_\delta}{\max\{b_{\text{eig,min}}\mathbb{1}_{\{p > n_t\}} + c_{\text{eig,min}}\mathbb{1}_{\{p \leq n_t\}}, 0\}}$. The detailed proof is in Appendix K. Then, by Eq. (13), Proposition 2, Proposition 3, and Eq. (12), we can get a high probability upper bound on $\|\boldsymbol{w}_0 - \hat{\boldsymbol{w}}_{\ell_2}\|_2^2$, i.e., Theorem 1. The detailed proof of Theorem 1 is in Appendix I. Notice that we can easily plug our estimation of the model error (Proposition 2 and Theorem 1) into Lemma 1 to get an estimation of the overall test error defined in Eq. (5), which is omitted in this paper by the limit of space.

## 5 CONCLUSION

We study the generalization performance of overfitted meta-learning under a linear model with Gaussian features. We characterize the descent curve of the model error for the overfitted min $\ell_2$-norm solution and show the differences compared with the underfitted meta-learning and overfitted classical (single-task) linear regression. Possible future directions include relaxing the assumptions and extending the result to other models related to DNNs (e.g., neural tangent kernel (NTK) models).

ACKNOWLEDGEMENT

The work of P. Ju and N. Shroff has been partly supported by the NSF grants NSF AI Institute (AI-EDGE) CNS-2112471, CNS-2106933, 2007231, CNS-1955535, and CNS-1901057, and in part by Army Research Office under Grant W911NF-21-1-0244. The work of Y. Liang has been partly supported by the NSF grants NSF AI Institute (AI-EDGE) CNS-2112471 and DMS-2134145.

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

# A    RELATED WORK

Our work is related to recent studies on characterizing the double-descent phenomenon for overfitted solutions of single-task linear regression. Some works study the min $\ell_2$-norm solutions for linear regression with simple features such as Gaussian or Fourier features (Belkin et al., 2018; 2019; Bartlett et al., 2020; Hastie et al., 2019; Muthukumar et al., 2019), where they show the existence of the double-descent phenomenon. Some others (Mitra, 2019; Ju et al., 2020) study the min $\ell_1$-norm overfitted solution and show it also has the double-descent phenomenon, but with a descent curve whose shape is very different from that of the min $\ell_2$-norm solution. Some recent works study the generalization performance when overparameterization in random feature (RF) models (Mei & Montanari, 2019), two-layer neural tangent kernel (NTK) models (Arora et al., 2019; Satpathi & Srikant, 2021; Ju et al., 2021), and three-layer NTK models (Ju et al., 2022). (RF and NTK models are linear approximations of shallow but wide neural networks.) These works show the benign overfitting for a certain set of learnable ground truth functions that depends on the network structure and which layer to train. However, all those works are on single-task learning, which cannot be directly used to characterize the generalization performance of meta-learning due to their differences in many aspects mentioned in the introduction.

Our work is related to a few recent works on the generalization performance of meta-learning. In Bernacchia (2021), the expected value of the test error for the overfitted min $\ell_2$-norm solution of MAML is provided. However, the result only works in the asymptotic regime where the number of features and the number of training samples go to infinity, which is different from ours where all quantities are finite. In Chen et al. (2022), a high probability upper bound on the test error of a similar overfitted min $\ell_2$-norm solution is given in the non-asymptotic regime. A significant difference between ours and Chen et al. (2022) is that our theoretical bound describes the shape of double descent in a straightforward manner (i.e., our theoretical bound consists of decoupled system parameters). In contrast, the bound in Chen et al. (2022) contains coupled parts (e.g., eigenvalues of weight matrices are affected by the number of tasks, samples, and features), which may not be directly used to analyze the shape of double descent. Besides, we also explain and verify the tightness of the bound, while Chen et al. (2022) does not[2]. In Huang et al. (2022), the authors focus on the generalization error during the SGD process when the training error is not zero, which is different from our focus on the overfitted solutions that make the training error equal to zero (i.e., the interpolators). Our work also differs from Bernacchia (2021); Chen et al. (2022); Huang et al. (2022) in the data generation model for the purpose of quantifying how the number of features affects the test error, which has been explained in detail at the beginning of Section 2.1.

| | Type of meta-learning | | Overparameterization | | Method | Focus of analysis |
|---|---|---|---|---|---|---|
| | Reps. | Nested | Per-task | Meta | | |
| Bai et al. (2021) | | ✓ | ✓ | | iMAML | Train-validation split |
| Chen & Chen (2022) | | ✓ | ✓ | | MAML, BMAML | Test risk |
| Saunshi et al. (2021) | | ✓ | | ✓ | MAML | Optimal step size |
| Bernacchia (2021) | ✓ | | ✓ | | - | Train-validation split |
| Sun et al. (2021) | ✓ | | | ✓ | - | Optimal representation |
| Zou et al. (2021) | | ✓ | | ✓ | MAML | Optimal step size |
| Chen et al. (2022) | | ✓ | | ✓ | MAML,iMAML | Benign overfitting |
| Ours | | ✓ | | ✓ | MAML | Descent curve shape[3] |

Table 1: Positioning our work in Table 1 of Chen et al. (2022) (which is shown here by the part above the last row).

# B    NOTATION TABLE

We summarize the important notations in Table 2.

---

[2]We use a more specific setup (such as Gaussian features) than that in Chen et al. (2022), which allows us to provide a more specific bound and verify its tightness.

[3]E.g., how the generalization performance of overfitted solutions changes with respect to the number of features.

| Symbol | Meaning | Type |
|---|---|---|
| $s$ | sparsity (the number of non-zero true parameters) | integer |
| $p$ | the number of chosen parameters/features | integer |
| $m$ | the number of training tasks | integer |
| $\boldsymbol{x}$ | the (general) input vector | vector $\mathbb{R}^p$ |
| $\boldsymbol{w}^{(i)}$ | the true parameters of the $i$-th task | vector $\mathbb{R}^p$ |
| $\epsilon$ | the noise | scalar |
| $\boldsymbol{x}_s$ | the input vector corresponding to the true parameters | vector $\mathbb{R}^s$ |
| $\boldsymbol{w}_s^{(i)}$ | the true parameters of the $i$-th task | vector $\mathbb{R}^s$ |
| $\boldsymbol{w}^{(i)}$ | the true parameters of the $i$-th task (padding with zeros) | vector $\mathbb{R}^p$ |
| $\boldsymbol{w}_s^r$ | the true parameters of the test task | vector $\mathbb{R}^s$ |
| $\boldsymbol{w}^r$ | the true parameters of the test task (padding with zeros) | vector $\mathbb{R}^p$ |
| $\boldsymbol{w}_s^0$ | mean value of $\boldsymbol{w}_s^{(i)}$ | vector $\mathbb{R}^s$ |
| $\boldsymbol{w}_0$ | mean value of $\boldsymbol{w}^{(i)}$ | vector $\mathbb{R}^p$ |
| $\hat{\boldsymbol{w}}$ | the (general) meta training result (before one-step gradient adaptation) | vector $\mathbb{R}^p$ |
| $\hat{\boldsymbol{w}}^r$ | the solution for the test task after one-step gradient adaptation on $\hat{\boldsymbol{w}}$ | vector $\mathbb{R}^p$ |
| $\hat{\boldsymbol{w}}^{(i)}$ | the solution for the $i$-th task after one-step gradient adaptation on $\hat{\boldsymbol{w}}$ | vector $\mathbb{R}^p$ |
| $\hat{\boldsymbol{w}}_{\ell_2}$ | the overfitted min $\ell_2$-norm solution | vector $\mathbb{R}^p$ |
| $\hat{\boldsymbol{w}}_{\text{ideal}}$ | the ideal overfitted solution | vector $\mathbb{R}^p$ |
| $\hat{\boldsymbol{w}}^r$ | the solution for the test task after one-step adaptation on $\hat{\boldsymbol{w}}$ | vector $\mathbb{R}^p$ |
| $n_t$ | the number of the training samples for each training task | integer |
| $n_v$ | the number of the validation samples for each training task | integer |
| $n_r$ | the number of the training samples for the test task | integer |
| $\mathbf{X}^{t(i)}$ | the matrix formed by $n_t$ training inputs of the $i$-th task | matrix $\mathbb{R}^{p \times n_t}$ |
| $\boldsymbol{y}^{t(i)}$ | the output vector corresponding to $\mathbf{X}^{t(i)}$ | vector $\mathbb{R}^{n_t}$ |
| $\boldsymbol{\epsilon}^{t(i)}$ | the noise in $\boldsymbol{y}^{t(i)}$ | vector $\mathbb{R}^{n_t}$ |
| $\mathbf{X}^{v(i)}$ | the matrix formed by $n_v$ validation inputs of the $i$-th task | matrix $\mathbb{R}^{p \times n_v}$ |
| $\boldsymbol{y}^{v(i)}$ | the output vector corresponding to $\mathbf{X}^{v(i)}$ | vector $\mathbb{R}^{n_v}$ |
| $\boldsymbol{\epsilon}^{v(i)}$ | the noise in $\boldsymbol{y}^{v(i)}$ | vector $\mathbb{R}^{n_v}$ |
| $\mathbf{X}^r$ | the matrix of $n_r$ training samples for the test task | matrix $\mathbb{R}^{p \times n_r}$ |
| $\boldsymbol{y}^r$ | the output corresponding to $\mathbf{X}^r$ | vector $\mathbb{R}^{n_r}$ |
| $\boldsymbol{\epsilon}^r$ | the noise in $\boldsymbol{y}^r$ | vector $\mathbb{R}^{n_r}$ |
| $\sigma$ | standard deviation of the noise in the training samples | scalar |
| $\sigma_r$ | standard deviation of the noise in the validation samples | scalar |
| $\nu, \nu_{(i)}$ | fluctuation of the ground truth for the training tasks | scalar |
| $\nu_r$ | fluctuation of the ground truth for the target task | scalar |
| $\mathcal{L}_{\text{inner}}^{(i)}$ | the inner loss (on the training samples of the $i$-th training task) | loss function |
| $\mathcal{L}_{\text{outer}}^{(i)}$ | the error for the validation samples after one-step gradient on the $i$-th task | loss function |
| $\mathcal{L}^{\text{meta}}$ | the meta loss (average of $\mathcal{L}_{\text{inner}}^{(i)}$) | loss function |
| $\mathcal{L}_{\text{test}}$ | the squared test error for the test task | loss function |
| $\alpha_t$ | step size of the one-step gradient on each training task | scalar |
| $\alpha_r$ | step size of the one-step gradient on the target task | scalar |

Table 2: Table of the notations.

## C  THE CALCULATION OF META LOSS

By the definition of $\mathcal{L}_{\text{inner}}^{(i)}$ in Eq. (3), we have

$$\frac{\partial \mathcal{L}_{\text{inner}}^{(i)}}{\partial \hat{\boldsymbol{w}}} = \frac{\partial \left( \boldsymbol{y}^{t(i)} - \mathbf{X}^{t(i)^T} \hat{\boldsymbol{w}} \right)}{\partial \hat{\boldsymbol{w}}} \frac{\partial \frac{1}{2} \left\| \boldsymbol{y}^{t(i)} - \mathbf{X}^{t(i)^T} \hat{\boldsymbol{w}} \right\|_2^2}{\partial \left( \boldsymbol{y}^{t(i)} - \mathbf{X}^{t(i)^T} \hat{\boldsymbol{w}} \right)}$$

$$= - \mathbf{X}^{t(i)} \left( \boldsymbol{y}^{t(i)} - \mathbf{X}^{t(i)^T} \hat{\boldsymbol{w}} \right)$$

$$= \mathbf{X}^{t(i)} \mathbf{X}^{t(i)^T} \hat{\boldsymbol{w}} - \mathbf{X}^{t(i)} \boldsymbol{y}^{t(i)}.$$

Plugging it into Eq. (3), we thus have

$$\hat{\boldsymbol{w}}^{(i)} = \left( \mathbf{I}_p - \frac{\alpha_t}{n_t} \mathbf{X}^{t(i)} \mathbf{X}^{t(i)^T} \right) \hat{\boldsymbol{w}} + \frac{\alpha_t}{n_t} \mathbf{X}^{t(i)} \boldsymbol{y}^{t(i)}. \tag{15}$$

Plugging Eq. (15) into Eq. (4), we thus have

$$\begin{aligned} \mathcal{L}_{\text{outer}}^{(i)} &= \frac{1}{2} \left\| \boldsymbol{y}^{v(i)} - \mathbf{X}^{v(i)^T} \hat{\boldsymbol{w}}^{(i)} \right\|_2^2 \\ &= \frac{1}{2} \left\| \boldsymbol{y}^{v(i)} - \mathbf{X}^{v(i)^T} \left( \left( \mathbf{I}_p - \frac{\alpha_t}{n_t} \mathbf{X}^{t(i)} \mathbf{X}^{t(i)^T} \right) \hat{\boldsymbol{w}} + \frac{\alpha_t}{n_t} \mathbf{X}^{t(i)} \boldsymbol{y}^{t(i)} \right) \right\|_2^2 \\ &= \frac{1}{2} \left\| \boldsymbol{y}^{v(i)} - \frac{\alpha_t}{n_t} \mathbf{X}^{v(i)^T} \mathbf{X}^{t(i)} \boldsymbol{y}^{t(i)} - \mathbf{X}^{v(i)^T} \left( \mathbf{I}_p - \frac{\alpha_t}{n_t} \mathbf{X}^{t(i)} \mathbf{X}^{t(i)^T} \right) \hat{\boldsymbol{w}} \right\|_2^2. \end{aligned}$$

By the definition of $\mathbf{B}$ in Eq. (7) and the definition of $\gamma$ in Eq. (7), we thus have

$$\mathcal{L}^{\text{meta}} = \frac{1}{2mn_v} = \frac{1}{2mn_v} \left\| \gamma - \mathbf{B}\hat{\boldsymbol{w}} \right\|_2^2.$$

Eq. (6) thus follows.

## D  CALCULATION OF DESCENT FLOOR

Define $h(p) := \frac{p - mn_v}{p} \left\| \boldsymbol{w}_0 \right\|_2^2 + \frac{b_\delta}{p - C_4 mn_v}$ where $p > C_4 mn_v$. We have

$$\begin{aligned} \frac{\partial h(p)}{\partial p} &= \frac{mn_v}{p^2} \left\| \boldsymbol{w}_0 \right\|_2^2 - \frac{b_\delta}{(p - C_4 mn_v)^2} \\ &= \frac{mn_v \left\| \boldsymbol{w}_0 \right\|_2^2}{(p - C_4 mn_v)^2} \left( \left( 1 - \frac{C_4 mn_v}{p} \right)^2 - \frac{b_\delta}{mn_v \left\| \boldsymbol{w}_0 \right\|_2^2} \right) \\ &= \frac{mn_v \left\| \boldsymbol{w}_0 \right\|_2^2}{(p - C_4 mn_v)^2} \left( 1 - \frac{C_4 mn_v}{p} + \sqrt{g} \right) \left( 1 - \frac{C_4 mn_v}{p} - \sqrt{g} \right) \\ &\qquad \text{(recall that } g := \frac{b_\delta}{mn_v \left\| \boldsymbol{w}_0 \right\|_2^2}). \end{aligned}$$

Notice that when $p > C_4 mn_v$ and $\boldsymbol{w}_0 \neq \mathbf{0}$, the first and the second factors are positive. Thus, we only need to consider the sign of $A := \left( 1 - \frac{C_4 mn_v}{p} - \sqrt{g} \right)$. If $g \geq 1$, then $A < 0$, which implies that $h(p)$ is monotone decreasing. If $g < 1$, then we have

$$A \begin{cases} < 0, & \text{when } p \in \left( C_4 mn_v, \frac{C_4 mn_v}{1 - \sqrt{g}} \right), \\ > 0, & \text{when } p > \frac{C_4 mn_v}{1 - \sqrt{g}}, \end{cases}$$

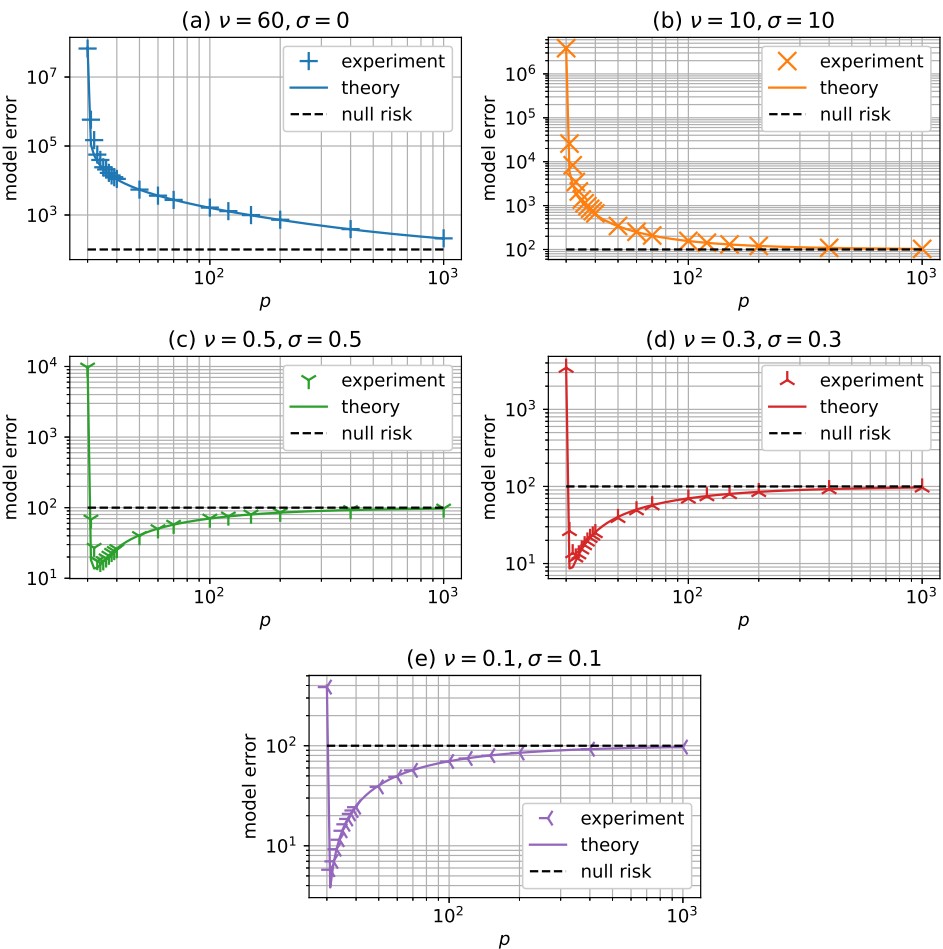

Figure 2: Comparison between the experimental values and the theoretical values of the model error.

which implies that $h(p)$ is monotone decreasing when $p \in \left( C_4 m n_v, \frac{C_4 m n_v}{1-\sqrt{g}} \right)$, and is monotone increasing when $p > \frac{C_4 m n_v}{1-\sqrt{g}}$. This suggests a descent floor at $p = \frac{C_4 m n_v}{1-\sqrt{g}}$, where at this point

$$
\begin{aligned}
\left. h(p) \right|_{p=\frac{C_4 m n_v}{1-\sqrt{g}}} &= \|\boldsymbol{w}_0\|_2^2 \left( 1 - \frac{1-\sqrt{g}}{C_4} + \frac{g m n_v}{C_4 m n_v \left( \frac{1}{1-\sqrt{g}} - 1 \right)} \right) \\
&= \|\boldsymbol{w}_0\|_2^2 \left( 1 - \frac{1-\sqrt{g}}{C_4} + \frac{g}{C_4 \left( \frac{1}{1-\sqrt{g}} - 1 \right)} \right) \\
&= \|\boldsymbol{w}_0\|_2^2 \left( 1 - \frac{(1-\sqrt{g})^2}{C_4} \right).
\end{aligned}
$$

# E ADDITIONAL EXPERIMENTS

## E.1 SIMULATIONS TO VERIFY THE TIGHTNESS OF THEOREM 1

In Fig. 2, we plot both the experimental values (denoted by separated markers) and the theoretical values (denoted by the continuous curve) of the model error. The simulation setup and (consequently) the experimental values are the same as those in Fig. 1 (note that we only show the points in the overparameterized region, i.e., $p \geq m n_v$). The theoretical value is calculated by Theorem 1

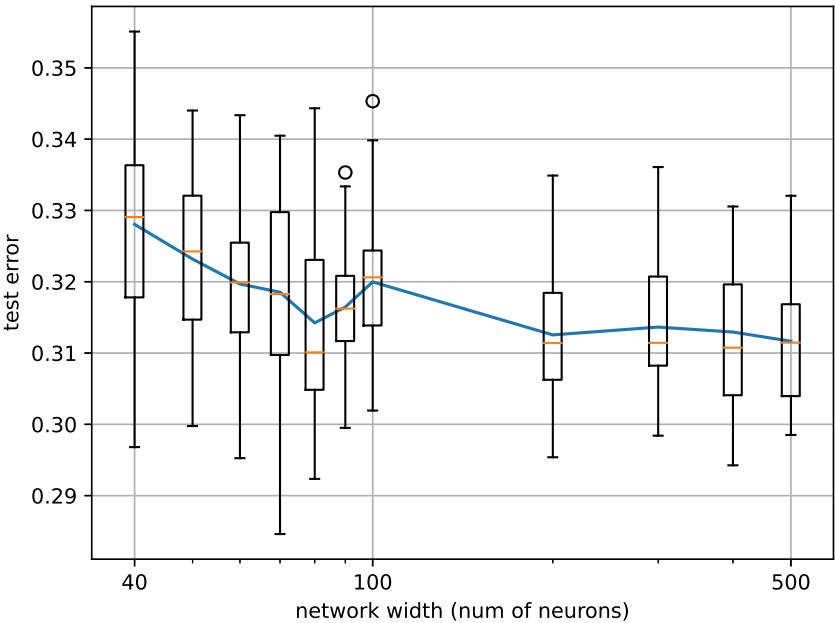

Figure 3: The test error for a two-layer fully connected neural network. The x-axis denotes the neural network width (i.e., the number of neurons in the hidden layer).

and the approximation Eq. (9) with constants[4] $C_1 = C_3 = 0.001$ and $C_2 = C_4 = 0.99995$. As we can see in Fig. 2, the experimental value points closely match the theoretical curves, which suggests that Theorem 1 and Eq. (9) are fairly tight.

### E.2 Experiments of meta-learning with Neural Network on Real-World Data

In this section, we further verify our theoretical findings by an experiment over a two-layer fully-connected neural network on the MNIST data set.

**Neural network structure:** The input dimension of the neural network is 49 (i.e., $7 \times 7$ gray-scale image shrunk from the original $28 \times 28$ gray-scale image). The input is multiplied by the first-layer weights and fully connected to the hidden layer that consists of ReLUs (rectified linear units $\max\{0, \cdot\}$) with bias. The output of these ReLUs is then multiplied by the second layer weights and then goes to the output layer. The output layer is a sigmoid activation function with bias.

**Experimental setup:** There are 4 training tasks and 1 test task. The objective of each task is to identify whether the input image belongs to a set of 5 different digits. Specifically, the sets for 4 training tasks are $\{1, 2, 3, 7, 9\}$, $\{0, 2, 3, 6, 9\}$, $\{0, 1, 4, 6, 8\}$, $\{1, 2, 3, 5, 7\}$, respectively. The set for the test task is $\{0, 2, 3, 6, 8\}$. For each training task, there are 1000 training samples and 100 validation samples. All samples (except the ones to calculate the test error) are corrupted by *i.i.d.* Gaussian noise in each pixel with zero mean and standard deviation 0.3. The number of samples for the one-step gradient is 1000 for these 4 training tasks and is 500 for the test task, i.e., $n_t = 1000$ and $n_r = 500$. The number of validation samples is $n_v = 100$ for each of these 4 training tasks. The initial weights are uniformly randomly chosen in the range $[0, 1]$.

**Training process:** We use gradient descent to train the neural network. The step size in the outer-loop training is 0.3 and the step size of the one-step gradient adaptation is $\alpha_t = \alpha_r = 0.05$. After training 500 epochs, the meta-training error for each simulation is lower than 0.025 (the range of the meta-training error is $[0, 1]$), which means that the trained model almost completely fits all validation samples.

---

[4]These constants are manually calibrated to fit the experimental values better.

**Simulation results and interpretation:** We run the simulation 30 times with different random seeds. In Fig. 3, we draw a box plot showing the test error for the test task, where the blue curve denotes the average value of these 30 runs. We can see that the overall trend of the test curve is decreasing, which suggests that more parameters help to enhance the overfitted generalization performance. Another interesting phenomenon we find from Fig. 3 is that the curve is not strictly monotonically decreasing and there exist some decent floors (e.g., when the network width is 80).

## F   DETAILS OF DERIVING EQ. (12)

Notice that

$$
\begin{aligned}
\left\langle \left(\mathbf{I}_p - \mathbf{B}^T(\mathbf{B}\mathbf{B}^T)^{-1}\mathbf{B}\right)\boldsymbol{w}_0,\ \mathbf{B}^T(\mathbf{B}\mathbf{B}^T)^{-1}\delta\gamma \right\rangle =&\ (\delta\gamma)^T(\mathbf{B}\mathbf{B}^T)^{-1}\mathbf{B}\left(\mathbf{I}_p - \mathbf{B}^T(\mathbf{B}\mathbf{B}^T)^{-1}\mathbf{B}\right)\boldsymbol{w}_0 \\
=&\ (\delta\gamma)^T\left((\mathbf{B}\mathbf{B}^T)^{-1}\mathbf{B} - (\mathbf{B}\mathbf{B}^T)^{-1}\mathbf{B}\right)\boldsymbol{w}_0 \\
=&\ 0.
\end{aligned}
\tag{16}
$$

We thus have

$$
\begin{aligned}
\|\boldsymbol{w}_0 - \hat{\boldsymbol{w}}_{\ell_2}\|_2^2 &= \left\|\boldsymbol{w}_0 - \mathbf{B}^T(\mathbf{B}\mathbf{B}^T)^{-1}\gamma\right\|_2^2 \text{ (by Eq. (10))} \\
&= \left\|\left(\mathbf{I}_p - \mathbf{B}^T(\mathbf{B}\mathbf{B}^T)^{-1}\mathbf{B}\right)\boldsymbol{w}_0 - \mathbf{B}^T(\mathbf{B}\mathbf{B}^T)^{-1}\delta\gamma\right\|_2^2 \text{ (by Eq. (11))} \\
&= \underbrace{\left\|\left(\mathbf{I}_p - \mathbf{B}^T(\mathbf{B}\mathbf{B}^T)^{-1}\mathbf{B}\right)\boldsymbol{w}_0\right\|_2^2}_{\text{Term 1}} + \underbrace{\left\|\mathbf{B}^T(\mathbf{B}\mathbf{B}^T)^{-1}\delta\gamma\right\|_2^2}_{\text{Term 2}} \text{ (by Eq. (16))}.
\end{aligned}
$$

## G   SOME USEFUL LEMMAS

### G.1   ESTIMATION ON LOGARITHM

**Lemma 2.** *For any $x > 0$, we must have $\ln x \geq 1 - \frac{1}{x}$.*

*Proof.* This can be derived by examining the monotonicity of $\ln x - (1 - \frac{1}{x})$. The complete proof can be found, e.g., in Lemma 33 of Ju et al. (2020). □

**Lemma 3.** *When $k \geq 16$, we must have $2\sqrt{\frac{\ln k}{k}} < 1$.*

*Proof.* We only need to prove that the function $g(k) := 4\ln k - k \leq 0$ when $k \geq 16$. To that end, when $k \geq 16$, we have

$$
\frac{\partial g(k)}{\partial k} = \frac{4}{k} - 1 \leq 0.
$$

Thus, $g(k)$ is monotone decreasing when $k \geq 16$. Also notice that $g(16) = 4\ln 16 - 16 \approx -4.91 < 0$. The result of this lemma thus follows. □

### G.2   ESTIMATION ON NORM AND EIGENVALUES

Let $\lambda_{\min}(\cdot)$ and $\lambda_{\min}(\cdot)$ denote the minimum and maximum singular value of a matrix.

**Lemma 4.** *Consider an orthonormal basis matrix $\mathbf{U} \in \mathbb{R}^{k \times k}$ and any vector $\boldsymbol{a} \in \mathbb{R}^{k \times 1}$. Then we must have*

$$
\|\mathbf{U}\boldsymbol{a}\|_2 = \|\boldsymbol{a}\|_2.
$$

*Proof.* We have

$$
\mathbf{U}\boldsymbol{a} = \sum_{i=1}^k \boldsymbol{a}_i\mathbf{U}_i,
\tag{17}
$$

where $\mathbf{U}_i$ is the $i$-th column of $\mathbf{U}$ and $\boldsymbol{a}_i$ is the $i$-th element of $\boldsymbol{a}$. Because $\mathbf{U}$ is an orthonormal basis matrix, we know that $\mathbf{U}_i$ are orthogonal to each other. In other words,

$$\mathbf{U}_i^T \mathbf{U}_j = \begin{cases} 0, & \text{if } i \neq j, \\ 1, & \text{if } i = j. \end{cases} \tag{18}$$

Thus, we have

$$
\begin{aligned}
&\|\mathbf{U}\boldsymbol{a}\|_2 \\
&= \sqrt{\left(\sum_{i=1}^k \boldsymbol{a}_i \mathbf{U}_i\right)^T \left(\sum_{i=1}^k \boldsymbol{a}_i \mathbf{U}_i\right)} \text{ (by Eq. (17))} \\
&= \sqrt{\sum_{i=1}^k \boldsymbol{a}_i^2} \text{ (by Eq. (18))} \\
&= \|\boldsymbol{a}\|_2 .
\end{aligned}
$$

The result of this lemma thus follows. $\qquad \square$

**Lemma 5.** *Consider a diagonal matrix $\mathbf{D} = \mathsf{diag}(d_1, d_2, \cdots, d_k) \in \mathbb{R}^{k \times k}$ where $\min_i d_i \geq 0$. For any vector $\boldsymbol{a}$, we must have*

$$\|\mathbf{D}\boldsymbol{a}\|_2 \in \left[\min_i d_i \|\boldsymbol{a}\|_2, \max_i d_i \|\boldsymbol{a}\|_2\right].$$

*Proof.* We have

$$\|\mathbf{D}\boldsymbol{a}\|_2 = \sqrt{\sum_{i=1}^k d_i^2 \boldsymbol{a}_i^2} \in \left[\min_i d_i \sqrt{\sum_{j=1}^k \boldsymbol{a}_j^2}, \ \max_i d_i \sqrt{\sum_{j=1}^k \boldsymbol{a}_j^2}\right] \in \left[\min_i d_i \|\boldsymbol{a}\|_2, \max_i d_i \|\boldsymbol{a}\|_2\right].$$

The result of this lemma thus follows. $\qquad \square$

**Lemma 6.** *For any $\mathbf{A} \in \mathbb{R}^{q \times k}$ and $\boldsymbol{a} \in \mathbb{R}^{k \times 1}$, we must have*

$$\|\mathbf{A}\boldsymbol{a}\|_2^2 \in \left[\lambda_{\min}(\mathbf{A}^T \mathbf{A}) \|\boldsymbol{a}\|_2^2, \ \lambda_{\max}(\mathbf{A}^T \mathbf{A}) \|\boldsymbol{a}\|_2^2\right].$$

*Proof.* Do singular value decomposition of $\mathbf{A}$ as $\mathbf{A} = \mathbf{U}\mathbf{D}\mathbf{V}^T$. Here $\mathbf{D} \in \mathbb{R}^{q \times k}$ is a diagonal matrix that consists of all singular values, $\mathbf{U} \in \mathbb{R}^{q \times q}$ and $\mathbf{V} \in \mathbb{R}^{k \times k}$ (and their transpose) are orthonormal basis matrices. We have

$$
\begin{aligned}
\|\mathbf{A}\boldsymbol{a}\|_2^2 = \boldsymbol{a}^T \mathbf{A}^T \mathbf{A}\boldsymbol{a} &= \boldsymbol{a}^T \mathbf{V}\mathbf{D}^T\mathbf{D}\mathbf{V}^T \boldsymbol{a} \\
&= \left\|\sqrt{\mathbf{D}^T\mathbf{D}}\mathbf{V}^T \boldsymbol{a}\right\|_2^2 \\
&\in \left[\lambda_{\min}(\mathbf{D}^T\mathbf{D}) \left\|\mathbf{V}^T\boldsymbol{a}\right\|_2^2, \ \lambda_{\max}(\mathbf{D}^T\mathbf{D}) \left\|\mathbf{V}^T\boldsymbol{a}\right\|_2^2\right] \text{ (by Lemma 5)} \\
&= \left[\lambda_{\min}(\mathbf{A}^T\mathbf{A}) \left\|\mathbf{V}^T\boldsymbol{a}\right\|_2^2, \ \lambda_{\max}(\mathbf{A}^T\mathbf{A}) \left\|\mathbf{V}^T\boldsymbol{a}\right\|_2^2\right] \text{ (by } \mathbf{A} = \mathbf{U}\mathbf{D}\mathbf{V}^T) \\
&= \left[\lambda_{\min}(\mathbf{A}^T\mathbf{A}) \|\boldsymbol{a}\|_2^2, \ \lambda_{\max}(\mathbf{A}^T\mathbf{A}) \|\boldsymbol{a}\|_2^2\right] \text{ (by Lemma 4)}.
\end{aligned}
$$

The result of this lemma thus follows. $\qquad \square$

The following lemma is useful to estimate the eigenvalues of a matrix whose off-diagonal elements are relatively small.

**Lemma 7** (Gershgorin's Circle Theorem (Marquis et al., 2016)). *If $A$ is an $n \times n$ matrix with complex entries $a_{i,j}$ then $r_i(A) = \sum_{j \neq i} |a_{i,j}|$ is defined as the sum of the magnitudes of the non-diagonal entries of the $i$-th row. Then a Gershgorin disc is the disc $D(a_{i,i}, r_i(A))$ centered at $a_{i,i}$ on the complex plane with radius $r_i(A)$. Theorem: Every eigenvalue of a matrix lies within at least one Gershgorin disc.*

### G.3 ESTIMATION ON RANDOM VARIABLES OF CERTAIN DISTRIBUTIONS

**Lemma 8** (Corollary 5.35 of Vershynin (2010))**.** *Let* $\mathbf{A}$ *be an* $N_1 \times N_2$ *matrix* $(N_1 \times N_2)$ *whose entries are independent standard normal random variables. Then for every* $t \geq 0$*, with probability at least* $1 - 2\exp(-t^2/2)$ *one has*

$$\sqrt{N_1} - \sqrt{N_2} - t \leq \lambda_{\min}(\mathbf{A}) \leq \lambda_{\max}(\mathbf{A}) \leq \sqrt{N_1} + \sqrt{N_2} + t.$$

**Lemma 9** (stated on pp. 1325 of Laurent & Massart (2000))**.** *Let* $U$ *follow* $\chi^2$ *distribution with D degrees of freedom. For any positive x, we have*

$$\Pr\left\{U - D \geq 2\sqrt{Dx} + 2x\right\} \leq e^{-x},$$
$$\Pr\left\{D - U \geq 2\sqrt{Dx}\right\} \leq e^{-x}.$$

**Lemma 10** (Lemma 31 of Ju et al. (2020))**.** *Let* $u_1, u_2, \cdots, u_k$ *and* $u_1, u_2, \cdots, u_k$ *denote* $2k$ *random variables that follow* i.i.d. *standard normal distribution. For any* $a > 0$*, we must have*

$$\Pr\left\{\left|\sum_{i=1}^{k} u_i v_i\right| > \frac{ka}{2}\right\} \leq 2\exp\left(-\frac{k}{2}\left(at + \ln\frac{2t}{a}\right)\right),$$

*where*

$$t = \frac{-1 + \sqrt{1 + a^2}}{a}.$$

**Lemma 11.** *For any* $k \geq 16$*, let* $u_1, u_2, \cdots, u_k$ *and* $u_1, u_2, \cdots, u_k$ *denote* $2k$ *random variables that follows* i.i.d. *standard normal distribution. For any* $q \leq k$ *and* $c \geq 0$*, we must have*

$$\Pr\left\{\left|\sum_{i=1}^{k} u_i v_i\right| > c \cdot \sqrt{k \ln q}\right\} \leq \frac{2}{e^{c \cdot 0.4 \ln q}}.$$

*Further, by letting* $c = 1$ *and* $q = k \geq 16$*, we have*

$$\Pr\left\{\left|\sum_{i=1}^{k} u_i v_i\right| > \sqrt{k \ln k}\right\} \leq \frac{2}{k^{0.4}}.$$

*By letting* $q = e$*, we have*

$$\Pr\left\{\left|\sum_{i=1}^{k} u_i v_i\right| > c\sqrt{k}\right\} \leq \frac{2}{e^{0.4c}}.$$

*Proof.* Recall the definition of $t$ in Lemma 10, we first want to prove $at + \ln\frac{2t}{a} \geq \frac{a^2}{2(\sqrt{1+a^2}+1)}$. To that end, we have

$$
\begin{aligned}
at + \ln\frac{2t}{a} &= -1 + \sqrt{1 + a^2} + \ln\frac{2(-1 + \sqrt{1 + a^2})}{a^2} \quad \text{(by the definition of } t \text{ in Lemma 10)} \\
&\geq -1 + \sqrt{1 + a^2} + 1 - \frac{a^2}{2(-1 + \sqrt{1 + a^2})} \quad \text{(by Lemma 2)} \\
&= \frac{a^2}{1 + \sqrt{1 + a^2}} + 1 - \frac{1 + \sqrt{1 + a^2}}{2} \quad \text{(since } a^2 = (-1 + \sqrt{1 + a^2})(1 + \sqrt{1 + a^2})) \\
&= \frac{a^2}{2(\sqrt{1 + a^2} + 1)}.
\end{aligned}
\tag{19}
$$

Now we let $a = 2c \cdot \sqrt{\frac{\ln q}{k}}$. Thus, we have

$$\frac{ka}{2} = c \cdot \sqrt{k \ln q}. \tag{20}$$

We have

$$
\begin{aligned}
a =& 2c \cdot \sqrt{\frac{\ln q}{k}} \\
\leq& 2c \cdot \sqrt{\frac{\ln k}{k}} \text{ (since } q \leq k) \\
\leq& c \text{ (by Lemma 3).}
\end{aligned}
\tag{21}
$$

Because $c \geq 1$, we have

$$
1 + \sqrt{1 + c^2} \leq c + \sqrt{c^2 + c^2} = c(1 + \sqrt{2}).
\tag{22}
$$

We thus have

$$
\begin{aligned}
at + \ln \frac{2t}{a} \geq& \frac{a^2}{2(\sqrt{1 + a^2} + 1)} \text{ (by Eq. (19))} \\
\geq& \frac{a^2}{2(1 + \sqrt{1 + c^2})} \text{ (by Eq. (21))} \\
\geq& \frac{a^2}{2c(1 + \sqrt{2})} \text{ (by Eq. (22))} \\
\geq& \frac{4c}{5} \frac{\ln q}{k} \text{ (since } a = 2c \cdot \sqrt{\frac{\ln q}{k}} \text{ and } \sqrt{2} \approx 1.414 \leq \frac{3}{2}).
\end{aligned}
\tag{23}
$$

Thus, we have

$$
\begin{aligned}
& 2 \exp \left( -\frac{k}{2} \left( at + \ln \frac{2t}{a} \right) \right) \\
\leq& 2 \exp \left( -\frac{k}{2} \cdot \frac{4c}{5} \frac{\ln q}{k} \right) \text{ (by Eq. (23))} \\
=& 2 \exp \left( -0.4c \ln q \right).
\end{aligned}
\tag{24}
$$

Plugging Eq. (20) and Eq. (24) into Lemma 10, the result of this lemma thus follows. $\qquad\square$

**Lemma 12** (Isserlis' theorem (Michalowicz et al., 2009)). *If* $(x_1, x_2, \cdots, x_n)$ *is a zero-mean multivariate normal random vector, then*

$$
\mathbb{E}[x_1 x_2 \cdots x_n] = \sum_{A \in \mathcal{A}_n^2} \prod_{(i,j) \in A} \mathbb{E}[x_i x_j],
$$

*where* $A$ *denotes a partition of* $1, 2, \cdots, n$ *into pairs, and* $\mathcal{A}_n^2$ *denotes all such partitions. For example,*

$$
\mathbb{E}[x_1 x_2 x_3 x_4] = \mathbb{E}[x_1 x_2] \mathbb{E}[x_3 x_4] + \mathbb{E}[x_1 x_3] \mathbb{E}[x_2 x_4] + \mathbb{E}[x_1 x_4] \mathbb{E}[x_2 x_3].
$$

The following lemma is mentioned in Bernacchia (2021) without a detailed proof. For the ease of readers, we provide a detailed proof of this lemma here.

**Lemma 13.** *Consider a random matrix* $X \in \mathbb{R}^{p \times n}$ *whose each element follows* i.i.d. *standard Gaussian distribution (i.e.,* i.i.d. $\mathcal{N}(0, 1)$*). We mush have*

$$
\begin{aligned}
\mathbb{E}[X^T X] &= p \mathbf{I}_n, \\
\mathbb{E}[X X^T] &= n \mathbf{I}_p, \\
\mathbb{E}[X X^T X X^T] &= n(n + p + 1) \mathbf{I}_p.
\end{aligned}
$$

*Proof.* Since each row of $X$ are *i.i.d.*, we immediately have $\mathbb{E}[X X^T] = n \mathbf{I}_p$ and $\mathbb{E}[X^T X] = p \mathbf{I}_n$. It remains to prove $\mathbb{E}[X X^T X X^T] = n(n + p + 1) \mathbf{I}_p$. To that end, we have

$$
\left[ X X^T \right]_{i,j} = X_i X_j^T = \sum_{k=1}^n X_{i,k} X_{j,k},
\tag{25}
$$

where $X_i$ denotes the $i$-th row of $X$, and $[\cdot]_{i,j}$ denotes the element in the $i$-th row, $j$-th column of the matrix. Thus, we have

$$
\begin{aligned}
\left[XX^TXX^T\right]_{i,j} &= \sum_{k=1}^{p} \left[XX^T\right]_{i,k} \left[XX^T\right]_{k,j} \\
&= \sum_{k=1}^{p} \left(\sum_{l=1}^{n} X_{i,l}X_{k,l}\right) \left(\sum_{l'=1}^{n} X_{j,l'}X_{k,l'}\right) \quad \text{(by Eq. (25))}.
\end{aligned}
\tag{26}
$$

Now we examine the value of $\mathbb{E}\, X_{i,l}X_{k,l}X_{j,l'}X_{k,l'}$ by Isserlis' theorem (Lemma 12).

$$
\begin{aligned}
&\mathbb{E}\left[X_{i,l}X_{k,l}X_{j,l'}X_{k,l'}\right] \\
&= \begin{cases}
0, & \text{when } i \neq j, \\
0, & \text{when } i = j \text{ and } k \neq i \text{ and } l \neq l', \\
1, & \text{when } i = j \text{ and } k \neq i \text{ and } l = l' \text{ (there are } n(p-1) \text{ such terms for each } i), \\
1, & \text{when } i = j = k \text{ and } l \neq l' \text{ (there are } n(n-1) \text{ such terms for each } i), \\
3, & \text{when } i = j = k \text{ and } l = l' \text{ (there are } n \text{ such terms for each } i).
\end{cases}
\end{aligned}
$$

By Eq. (26), we thus have

$$
\mathbb{E}[XX^TXX^T] = (n(p-1) + n(n-1) + 3n)\mathbf{I}_p = n(n+p+1)\mathbf{I}_p.
$$

The result of this lemma thus follows. $\qquad\square$

**Lemma 14** (Lemma 24 of Ju et al. (2020)). *Considering a standard Gaussian distribution $a \sim \mathcal{N}(0,1)$, when $t \geq 0$, we have*

$$
\frac{\sqrt{2/\pi}\, e^{-t^2/2}}{t^2 + \sqrt{t^2+4}} \leq \Pr\{a \geq t\} \leq \frac{\sqrt{2/\pi}\, e^{-t^2/2}}{t + \sqrt{t^2 + \frac{8}{\pi}}}.
$$

*Notice that $t + \sqrt{t^2 + \frac{8}{\pi}} \geq 2\sqrt{\frac{2}{\pi}}$ when $t \geq 0$. We thus have*

$$
\Pr\{a \geq t\} \leq \frac{1}{2}e^{-t^2/2}.
$$

*By the symmetry of the standard Gaussian distribution, we thus have*

$$
\Pr\{|a| \geq t\} \leq e^{-t^2/2}.
$$

## H   PROOF OF LEMMA 1

*Proof.* Similar to Eq. (15), we have

$$
\hat{\boldsymbol{w}}^r = \left(\mathbf{I}_p - \frac{\alpha_r}{n_r}\mathbf{X}^r\mathbf{X}^{rT}\right)\hat{\boldsymbol{w}} + \frac{\alpha_r}{n_r}\mathbf{X}^r\boldsymbol{y}^r.
\tag{27}
$$

By Eq. (27), we can express the learned result for the test task as

$$
\begin{aligned}
\hat{\boldsymbol{w}}^r &= \left(\mathbf{I}_p - \frac{\alpha_r}{n_r}\mathbf{X}^r\mathbf{X}^{rT}\right)\hat{\boldsymbol{w}} + \frac{\alpha_r}{n_r}\mathbf{X}^r\boldsymbol{y}^r \\
&= \left(\mathbf{I}_p - \frac{\alpha_r}{n_r}\mathbf{X}^r\mathbf{X}^{rT}\right)\hat{\boldsymbol{w}} + \frac{\alpha_r}{n_r}\mathbf{X}^r\left(\mathbf{X}^{rT}\boldsymbol{w}^r + \boldsymbol{\epsilon}^r\right) \quad \text{(by Eq. (2))}.
\end{aligned}
\tag{28}
$$

Considering a new input $\boldsymbol{x}$ for the test task, the ground-truth is $\boldsymbol{x}^T\boldsymbol{w}^r$. The distance between the ground-truth and the output of the learned model is

$$
\left\|\boldsymbol{x}^T\boldsymbol{w}^r - \boldsymbol{x}^T\hat{\boldsymbol{w}}^r\right\|_2.
$$

Taking the expectation on $\boldsymbol{x}$ for the square of the distance, we have

$$
\begin{aligned}
\mathbb{E}_{\boldsymbol{x}} \left\| \boldsymbol{x}^T \boldsymbol{w}^r - \boldsymbol{x}^T \hat{\boldsymbol{w}}^r \right\|_2^2 &= \mathbb{E}_{\boldsymbol{x}} \left\| \boldsymbol{x}^T (\boldsymbol{w}^r - \hat{\boldsymbol{w}}^r) \right\|_2^2 \\
&= \mathbb{E}_{\boldsymbol{x}} (\boldsymbol{w}^r - \hat{\boldsymbol{w}}^r)^T \boldsymbol{x} \boldsymbol{x}^T (\boldsymbol{w}^r - \hat{\boldsymbol{w}}^r) \\
&= (\boldsymbol{w}^r - \hat{\boldsymbol{w}}^r)^T \mathbb{E}_{\boldsymbol{x}} [\boldsymbol{x} \boldsymbol{x}^T] (\boldsymbol{w}^r - \hat{\boldsymbol{w}}^r) \\
&= \left\| \boldsymbol{w}^r - \hat{\boldsymbol{w}}^r \right\|_2^2 \text{ (since } \mathbb{E}_{\boldsymbol{x}} [\boldsymbol{x} \boldsymbol{x}^T] = \mathbf{I}_p \text{ by Assumption 1).}
\end{aligned}
$$

Notice that

$$
\begin{aligned}
\hat{\boldsymbol{w}}^r - \boldsymbol{w}^r &= \left( \mathbf{I}_p - \frac{\alpha_r}{n_r} \mathbf{X}^r \mathbf{X}^{rT} \right) \hat{\boldsymbol{w}} + \left( \frac{\alpha_r}{n_r} \mathbf{X}^r \mathbf{X}^{rT} - \mathbf{I}_p \right) \hat{\boldsymbol{w}}^r + \frac{\alpha_r}{n_r} \mathbf{X}^r \boldsymbol{\epsilon}^r \text{ (by Eq. (28))} \\
&= \left( \mathbf{I}_p - \frac{\alpha_r}{n_r} \mathbf{X}^r \mathbf{X}^{rT} \right) (\hat{\boldsymbol{w}} - \boldsymbol{w}^r) + \frac{\alpha_r}{n_r} \mathbf{X}^r \boldsymbol{\epsilon}^r
\end{aligned}
$$

Then, taking the expectation on $\boldsymbol{\epsilon}^r$, we have

$$
\begin{aligned}
\mathbb{E}_{\mathbf{X}^r, \boldsymbol{\epsilon}^r} \left\| \boldsymbol{w}^r - \hat{\boldsymbol{w}}^r \right\|_2^2 &= \mathbb{E}_{\mathbf{X}^r, \boldsymbol{\epsilon}^r} \left\| \left( \mathbf{I}_p - \frac{\alpha_r}{n_r} \mathbf{X}^r \mathbf{X}^{rT} \right) (\hat{\boldsymbol{w}} - \boldsymbol{w}^r) \right\|_2^2 + \mathbb{E}_{\mathbf{X}^r, \boldsymbol{\epsilon}^r} \left\| \frac{\alpha_r}{n_r} \mathbf{X}^r \boldsymbol{\epsilon}^r \right\|_2^2 \\
&\quad \text{(since } \boldsymbol{\epsilon}^r \text{ is independent of other random variables)} \\
&= (\hat{\boldsymbol{w}} - \boldsymbol{w}^r)^T \mathbb{E}_{\mathbf{X}^r} \left[ \left( \mathbf{I}_p - \frac{\alpha_r}{n_r} \mathbf{X}^r \mathbf{X}^{rT} \right)^T \left( \mathbf{I}_p - \frac{\alpha_r}{n_r} \mathbf{X}^r \mathbf{X}^{rT} \right) \right] (\hat{\boldsymbol{w}} - \boldsymbol{w}^r) \\
&\quad + \frac{\alpha_r^2}{n_r^2} \mathbb{E}_{\boldsymbol{\epsilon}^r} \left[ \boldsymbol{\epsilon}^{rT} \mathbb{E}_{\mathbf{X}^r} [\mathbf{X}^{rT} \mathbf{X}^r] \boldsymbol{\epsilon}^r \right] \\
&= \left( 1 - 2\alpha_r + \frac{\alpha_r^2}{n_r} (n_r + p + 1) \right) \left\| \hat{\boldsymbol{w}} - \boldsymbol{w}^r \right\|_2^2 + \frac{\alpha_r^2 p}{n_r} \sigma_r^2 \text{ (by Lemma 13).}
\end{aligned}
$$

(29)

Notice that

$$
\begin{aligned}
\mathbb{E}_{\boldsymbol{w}^r} \left\| \hat{\boldsymbol{w}} - \boldsymbol{w}^r \right\|_2^2 &= \mathbb{E}_{\boldsymbol{w}^r} \left\| \hat{\boldsymbol{w}} - \boldsymbol{w}_0 - (\boldsymbol{w}^r - \boldsymbol{w}_0) \right\|_2^2 \\
&= \left\| \hat{\boldsymbol{w}} - \boldsymbol{w}_0 \right\|_2^2 + \nu_r^2 \text{ (by Assumption 2, } (\boldsymbol{w}^r - \boldsymbol{w}_0) \text{ has zero mean).}
\end{aligned}
$$

(30)

The result of this proposition thus follows by plugging Eq. (30) into Eq. (29). $\qquad \square$

## H.1 PROOF OF PROPOSITION 1

*Proof.* For ease of notation, we let $K := \left\| \hat{\boldsymbol{w}} - \boldsymbol{w}_0 \right\|_2^2 + \nu_r^2$. From Lemma 1, we can see that $f_{\text{test}} \left( \left\| \hat{\boldsymbol{w}} - \boldsymbol{w}_0 \right\|_2^2 \right)$ is a quadratic function of $\alpha_r$:

$$
f_{\text{test}} \left( \left\| \hat{\boldsymbol{w}} - \boldsymbol{w}_0 \right\|_2^2 \right) = \left( \left( \frac{p+1}{n_r} + 1 \right) K + \frac{p \sigma_r^2}{n_r} \right) \alpha_r^2 - 2K \alpha_r + K.
$$

Thus, to minimize the test loss shown by Lemma 1, we can calculate the optimal choice of $\alpha_r$ as

$$
[\alpha_r]_{\text{opti}} = \frac{K}{\left( 1 + \frac{p+1}{n_r} \right) K + \frac{p \sigma_r^2}{n_r}}.
$$

(Thus, we have $[\alpha_r]_{\text{opti}} = \frac{n_r}{n_r + p + 1}$ when $\sigma_r = 0$.)

Plugging $[\alpha_r]_{\text{opti}}$ into $f_{\text{test}} \left( \left\| \hat{\boldsymbol{w}} - \boldsymbol{w}_0 \right\|_2^2 \right)$, we thus have

$$
f_{\text{test}} \left( \left\| \hat{\boldsymbol{w}} - \boldsymbol{w}_0 \right\|_2^2 \right) \bigg|_{\alpha_r = [\alpha_r]_{\text{opti}}} = K - \frac{K^2}{\left( 1 + \frac{p+1}{n_r} \right) K + \frac{p \sigma_r^2}{n_r}} = K \cdot \frac{p + 1 + \frac{p \sigma_r^2}{K}}{n_r + p + 1 + \frac{p \sigma_r^2}{K}}.
$$

(31)

The right-hand side of Eq. (31) increases when $\sigma_r^2$ increases. Therefore, by letting $\sigma_r = 0$ and $\sigma_r = \infty$, we can get the lower and upper bound of Eq. (31), i.e.,

$$f_{\text{test}}\left(\|\hat{\boldsymbol{w}} - \boldsymbol{w}_0\|_2^2\right)\bigg|_{\alpha_r = [\alpha_r]_{\text{opti}}} \in \left[K \cdot \frac{p+1}{n_r + p + 1}, \; K\right].$$

The result of this lemma thus follows.

$\square$

## I   PROOF OF THEOREM 1

We summarize the definition of the quantities related to the definition of $b_w$ as follows:

$$\alpha_t' := \frac{\alpha_t}{n_t}\left(\sqrt{p} + \sqrt{n_t} + \ln\sqrt{n_t}\right)^2,$$

$$b_{\text{eig,min}} := p + \left(\max\{0, 1 - \alpha_t'\}^2 - 1\right)n_t - \left((n_v + 1)\max\{\alpha_t', 1 - \alpha_t'\}^2 + 6mn_v\right)\sqrt{p}\ln p,$$

$$c_{\text{eig,min}} := \max\{0, 1 - \alpha_t'\}^2 p - 2mn_v\max\{\alpha_t', 1 - \alpha_t'\}^2\sqrt{p\ln p},$$

$$D := \left(\max\left\{\left|1 - \alpha_t\frac{n_t + 2\sqrt{n_t\ln(sn_t)} + 2\ln(sn_t)}{n_t}\right|, \; \left|1 - \alpha_t\frac{n_t - 2\sqrt{n_t\ln(sn_t)}}{n_t}\right|\right\}\right)^2,$$

$$b_\delta := mn_v\sigma^2\left(1 + \frac{\alpha_t^2 p(\ln n_t)^2\ln p}{n_t}\right) + mn_v\nu^2 \cdot 2\ln(sn_t)\cdot\left(D + \frac{\alpha_t^2(p-1)}{n_t}6.25(\ln(spn_t))^2\right),$$

$$b_{\boldsymbol{w}_0} := \frac{(p - mn_v) + 2\sqrt{(p - mn_v)\ln p} + 2\ln p}{p - 2\sqrt{p\ln p}}\|\boldsymbol{w}_0\|_2^2,$$

$$b_w^{\text{ideal}} := \frac{b_\delta}{\max\{b_{\text{eig,min}}\mathbb{1}_{\{p > n_t\}} + c_{\text{eig,min}}\mathbb{1}_{\{p \leq n_t\}}, \; 0\}},$$

$$b_w := b_{\boldsymbol{w}_0} + b_w^{\text{ideal}}. \tag{32}$$

*Proof.* Define the event in Proposition 2 as

$$\mathcal{A}_{\text{target}} := \left\{\mathbb{E}_{\boldsymbol{w}^{(1:m)}, \boldsymbol{\epsilon}^{t(1:m)}, \boldsymbol{\epsilon}^{v(1:m)}}\|\hat{\boldsymbol{w}}_{\text{ideal}} - \boldsymbol{w}_0\|_2^2 \leq b_w^{\text{ideal}}\right\}.$$

Define the event in Proposition 3 as

$$\mathcal{A}_{\text{target,3}} := \left\{\text{Term 1 of Eq. (12)} \leq b_{\boldsymbol{w}_0}\right\}.$$

By the definition of $b_w$ in Eq. (32) and the definition of $\mathcal{A}_{\text{target,3}}$ and $\mathcal{A}_{\text{target}}$, we have $\mathcal{A}_{\text{target,3}} \cap \mathcal{A}_{\text{target}} \implies \|\hat{\boldsymbol{w}}_{\ell_2} - \boldsymbol{w}_0\|_2^2 \leq b_w$. Thus, we have

$$\begin{aligned}
\Pr\left\{\|\hat{\boldsymbol{w}}_{\ell_2} - \boldsymbol{w}_0\|_2^2 \leq b_w\right\} &\geq \Pr\left\{\mathcal{A}_{\text{target,3}} \cap \mathcal{A}_{\text{target}}\right\} \\
&= 1 - \Pr\left\{\mathcal{A}_{\text{target,3}}^c \cup \mathcal{A}_{\text{target}}^c\right\} \\
&\geq 1 - \Pr[\mathcal{A}_{\text{target,3}}^c] - \Pr[\mathcal{A}_{\text{target}}^c] \text{ (by the union bound)} \\
&\geq 1 - \frac{2}{p} - \frac{26m^2 n_v^2}{\min\{n_t, p\}^{0.4}} \text{ (by Proposition 3 and Proposition 2)} \\
&\geq 1 - \frac{27m^2 n_v^2}{\min\{n_t, p\}^{0.4}}. \tag{33}
\end{aligned}$$

The last inequality is because $\frac{2}{p} \leq \frac{1}{p^{0.4}}$ since $p^{0.6} \geq 256^{0.6} \approx 27.86 \geq 2$. The result of this theorem thus follows. $\square$

## J PROOF OF PROPOSITION 3

Define $\mathbf{P} := \mathbf{B}^T(\mathbf{B}\mathbf{B}^T)^{-1}\mathbf{B}$. In this section, we focus on estimating $\|(\mathbf{I}_p - \mathbf{P})\,\boldsymbol{w}_0\|_2^2$. Since $\mathbf{P}^2 = \mathbf{P}$, we know $\mathbf{P}$ is indeed a projection from $\mathbb{R}^p$ to the subspace spanned by the columns of $\mathbf{B}^T$ (i.e., the rows of $\mathbf{B}$). The following Lemma 15 shows that the subspace spanned by the columns of $\mathbf{B}^T$ has rotational symmetry. Then, we will use Lemma 16 to show how the rotational symmetry helps in estimating the expected value of the squared norm of the projected vector. We then prove Proposition 3 by utilizing Lemma 15 and Lemma 16.

**Lemma 15.** *The subspace spanned by the columns of $\mathbf{B}^T$ has rotational symmetry (with respect to the randomness of $\mathbf{X}^{t(1:m)}$ and $\mathbf{X}^{v(1:m)}$). Specifically, for any rotation $\mathbf{S} \in \mathsf{SO}(p)$ where $\mathsf{SO}(p) \subseteq \mathbb{R}^{p \times p}$ denotes the set of all rotations in $p$-dimensional space, the rotated random matrix $\mathbf{S}\mathbf{B}^T$ shares the same probability distribution with the original $\mathbf{B}^T$.*

*Proof.* Notice that for any $i = 1, 2, \cdots, m$,

$$
\mathbf{X}^{v(i)T} \left( \mathbf{I}_p - \frac{\alpha_t}{n_t} \mathbf{X}^{t(i)} \mathbf{X}^{t(i)T} \right) \mathbf{S}^T
$$
$$
= \mathbf{X}^{v(i)T} \mathbf{S}^T - \frac{\alpha_t}{n_t} \mathbf{X}^{v(i)T} \mathbf{S}^{-1} \mathbf{S} \mathbf{X}^{t(i)} \mathbf{X}^{t(i)T} \mathbf{S}^T
$$
$$
= \left( \mathbf{S} \mathbf{X}^{v(i)} \right)^T - \frac{\alpha_t}{n_t} \left( \mathbf{S} \mathbf{X}^{v(i)} \right)^T \left( \mathbf{S} \mathbf{X}^{t(i)} \right) \left( \mathbf{S} \mathbf{X}^{t(i)} \right)^T \quad \text{(since } \mathbf{S}^{-1} = \mathbf{S}^T \text{ because } \mathbf{S} \text{ is a rotation)}
$$
$$
= \left( \mathbf{S} \mathbf{X}^{v(i)} \right)^T \left( \mathbf{I}_p - \frac{\alpha_t}{n_t} \left( \mathbf{S} \mathbf{X}^{t(i)} \right) \left( \mathbf{S} \mathbf{X}^{t(i)} \right)^T \right).
$$

We thus have

$$
\mathbf{S}\mathbf{B}^T = 
\begin{bmatrix}
\mathbf{X}^{v(1)T} \left( \mathbf{I}_p - \frac{\alpha_t}{n_t} \mathbf{X}^{t(1)} \mathbf{X}^{t(1)T} \right) \mathbf{S}^T \\
\mathbf{X}^{v(2)T} \left( \mathbf{I}_p - \frac{\alpha_t}{n_t} \mathbf{X}^{t(2)} \mathbf{X}^{t(2)T} \right) \mathbf{S}^T \\
\vdots \\
\mathbf{X}^{v(m)T} \left( \mathbf{I}_p - \frac{\alpha_t}{n_t} \mathbf{X}^{t(m)} \mathbf{X}^{t(m)T} \right) \mathbf{S}^T
\end{bmatrix}^T
$$
$$
=
\begin{bmatrix}
\left( \mathbf{S}\mathbf{X}^{v(1)} \right)^T \left( \mathbf{I}_p - \frac{\alpha_t}{n_t} \left( \mathbf{S}\mathbf{X}^{t(1)} \right) \left( \mathbf{S}\mathbf{X}^{t(1)} \right)^T \right) \\
\left( \mathbf{S}\mathbf{X}^{v(2)} \right)^T \left( \mathbf{I}_p - \frac{\alpha_t}{n_t} \left( \mathbf{S}\mathbf{X}^{t(2)} \right) \left( \mathbf{S}\mathbf{X}^{t(2)} \right)^T \right) \\
\vdots \\
\left( \mathbf{S}\mathbf{X}^{v(m)} \right)^T \left( \mathbf{I}_p - \frac{\alpha_t}{n_t} \left( \mathbf{S}\mathbf{X}^{t(m)} \right) \left( \mathbf{S}\mathbf{X}^{t(m)} \right)^T \right)
\end{bmatrix}^T .
\tag{34}
$$

Because of the rotational symmetry of Gaussian distribution, we know that the rotated random matrices $\mathbf{S}\mathbf{X}^{v(i)}$ and $\mathbf{S}\mathbf{X}^{t(i)}$ have the same probability distribution with the original random matrices $\mathbf{X}^{v(i)}$ and $\mathbf{X}^{t(i)}$, respectively. Therefore, by Eq. (34), we can conclude that $\mathbf{S}\mathbf{B}^T$ has the same probability distribution as $\mathbf{B}^T$. The result of this lemma thus follows. $\quad\square$

The following lemma shows how to use rotational symmetry to calculate the expected value of the squared norm of the projected vector. Such a result has also been used in literature (e.g., (Belkin et al., 2020)).

**Lemma 16.** *Considering any random projection $\mathbf{P}_0 \in \mathbb{R}^{p \times p}$ to a $k$-dim subspace where the subspace has rotational symmetry, then for any given $\boldsymbol{v} \in \mathbf{R}^{p \times 1}$ we must have*

$$
\mathop{\mathbb{E}}_{\mathbf{P}_0} \|\mathbf{P}_0 \boldsymbol{v}\|_2^2 = \frac{k}{p} \|\boldsymbol{v}\|_2^2.
$$

*Proof.* Since the subspace has rotational symmetry, to calculate the expected value, we can fix a subspace and integral over all rotations. Specifically, consider any fixed projection $\mathbf{A}$ that projects

to $k$-dim subspace that is spanned by a set of orthogonal vectors $\boldsymbol{a}_1, \boldsymbol{a}_2, \cdots, \boldsymbol{a}_k \in \mathbb{R}^p$. Therefore, after projecting $\boldsymbol{v}$ with $\mathbf{A}$, the squared norm of the projected vector is equal to

$$\|\mathbf{A}\boldsymbol{v}\|_2^2 = \sum_{i=1}^{k} \langle \boldsymbol{a}_i, \boldsymbol{v} \rangle^2.$$

Noticing that applying a rotation in the projected space of $\mathbf{A}$ is equivalent to applying the rotation to $\boldsymbol{a}_1, \boldsymbol{a}_2, \cdots, \boldsymbol{a}_k$, we then have

$$\underset{\mathbf{P}_0}{\mathbb{E}} \|\mathbf{P}_0 \boldsymbol{v}\|_2^2$$

$$= \int_{\mathsf{so}(p)} \sum_{i=1}^{k} \langle \mathbf{S}\boldsymbol{a}_i, \boldsymbol{v} \rangle^2 d\mathbf{S}$$

$$= \int_{\mathsf{so}(p)} \sum_{i=1}^{k} \langle \mathbf{S}^{-1}\mathbf{S}\boldsymbol{a}_i, \mathbf{S}^{-1}\boldsymbol{v} \rangle^2 d\mathbf{S}$$

(since making a rotation $\mathbf{S}^{-1}$ on both vectors does not change the inner product value)

$$= \int_{\mathsf{so}(p)} \sum_{i=1}^{k} \langle \boldsymbol{a}_i, \mathbf{S}^{-1}\boldsymbol{v} \rangle^2 d\mathbf{S}$$

$$= \int_{\mathsf{so}(p)} \sum_{i=1}^{k} \langle \boldsymbol{a}_i, \mathbf{S}\boldsymbol{v} \rangle^2 d\mathbf{S}$$

$$= \frac{\|\boldsymbol{v}\|_2^2}{S_{p-1}} \int_{\mathcal{S}^{p-1}} \sum_{i=1}^{k} \langle \boldsymbol{a}_i, \tilde{\boldsymbol{v}} \rangle^2 d\tilde{\boldsymbol{v}}$$

$$= \frac{\|\boldsymbol{v}\|_2^2}{S_{p-1}} \sum_{i=1}^{k} \int_{\mathcal{S}^{p-1}} \langle \boldsymbol{a}_i, \tilde{\boldsymbol{v}} \rangle^2 d\tilde{\boldsymbol{v}}, \tag{35}$$

where $\mathcal{S}^{p-1}$ denotes the unit sphere in $\mathbb{R}^p$ and $S_{p-1}$ denotes its area. Since $\mathbf{A}$ can be any fixed projection to $k$-dim subspace, without loss of generality, we can simply let $\boldsymbol{a}_i$ be the $i$-th standard basis in $\mathbb{R}^p$ (i.e., only the $i$-th element is nonzero and is equal to 1). (Note that there are $p$ standard bases although $\mathbf{A}$ is spanned by only the first $k$ standard bases.) In this situation, we have

$$\int_{\mathcal{S}^{p-1}} \langle \boldsymbol{a}_i, \tilde{\boldsymbol{v}} \rangle^2 d\tilde{\boldsymbol{v}} = \int_{\mathcal{S}^{p-1}} \langle \boldsymbol{a}_j, \tilde{\boldsymbol{v}} \rangle^2 d\tilde{\boldsymbol{v}}, \text{ for all } i, j \in \{1, 2, \cdots, p\},$$

and

$$\sum_{i=1}^{p} \int_{\mathcal{S}^{p-1}} \langle \boldsymbol{a}_i, \tilde{\boldsymbol{v}} \rangle^2 d\tilde{\boldsymbol{v}} = \int_{\mathcal{S}^{p-1}} \sum_{i=1}^{p} \langle \boldsymbol{a}_i, \tilde{\boldsymbol{v}} \rangle^2 d\tilde{\boldsymbol{v}} = \int_{\mathcal{S}^{p-1}} \|\tilde{\boldsymbol{v}}\|_2^2 d\tilde{\boldsymbol{v}} = S_{p-1}.$$

Therefore, we have

$$\int_{\mathcal{S}^{p-1}} \langle \boldsymbol{a}_i, \tilde{\boldsymbol{v}} \rangle^2 d\tilde{\boldsymbol{v}} = \frac{1}{p} S_{p-1}.$$

By Eq. (35), we thus have

$$\underset{\mathbf{P}_0}{\mathbb{E}} \|\mathbf{P}_0 \boldsymbol{v}\|_2^2 = \frac{k}{p} \|\boldsymbol{v}\|_2^2.$$

The result of this lemma thus follows. $\qquad \square$

Now we are ready to prove Proposition 3.

*Proof of Proposition 3.* Define

$$\theta := \arccos \frac{\langle \boldsymbol{w}_0, \mathbf{P}\boldsymbol{w}_0 \rangle}{\|\boldsymbol{w}_0\|_2^2} \in [0, \pi/2].$$

Thus, we have

$$\|(\mathbf{I} - \mathbf{P})\boldsymbol{w}_0\|_2^2 = \sin^2\theta \cdot \|\boldsymbol{w}_0\|_2^2. \tag{36}$$

By Lemma 15, we know that the distribution of the hyper-plane spanned by the rows of $\mathbf{B}$ has rotational symmetry (which is a $mn_v$-dimensional space). Therefore, $\theta$ follows the same distribution of the angle between a uniformly distributed random vector $\boldsymbol{a} \in \mathbb{R}^p$ and a fixed $mn_v$-dimensional hyper-plane. To characterize such distribution of $\theta$ (or equivalently, $\sin\theta$), without loss of generality, we let $\boldsymbol{a} \sim \mathcal{N}(0, \mathbf{I}_p)$ and the hyper-plane be spanned by the first $mn_v$ standard bases of $\mathbb{R}^p$. Thus, we have

$$\cos^2\theta \sim \frac{\|\boldsymbol{a}_{[1:mn_v]}\|_2^2}{\|\boldsymbol{a}\|_2^2}, \quad \sin^2\theta \sim \frac{\|\boldsymbol{a}_{[mn_v+1:p]}\|_2^2}{\|\boldsymbol{a}\|_2^2}. \tag{37}$$

Notice that $\|\boldsymbol{a}_{[mn_v+1:p]}\|_2^2$ and $\|\boldsymbol{a}\|_2^2$ follow $\chi^2$ distribution with $p - mn_v$ degrees and $p$ degrees of freedom, respectively. By Lemma 9 and letting $x = \ln p$ in Lemma 9, we have

$$\Pr\left\{\|\boldsymbol{a}\|_2^2 \leq p - 2\sqrt{p\ln p}\right\} \leq \frac{1}{p}, \tag{38}$$

$$\Pr\left\{\|\boldsymbol{a}\|_2^2 \geq p + 2\sqrt{p\ln p} + 2\ln p\right\} \leq \frac{1}{p}, \tag{39}$$

$$\Pr\left\{\|\boldsymbol{a}_{[mn_v+1:p]}\|_2^2 \geq (p - mn_v) + 2\sqrt{(p - mn_v)\ln p} + 2\ln p\right\} \leq \frac{1}{p}. \tag{40}$$

$$\Pr\left\{\|\boldsymbol{a}_{[mn_v+1:p]}\|_2^2 \leq (p - mn_v) + 2\sqrt{(p - mn_v)\ln p}\right\} \leq \frac{1}{p}. \tag{41}$$

Because $p \geq 16$, by Lemma 3, we have

$$2\sqrt{\frac{\ln p}{p}} < 1 \implies 2\frac{\sqrt{p\ln p}}{p} < 1 \implies p - 2\sqrt{p\ln p} > 0. \tag{42}$$

We define

$$\mathcal{A}_{\text{target},3} \coloneqq \left\{\text{Term 1 of Eq. (12)} \leq b_{\boldsymbol{w}_0}\right\},$$
$$\tilde{\mathcal{A}}_{\text{target},3} \coloneqq \left\{\text{Term 1 of Eq. (12)} \geq \tilde{b}_{\boldsymbol{w}_0}\right\}.$$

We thus have

$$\Pr\left\{\mathcal{A}_{\text{target},3}^c\right\}$$
$$= \Pr\left\{\sin^2\theta \geq \frac{(p - mn_v) + 2\sqrt{(p - mn_v)\ln p} + 2\ln p}{p - 2\sqrt{p\ln p}}\right\}$$
$$\quad \text{(by Eq. (36) and the definition of } \mathbf{P})$$
$$= \Pr\left\{\frac{\|\boldsymbol{a}_{[mn_v+1:p]}\|_2^2}{\|\boldsymbol{a}\|_2^2} \geq \frac{(p - mn_v) + 2\sqrt{(p - mn_v)\ln p} + 2\ln p}{p - 2\sqrt{p\ln p}}\right\} \quad \text{(by Eq. (37))}$$
$$\leq \Pr\left\{\|\boldsymbol{a}\|_2^2 \leq p - 2\sqrt{p\ln p}\right\}$$
$$\quad + \Pr\left\{\|\boldsymbol{a}_{[mn_v+1:p]}\|_2^2 \geq (p - mn_v) + 2\sqrt{(p - mn_v)\ln p} + 2\ln p\right\}$$
$$\quad \text{(by Eq. (42) and the union bound)}$$
$$\leq \frac{2}{p} \quad \text{(by Eq. (38) and Eq. (40))}. \tag{43}$$

By Eq. (36) and Eq. (43), we thus have

$$\Pr[\mathcal{A}_{\text{target},3}] \geq 1 - \frac{2}{p}.$$

Similarly, using Eq. (39) and Eq. (41) (also by the union bound), we can prove

$$\Pr[\tilde{\mathcal{A}}_{\text{target},3}] \geq 1 - \frac{2}{p}.$$

By Lemma 16, we have $\mathbb{E}[\|\mathbf{P}\boldsymbol{w}_0\|_2^2] = \frac{mn_v}{p}\|\boldsymbol{w}_0\|_2^2$. Thus, we have

$$\mathbb{E}[\text{Term 1 of Eq. (12)}] = \frac{p - mn_v}{p}\|\boldsymbol{w}_0\|_2^2.$$

The result of this lemma thus follows. $\qquad\square$

## K  PROOF OF PROPOSITION 2

*Proof.* Define the event in Proposition 2 as

$$\mathcal{A}_{\text{target}} := \left\{ \mathop{\mathbb{E}}_{\boldsymbol{w}^{(1:m)}, \boldsymbol{\epsilon}^{t(1:m)}, \boldsymbol{\epsilon}^{v(1:m)}} \|\hat{\boldsymbol{w}}_{\text{ideal}} - \boldsymbol{w}_0\|_2^2 \leq b_w^{\text{ideal}} \right\}.$$

Define

$$\mathcal{A}_{\text{target},1} := \left\{ \lambda_{\max}(\mathbf{B}\mathbf{B}^T) \geq b_{\text{eig,min}} \mathbb{1}_{\{p > n_t\}} + c_{\text{eig,min}} \mathbb{1}_{\{p \leq n_t\}} \right\}$$

Combining Proposition 4 and Proposition 5 with the union bound, we have

$$\Pr[\mathcal{A}_{\text{target},1}] \geq 1 - \frac{23m^2n_v^2}{\min\{p, n_t\}^{0.4}}. \tag{44}$$

We adopt the event in Proposition 6 as

$$\mathcal{A}_{\text{target},2} := \left\{ \mathop{\mathbb{E}}_{\boldsymbol{w}^{(1:m)}, \boldsymbol{\epsilon}^{t(1:m)}, \boldsymbol{\epsilon}^{v(1:m)}} \|\delta\gamma\|_2^2 \leq b_\delta \right\}.$$

By Eq. (14), we have $\bigcap_{i=1}^2 \mathcal{A}_{\text{target},i} \implies \mathcal{A}_{\text{target}}$. Thus, we have

$$\Pr[\mathcal{A}_{\text{target}}] \geq \Pr\left\{ \bigcap_{i=1}^2 \mathcal{A}_{\text{target},i} \right\}$$

$$= 1 - \Pr\left\{ \bigcup_{i=1}^2 \mathcal{A}_{\text{target},i}^c \right\}$$

$$\geq 1 - \sum_{i=1}^2 \Pr[\mathcal{A}_{\text{target},i}^c] \text{ (by the union bound)}$$

$$\geq 1 - \frac{23m^2n_v^2}{\min\{p, n_t\}^{0.4}} - \frac{5mn_v}{n_t} - \frac{2mn_v}{p^{0.4}} \text{ (by Eq. (44) and Proposition 6)}$$

$$\geq 1 - \frac{26m^2n_v^2}{\min\{p, n_t\}^{0.4}}.$$

The last inequality is because $m \geq 1$, $n_v \geq 1$, and

$$n_t \geq 256 \implies n_t^{0.6} \geq 256^{0.6} \approx 27.86 \implies \frac{5}{n_t^{0.6}} \leq 1.$$

The result of this proposition thus follows. $\qquad\square$

## L  PROOF OF PROPOSITION 4

We prove Proposition 4 by estimating every element of $\mathbf{B}\mathbf{B}^T$. We split $\mathbf{B}\mathbf{B}^T$ into $m \times m$ blocks (each block is of size $n_v \times n_v$). Recalling the definition of $\mathbf{B}$ in Eq. (7), we identify three types of elements in $\mathbf{B}\mathbf{B}^T$ as diagonal elements (type 1), off-diagonal elements of diagonal block (type 2), and other elements (type 3). Fig. 4 illustrates these three types of elements when $m = 2$ and $n_v = 3$. In the rest of this part, we will first define and estimate each type of elements separately in Appendices L.1, L.2, and L.3. Using these results, we will then estimate the eigenvalues of $\mathbf{B}\mathbf{B}^T$ and finish the proof of Proposition 4.

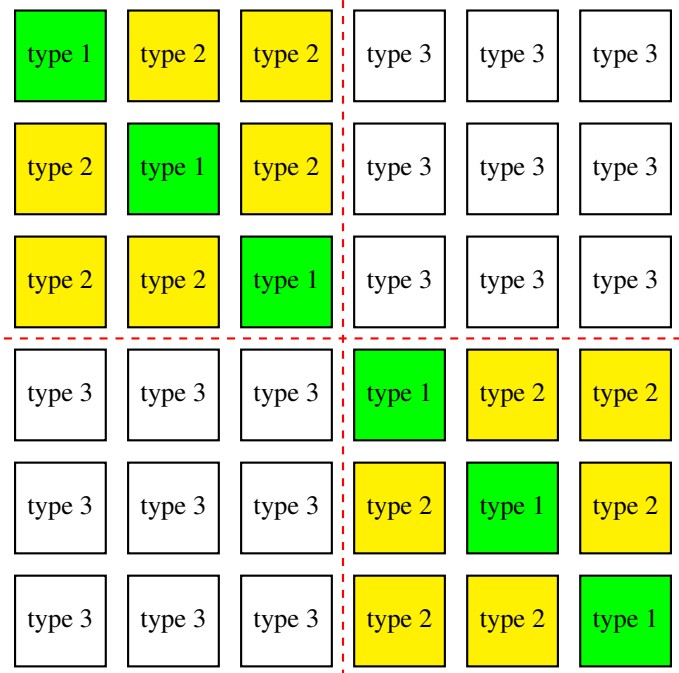

Figure 4: Illustration of three types of the elements in $\mathbf{BB}^T$ when $m = 2$ and $n_v = 3$. In this case $\mathbf{BB}^T$ is a $6 \times 6$ matrix (i.e., $\mathbb{R}^{(mn_v) \times (mn_v)}$). There are 4 (i.e., $m^2$) blocks (divided by the dashed red line) and each block is of size $3 \times 3$ (i.e., $n_v \times n_v$).

TYPE 1: DIAGONAL ELEMENTS

Type 1 elements are the diagonal elements of $\mathbf{BB}^T$, which can be denoted as

$$
[\mathbf{BB}^T]_{(i-1)n_v+j, \ (i-1)n_v+j}
$$
$$
= [\mathbf{X}^{v(i)}]_j^T \left( \mathbf{I}_p - \frac{\alpha_t}{n_t} \mathbf{X}^{t(i)} \mathbf{X}^{t(i)^T} \right)^T \left( \mathbf{I}_p - \frac{\alpha_t}{n_t} \mathbf{X}^{t(i)} \mathbf{X}^{t(i)^T} \right) [\mathbf{X}^{v(i)}]_j, \tag{45}
$$

where $i \in \{1, 2, \cdots, m\}$ corresponds to the $i$-th training task, $j \in \{1, 2, \cdots, n_v\}$ corresponds to the input vector of the $j$-th validation sample (of the $i$-th training task). To estimate Eq. (45), we have the following lemma.

**Lemma 17.** *For any $i \in \{1, 2, \cdots, m\}$ and any $j \in \{1, 2, \cdots, n_v\}$, when $p \geq n_t \geq 256$, we must have*

$$
\Pr \left\{ [\mathbf{BB}^T]_{(i-1)n_v+j, \ (i-1)n_v+j} \in [\underline{b}_1, \ \overline{b}_1] \right\} \geq 1 - \frac{5}{\sqrt{n_t}},
$$

*where*

$$
\underline{b}_1 := \max \left\{ 0, \ 1 - \alpha_t' \right\}^2 \left( n_t - \sqrt{2n_t \ln n_t} \right) + p - n_t - \sqrt{2(p - n_t) \ln n_t},
$$
$$
\overline{b}_1 := \max \{ \alpha_t', 1 - \alpha_t' \}^2 \left( n_t + \sqrt{2n_t \ln n_t} + \ln n_t \right) + p - n_t + \sqrt{2(p - n_t) \ln n_t} + \ln n_t.
$$

*Proof.* See Appendix L.1. □

TYPE 2: OFF-DIAGONAL ELEMENTS OF DIAGONAL BLOCKS

Type 2 elements are the off-diagonal elements of diagonal blocks. Similar to Eq. (45), Type 2 elements can be denoted by

$$
\begin{aligned}
&[\mathbf{BB}^T]_{(i-1)n_v+j,\ (i-1)n_v+k} \\
&=[\mathbf{X}^{v(i)}]_j^T \left(\mathbf{I}_p - \frac{\alpha_t}{n_t}\mathbf{X}^{t(i)}\mathbf{X}^{t(i)^T}\right)^T \left(\mathbf{I}_p - \frac{\alpha_t}{n_t}\mathbf{X}^{t(i)}\mathbf{X}^{t(i)^T}\right)[\mathbf{X}^{v(i)}]_k,
\end{aligned}
\tag{46}
$$

where $j \neq k$. We have the following lemma.

**Lemma 18.** *For any $i \in \{1, 2, \cdots, m\}$ and any $j, k \in \{1, 2, \cdots, n_v\}$ that $j \neq k$, when $p \geq n_t \geq 256$, we must have*

$$
\Pr\left\{\left|[\mathbf{BB}^T]_{(i-1)n_v+j,\ (i-1)n_v+k}\right| \leq \bar{b}_2\right\} \geq 1 - \frac{5}{n_t^{0.4}},
$$

*where*

$$
\bar{b}_2 := \max\{\alpha_t', 1 - \alpha_t'\}^2 \sqrt{n_t \ln n_t} + \sqrt{p}\ln p.
$$

*Proof.* See Appendix L.2. $\qquad\square$

TYPE 3: OTHER ELEMENTS (ELEMENTS OF OFF-DIAGONAL BLOCKS)

Type 3 elements are other elements that are not Type 1 or Type 2. In other words, Type 3 elements belong to off-diagonal blocks. Similar to Eq. (45) and Eq. (46), Type 3 elements can be denoted by

$$
\begin{aligned}
&[\mathbf{BB}^T]_{(i-1)n_v+j,\ (l-1)n_v+k} \\
&=[\mathbf{X}^{v(i)}]_j^T \left(\mathbf{I}_p - \frac{\alpha_t}{n_t}\mathbf{X}^{t(i)}\mathbf{X}^{t(i)^T}\right)^T \left(\mathbf{I}_p - \frac{\alpha_t}{n_t}\mathbf{X}^{t(l)}\mathbf{X}^{t(l)^T}\right)[\mathbf{X}^{v(l)}]_k \\
&=\left\langle \left(\mathbf{I}_p - \frac{\alpha_t}{n_t}\mathbf{X}^{t(i)}\mathbf{X}^{t(i)^T}\right)[\mathbf{X}^{v(i)}]_j,\ \left(\mathbf{I}_p - \frac{\alpha_t}{n_t}\mathbf{X}^{t(l)}\mathbf{X}^{t(l)^T}\right)[\mathbf{X}^{v(l)}]_k \right\rangle
\end{aligned}
\tag{47}
$$

where $i \neq l$. We have the following lemma.

**Lemma 19.** *For any $i, l \in \{1, 2, \cdots, m\}$ and any $j, k \in \{1, 2, \cdots, n_v\}$ that $i \neq l$, when $p \geq n_t \geq 256$, we must have*

$$
\Pr\left\{\left|[\mathbf{BB}^T]_{(i-1)n_v+j,\ (l-1)n_v+k}\right| \leq \bar{b}_3\right\} \geq 1 - \frac{13}{n_t^{0.4}},
$$

*where*

$$
\bar{b}_3 := 6\sqrt{p \ln p}.
$$

*Proof.* See Appendix L.3. $\qquad\square$

Now we are ready to prove Proposition 4.

*Proof of Proposition 4.* Define a few events as follows:

$$
\begin{aligned}
\mathcal{A}_1 &:= \left\{\text{all type 1 elements of } \mathbf{BB}^T \text{ are in } [\underline{b}_1,\ \bar{b}_1]\right\}, \\
\mathcal{A}_2 &:= \left\{\text{all type 2 elements of } \mathbf{BB}^T \text{ are in } [-\bar{b}_2,\ \bar{b}_2]\right\}, \\
\mathcal{A}_3 &:= \left\{\text{all type 3 elements of } \mathbf{BB}^T \text{ are in } [-\bar{b}_3,\ \bar{b}_3]\right\}.
\end{aligned}
$$

Notice that there are $mn_v$ type 1 elements, $mn_v(n_v - 1)$ type 2 elements, and $m(m - 1)n_v^2$ type 3 elements. By the union bound, Lemmas 17, 18, and 19, we have

$$
\Pr\{\mathcal{A}_1^c\} \leq \frac{5}{\sqrt{n_t}} \cdot mn_v \leq \frac{5}{n_t^{0.4}} \cdot mn_v,
\tag{48}
$$

$$
\Pr\{\mathcal{A}_2^c\} \leq \frac{5}{n_t^{0.4}} \cdot mn_v(n_v - 1),
\tag{49}
$$

$$
\Pr\{\mathcal{A}_3^c\} \leq \frac{13}{n_t^{0.4}} \cdot m(m - 1)n_v^2.
\tag{50}
$$

We first prove that $\bigcap_{i=1}^{3} \mathcal{A}_i \implies \mathcal{A}_{\text{target},1}$. To that end, recall the definition of disc $D(a_{i,i}, r_i(A))$ and the radius of the disc $r_i(A)$ for a matrix $A$ in Lemma 7. We now apply Lemma 7 on $\mathbf{BB}^T$. Because of $\mathcal{A}_2$ and $\mathcal{A}_3$, for any $i \in \{1, 2, \cdots, mn_v\}$, we have

$$r_i(\mathbf{BB}^T) = \sum_{j \in \{1,2,\cdots,mn_v\}, \ j \neq i} \left| [\mathbf{BB}^T]_{i,j} \right| \leq (n_v - 1)\bar{b}_2 + (m-1)n_v \bar{b}_3.$$

Because of $\bigcap_{i=1}^{3} \mathcal{A}_i$ and Lemma 7, we have all eigenvalues of $\mathbf{BB}^T$ is in

$$\left[ \underline{b}_1 - \left( (n_v - 1)\bar{b}_2 + (m-1)n_v \bar{b}_3 \right), \bar{b}_1 + \left( (n_v - 1)\bar{b}_2 + (m-1)n_v \bar{b}_3 \right) \right].$$

Since $p \geq n_t \geq 256$, we have

$$\sqrt{2(p - n_t)\ln n_t} \leq \sqrt{2p\ln p} \leq \sqrt{2p}\ln p, \text{ and } \sqrt{n_t \ln n_t} \leq \sqrt{p \ln p} \leq \sqrt{p}\ln p. \tag{51}$$

Since $n_t \geq 256$, by Eq. (60) of Lemma 20, we have $\ln n_t \leq \frac{1}{45} n_t$. Thus, we have

$$\ln n_t = \sqrt{\ln n_t \ln n_t} \leq \sqrt{\frac{1}{45}} \sqrt{n_t \ln n_t}. \tag{52}$$

Recalling the values of $\underline{b}_1, \bar{b}_1, \bar{b}_2,$ and $\bar{b}_3$ in Lemmas 17, 18, and 19, we have

$$\begin{aligned}
&\underline{b}_1 - \left( (n_v - 1)\bar{b}_2 + (m-1)n_v \bar{b}_3 \right) \\
&= \max\{0, \ 1 - \alpha'_t\}^2 \left( n_t - \sqrt{2n_t \ln n_t} \right) + p - n_t - \sqrt{2(p - n_t)\ln n_t} \\
&\quad - (n_v - 1) \left( \max\{\alpha'_t, 1 - \alpha'_t\}^2 \sqrt{n_t \ln n_t} + \sqrt{p}\ln p \right) \\
&\quad - (m-1)n_v \cdot 6\sqrt{p \ln p} \\
&\geq p + \left( \max\{0, 1 - \alpha'_t\}^2 - 1 \right) n_t \\
&\quad - \left( \left( \sqrt{2} + (n_v - 1) \right) \max\{\alpha'_t, 1 - \alpha'_t\}^2 + \sqrt{2} + (n_v - 1) + 6(m-1)n_v \right) \sqrt{p}\ln p \\
&\quad \text{(by Eq. (51) and } \max\{0, 1 - \alpha'_t\}^2 \leq \max\{\alpha'_t, 1 - \alpha'_t\}^2) \\
&\geq p + \left( \max\{0, 1 - \alpha'_t\}^2 - 1 \right) n_t - \left( (n_v + 1) \max\{\alpha'_t, 1 - \alpha'_t\}^2 + 6mn_v \right) \sqrt{p}\ln p \\
&= b_{\text{eig,min}},
\end{aligned}$$

and

$$\begin{aligned}
&\bar{b}_1 + \left( (n_v - 1)\bar{b}_2 + (m-1)n_v \bar{b}_3 \right) \\
&= \max\{\alpha'_t, 1 - \alpha'_t\}^2 \left( n_t + \sqrt{2n_t \ln n_t} + \ln n_t \right) + p - n_t + \sqrt{2(p - n_t)\ln n_t} + \ln n_t \\
&\quad + (n_v - 1) \left( \max\{\alpha'_t, 1 - \alpha'_t\}^2 \sqrt{n_t \ln n_t} + \sqrt{p}\ln p \right) \\
&\quad + (m-1)n_v \cdot 6\sqrt{p \ln p} \\
&\leq p + \left( \max\{\alpha'_t, 1 - \alpha'_t\}^2 - 1 \right) n_t \\
&\quad + \left( \left( \sqrt{2} + (n_v - 1) \right) \max\{\alpha'_t, 1 - \alpha'_t\}^2 + \frac{1}{\sqrt{45}} + \sqrt{2} + (n_v - 1) + 6(m-1)n_v \right) \sqrt{p}\ln p \\
&\quad \text{(by Eq. (51) and } \max\{0, 1 - \alpha'_t\}^2 \leq \max\{\alpha'_t, 1 - \alpha'_t\}^2) \\
&\leq p + \left( \max\{\alpha'_t, 1 - \alpha'_t\}^2 - 1 \right) n_t + \left( (n_v + 1) \max\{\alpha'_t, 1 - \alpha'_t\}^2 + 6mn_v \right) \sqrt{p}\ln p \\
&= b_{\text{eig,max}}.
\end{aligned}$$

Define

$$\mathcal{A}_{\text{target},1}^{(p \geq n_t)} := \left\{ b_{\text{eig,min}} \leq \lambda_{\min}(\mathbf{BB}^T) \leq \lambda_{\max}(\mathbf{BB}^T) \leq b_{\text{eig,max}} \right\},$$

Therefore, we have proven that $\bigcap_{i=1}^{3} \mathcal{A}_i \implies \mathcal{A}_{\text{target},1}^{(p \geq n_t)}$.

It remains to estimate the probability of $\mathcal{A}_{\text{target},1}^{(p \geq n_t)}$. To that end, we have

$$
\begin{aligned}
\Pr\left\{\mathcal{A}_{\text{target},1}^{(p \geq n_t)}\right\} &\geq \Pr\left\{\bigcap_{i=1}^{3} \mathcal{A}_i\right\} \text{ (since } \bigcap_{i=1}^{3} \mathcal{A}_i \implies \mathcal{A}_{\text{target},1}^{(p \geq n_t)}) \\
&= 1 - \Pr\left\{\bigcup_{i=1}^{3} \mathcal{A}_i^c\right\} \\
&\geq 1 - \sum_{i=1}^{3} \Pr\left\{\mathcal{A}_i^c\right\} \text{ (by the union bound)} \\
&\geq 1 - \frac{23 m^2 n_v^2}{n_t^{0.4}} \text{ (by Eqs. (48)(49)(50)).}
\end{aligned}
$$

The result of this proposition thus follows. $\qquad\square$

### L.1 PROOF OF LEMMA 17

Since all training inputs are independent with each other, without loss of generality, we let $i = 1$ and replace $[\mathbf{X}^{v(i)}]_j$ by a random vector $\boldsymbol{a} \sim \mathcal{N}(\mathbf{0}, \mathbf{I}_p) \in \mathbb{R}^{p \times 1}$ which is independent of $\mathbf{X}^{t(i)}$. In other words, estimating Eq. (45) is equivalent to estimate

$$
\left\|\left(\mathbf{I}_p - \frac{\alpha_t}{n_t}\mathbf{X}^{t(1)}\mathbf{X}^{t(1)^T}\right)\boldsymbol{a}\right\|_2^2.
$$

We further introduce some extra notations as follows. Since $p \geq n_t$, we can define the singular values of $\mathbf{X}^{t(1)} \in \mathbb{R}^{p \times n_t}$ as

$$
0 \leq \lambda_1^{t(1)} \leq \lambda_2^{t(1)} \leq \cdots \lambda_{n_t}^{t(1)}. \tag{53}
$$

Define

$$
\Lambda^{t(1)} := \text{diag}\left(\lambda_1^{t(1)}, \lambda_2^{t(1)}, \cdots, \lambda_{n_t}^{t(1)}\right) \in \mathbb{R}^{n_t \times n_t}.
$$

Do singular value decomposition of $\mathbf{X}^{t(1)}$, we have

$$
\mathbf{X}^{t(1)} = \mathbf{U}^{t(1)}\mathbf{D}^{t(1)}\mathbf{V}^{t(1)^T}. \tag{54}
$$

Notice that $\mathbf{U}^{t(1)} \in \mathbb{R}^{p \times p}$ is an orthogonal matrix, $\mathbf{D}^{t(1)} = \begin{bmatrix} \Lambda^{t(1)} \\ \mathbf{0} \end{bmatrix} \in \mathbb{R}^{p \times n_t}$, and $\mathbf{V}^{t(1)} \in \mathbb{R}^{n_t \times n_t}$ is an orthogonal matrix. Using these notations, we thus have

$$
\begin{aligned}
\mathbf{I}_p - \frac{\alpha_t}{n_t}\mathbf{X}^{t(1)}\mathbf{X}^{t(1)^T} &= \mathbf{U}^{t(1)}\left(\mathbf{I}_p - \frac{\alpha_t}{n_t}\mathbf{D}^{t(1)}\mathbf{D}^{t(1)^T}\right)\mathbf{U}^{t(1)^T} \text{ (by Eq. (54))} \\
&= \mathbf{U}^{t(1)}\begin{bmatrix} \mathbf{I}_{n_t} - \frac{\alpha_t}{n_t}\Lambda^{t(1)^2} & \mathbf{0} \\ \mathbf{0} & \mathbf{I}_{p-n_t} \end{bmatrix}\mathbf{U}^{t(1)^T} \text{ (by the definition of } \mathbf{D}^{t(1)}). \tag{55}
\end{aligned}
$$

The following two lemmas will be useful in the proof of Lemma 17.

**Lemma 20.** *If $x \geq 16$, then*

$$
\frac{3}{16}x \geq \ln x, \tag{56}
$$

$$
x + \sqrt{2x \ln x} + \ln x \leq 2x, \tag{57}
$$

$$
x - \sqrt{2x \ln x} \geq \frac{x}{3}, \tag{58}
$$

$$
x + \ln x \leq \frac{6}{7}\sqrt{2}x. \tag{59}
$$

*Further, if $x \geq 256$, then*

$$
\frac{1}{45}x \geq \ln x. \tag{60}
$$

*Proof.* We prove each equation sequentially as follows.

**Proof of Eq. (56):** When $x \geq 16$, we have

$$\frac{\partial \left( \frac{3}{16}x - \ln x \right)}{\partial x} = \frac{3}{16} - \frac{1}{x} > 0.$$

Thus, $\frac{3}{16}x - \ln x$ is monotone increasing when $x \geq 16$. Thus, in order to prove $\frac{3}{16}x \geq \ln x$ for all $x \geq 16$, we only need to prove $\frac{3}{16} \cdot 16 \geq \ln 16$. Notice that $\ln(16) \approx 2.7726 < 3$. Therefore, Eq. (56) holds.

**Proof of Eq. (57):** Using Eq. (56), we have

$$\sqrt{2x \ln x} + \ln x \leq \left( \sqrt{\frac{3}{8}} + \frac{3}{16} \right) x \approx 0.7999x \leq x.$$

Eq. (57) thus follows.

**Proof of Eq. (58):** Doing square root on both sides of $\frac{3}{16}x \geq \ln x$, we thus have

$$\frac{\sqrt{3}}{4}\sqrt{x} \geq \sqrt{\ln x}$$

$$\implies \sqrt{\frac{3}{8}} \cdot x \geq \sqrt{2x \ln x}$$

$$\implies x - \sqrt{2x \ln x} \geq \left( 1 - \sqrt{\frac{3}{8}} \right) x \geq \frac{1}{3}x \text{ (since } 1 - \sqrt{\frac{3}{8}} \approx 0.3876 \geq \frac{1}{3}\text{).}$$

Eq. (58) thus follows.

**Proof of Eq. (59):** We have

$$x + \ln x \leq x + \frac{3}{16}x \text{ (by Eq. (56))}$$

$$= \frac{19}{16\sqrt{2}}\sqrt{2}x$$

$$\leq \frac{6}{7}\sqrt{2}x \text{ (since } \frac{19}{16\sqrt{2}} \approx 0.8397 \leq \frac{6}{7}\text{).}$$

Eq. (59) thus follows.

**Proof of Eq. (60):** When $x \geq 256$, we have

$$\frac{\partial \left( \frac{1}{45}x - \ln x \right)}{\partial x} = \frac{1}{45} - \frac{1}{x} > 0.$$

Thus, $\frac{1}{45}x - \ln x$ is monotone increasing when $x \geq 256$. Thus, in order to prove $\frac{1}{45}x \geq \ln x$ for all $x \geq 256$, we only need to prove $\frac{1}{45} \cdot 256 \geq \ln 256$. Notice that $256/45 - \ln 256 \approx 0.1437 \geq 0$. Therefore, Eq. (60) holds. □

Now we are ready to prove Lemma 17.

*Proof of Lemma 17.* Recalling $\mathbf{U}^{t(1)}$ in Eq. (54), we define

$$\boldsymbol{a}' := \mathbf{U}^{t(1)^T}\boldsymbol{a} \in \mathbb{R}^{p \times 1}, \tag{61}$$

$$\chi^2_{n_t} := \sum_{i=1}^{n_t} \boldsymbol{a}_i'^2, \quad \chi^2_{p-n_t} := \sum_{i=p-n_t+1}^{p} \boldsymbol{a}_i'^2. \tag{62}$$

We then have

$$
\left\| \left( \mathbf{I}_p - \frac{\alpha_t}{n_t} \mathbf{X}^{t(1)} \mathbf{X}^{t(1)T} \right) \boldsymbol{a} \right\|_2^2
$$

$$
= \boldsymbol{a}^T \mathbf{U}^{t(1)} \begin{bmatrix} \left( \mathbf{I}_{n_t} - \frac{\alpha_t}{n_t} \Lambda^{t(1)2} \right)^2 & \mathbf{0} \\ \mathbf{0} & \mathbf{I}_{p-n_t} \end{bmatrix} \mathbf{U}^{t(1)T} \boldsymbol{a} \quad \text{(by Eq. (55))}
$$

$$
= \boldsymbol{a}'^T \begin{bmatrix} \left( \mathbf{I}_{n_t} - \frac{\alpha_t}{n_t} \Lambda^{t(1)2} \right)^2 & \mathbf{0} \\ \mathbf{0} & \mathbf{I}_{p-n_t} \end{bmatrix} \boldsymbol{a}' \quad \text{(by Eq. (61))}
$$

$$
= \sum_{i=1}^{n_t} \left( 1 - \frac{\alpha_t}{n_t} \lambda_i^{t(1)2} \right)^2 \boldsymbol{a}_i'^2 + \sum_{i=n_t+1}^{p} \boldsymbol{a}_i'^2
$$

$$
\in \left[ \min_{j \in \{1,2,\cdots,n_t\}} \left\{ \left( 1 - \frac{\alpha_t}{n_t} \lambda_j^{t(1)} \right)^2 \right\} \sum_{i=1}^{n_t} \boldsymbol{a}_i'^2 + \sum_{i=n_t+1}^{p} \boldsymbol{a}_i'^2, \right.
$$

$$
\left. \max_{j \in \{1,2,\cdots,n_t\}} \left( 1 - \frac{\alpha_t}{n_t} \lambda_j^{t(1)} \right)^2 \sum_{i=1}^{n_t} \boldsymbol{a}_i'^2 + \sum_{i=n_t+1}^{p} \boldsymbol{a}_i'^2 \right]
$$

$$
= \left[ \max \left\{ 0, 1 - \frac{\alpha_t}{n_t} \lambda_{n_t}^{t(1)2} \right\}^2 \chi_{n_t}^2 + \chi_{p-n_t}^2, \right.
$$

$$
\left. \left( \max \left\{ \left| 1 - \frac{\alpha_t}{n_t} \lambda_{n_t}^{t(1)2} \right|, \left| 1 - \frac{\alpha_t}{n_t} \lambda_1^{t(1)2} \right| \right\} \right)^2 \chi_{n_t}^2 + \chi_{p-n_t}^2 \right] \quad \text{(by Eq. (62) and Eq. (53)).}
$$

$$
\tag{63}
$$

Because of rotational symmetry of normal distribution of $\boldsymbol{a}$, we know that $\chi_{numTrain}^2$ and $\chi_{p-n_t}^2$ follows $\chi^2$ distribution of $p$ and $p - n_t$ degrees of freedom, respectively[5]. We define several events as follows:

$$
\mathcal{A}_1 := \left\{ \chi_{n_t}^2 > n_t + 2\sqrt{n_t \ln \sqrt{n_t}} + 2 \ln \sqrt{n_t} \right\},
$$

$$
\mathcal{A}_2 := \left\{ \chi_{p-n_t}^2 > p - n_t + 2\sqrt{(p - n_t) \ln \sqrt{n_t}} + 2 \ln \sqrt{n_t} \right\},
$$

$$
\mathcal{A}_3 := \left\{ \chi_{p-n_t}^2 < p - n_t - 2\sqrt{(p - n_t) \ln \sqrt{n_t}} \right\},
$$

$$
\mathcal{A}_4 := \left\{ \lambda_{n_t}^{t(1)} > \sqrt{p} + \sqrt{n_t} + \ln \sqrt{n_t} \right\},
$$

$$
\mathcal{A}_5 := \left\{ \lambda_1^{t(1)} < \sqrt{p} - \sqrt{n_t} - \ln \sqrt{n_t} \right\},
$$

$$
\mathcal{A}_6 := \left\{ \chi_{n_t}^2 < n_t - 2\sqrt{n_t \ln \sqrt{n_t}} \right\}.
$$

---

[5]$\mathbf{X}^{v(i)}$ and $\mathbf{X}^{t(i)}$ are independent, so $\boldsymbol{a}$ and $\mathbf{U}^{t(1)}$ are also independent. The calculation of $\chi_{n_t}^2$ (or $\chi_{p-n_t}^2$) can utilize the rotational symmetry of $\boldsymbol{a}$ (or $\boldsymbol{a}'$) is because $\chi_{n_t}^2$ (or $\chi_{p-n_t}^2$) represents the squared norm of the result that projects $\boldsymbol{a}'$ into a $n_t$-dim (or $(p - n_t)$-dim) subspace.

We have

$$\Pr_{\mathbf{X}^{t(1)},\boldsymbol{a}}\left\{\bigcap_{i=1}^6 \mathcal{A}_i^c\right\} = 1 - \Pr_{\mathbf{X}^{t(1)},\boldsymbol{a}}\left\{\bigcup_{i=1}^6 \mathcal{A}_i\right\}$$

$$\geq 1 - \sum_{i=1}^3 \Pr_{\boldsymbol{a}}\{\mathcal{A}_i\} - \Pr_{\mathbf{X}^{t(1)}}\{\mathcal{A}_4 \cup \mathcal{A}_5\} - \Pr_{\boldsymbol{a}}\{\mathcal{A}_6\} \text{ (by the union bound)}$$

$$\geq 1 - 4e^{-\ln\sqrt{n_t}} - 2e^{-(\ln\sqrt{n_t})^2/2} \text{ (by Lemma 9 and Lemma 8)}$$

$$= 1 - \frac{4}{\sqrt{n_t}} - \frac{2}{\exp\left(\frac{1}{2}\ln\sqrt{n_t} \cdot \ln\sqrt{n_t}\right)}$$

$$\geq 1 - \frac{4}{\sqrt{n_t}} - \frac{2}{\exp\left(\ln\sqrt{n_t}\right)} \text{ (since } \ln\sqrt{n_t} \geq 2 \text{ when } n_t \geq 256)$$

$$= 1 - \frac{6}{\sqrt{n_t}}. \tag{64}$$

Define the target event

$$\mathcal{A} := \left\{ \left\|\left(\mathbf{I}_p - \frac{\alpha_t}{n_t}\mathbf{X}^{t(1)}\mathbf{X}^{t(1)T}\right)\boldsymbol{a}\right\|_2^2 \in [\underline{b}_1, \bar{b}_1] \right\}.$$

It remains to prove that $\bigcap_{i=1}^6 \mathcal{A}_i^c \implies \mathcal{A}$. To that end, when $\bigcap_{i=1}^6 \mathcal{A}_i^c$, we have

$$\left\|\left(\mathbf{I}_p - \frac{\alpha_t}{n_t}\mathbf{X}^{t(1)}\mathbf{X}^{t(1)T}\right)\boldsymbol{a}\right\|_2^2$$

$$\geq \max\left\{0,\ 1 - \lambda_{n_t}^{t(1)2}\right\}^2 \chi_{n_t}^2 + \chi_{p-n_t}^2 \text{ (by Eq. (63))}$$

$$\geq \max\{0,\ 1 - \alpha_t'\}^2 \left(n_t - \sqrt{2n_t \ln n_t}\right) + p - n_t - \sqrt{2(p-n_t)\ln n_t}$$

$$\text{(by } \mathcal{A}_3^c \text{ and } \mathcal{A}_6^c)$$

$$= \underline{b}_1,$$

and

$$\left\|\left(\mathbf{I}_p - \frac{\alpha_t}{n_t}\mathbf{X}^{t(1)}\mathbf{X}^{t(1)T}\right)\boldsymbol{a}\right\|_2^2$$

$$\leq \left(\max\left\{\left|1 - \frac{\alpha_t}{n_t}\lambda_{n_t}^{t(1)2}\right|,\ \left|1 - \frac{\alpha_t}{n_t}\lambda_1^{t(1)2}\right|\right\}\right)^2 \chi_{n_t}^2 + \chi_{p-n_t}^2 \text{ (by Eq. (63))}$$

$$\leq \left(\max\left\{\left|1 - \frac{\alpha_t}{n_t}(\sqrt{p} - \sqrt{n_t} - \ln\sqrt{n_t})^2\right|,\ \left|1 - \frac{\alpha_t}{n_t}(\sqrt{p} + \sqrt{n_t} + \ln\sqrt{n_t})^2\right|\right\}\right)^2$$

$$\cdot \left(n_t + \sqrt{2n_t \ln n_t} + \ln n_t\right) + p - n_t + \sqrt{2(p-n_t)\ln n_t} + \ln n_t$$

$$\text{(by } \mathcal{A}_1^c, \mathcal{A}_2^c, \mathcal{A}_4^c \text{ and } \mathcal{A}_5^c)$$

$$\leq \max\{\alpha_t', 1 - \alpha_t'\}^2 \cdot \left(n_t + \sqrt{2n_t \ln n_t} + \ln n_t\right) + p - n_t + \sqrt{2(p-n_t)\ln n_t} + \ln n_t$$

$$= \bar{b}_1.$$

$$\square$$

### L.2 PROOF OF LEMMA 18

Since all samples are independent with each other, without loss of generality, we let $i = 1$ and replace $[\mathbf{X}^{v(i)}]_j$, $[\mathbf{X}^{v(i)}]_k$ by two *i.i.d.* random vector $\boldsymbol{a}, \boldsymbol{b} \sim \mathcal{N}(\mathbf{0}, \mathbf{I}_p) \in \mathbb{R}^{p \times 1}$ which are independent of $\mathbf{X}^{t(i)}$. In other words, estimating Eq. (46) is equivalent to estimate

$$\boldsymbol{a}^T \left(\mathbf{I}_p - \frac{\alpha_t}{n_t}\mathbf{X}^{t(1)}\mathbf{X}^{t(1)T}\right)^T \left(\mathbf{I}_p - \frac{\alpha_t}{n_t}\mathbf{X}^{t(1)}\mathbf{X}^{t(1)T}\right)\boldsymbol{b}.$$

*Proof of Lemma 18.* Recalling $\mathbf{U}^{t(1)}$ in Eq. (54), we define

$$\boldsymbol{a}' := \mathbf{U}^{t(1)^T}\boldsymbol{a} \in \mathbb{R}^{p\times 1}, \quad \boldsymbol{b}' := \mathbf{U}^{t(1)^T}\boldsymbol{b} \in \mathbb{R}^{p\times 1}, \tag{65}$$

$$\phi_{n_t} := \sum_{i=1}^{n_t} \boldsymbol{a}'_i\boldsymbol{b}'_i, \quad \phi_{p-n_t} := \sum_{i=p-n_t+1}^{p} \boldsymbol{a}'_i\boldsymbol{b}'_i, \tag{66}$$

We then have

$$\left| \boldsymbol{a}^T \left( \mathbf{I}_p - \frac{\alpha_t}{n_t}\mathbf{X}^{t(1)}\mathbf{X}^{t(1)^T} \right)^2 \boldsymbol{b} \right|$$

$$= \left| \boldsymbol{a}^T\mathbf{U}^{t(1)} \left[ \begin{matrix} \left(\mathbf{I}_{n_t} - \frac{\alpha_t}{n_t}\Lambda^{t(1)^2}\right)^2 & \mathbf{0} \\ \mathbf{0} & \mathbf{I}_{p-n_t} \end{matrix} \right] \mathbf{U}^{t(1)^T}\boldsymbol{b} \right| \text{ (by Eq. (55))}$$

$$= \left| \boldsymbol{a}'^T \left[ \begin{matrix} \left(\mathbf{I}_{n_t} - \frac{\alpha_t}{n_t}\Lambda^{t(1)^2}\right)^2 & \mathbf{0} \\ \mathbf{0} & \mathbf{I}_{p-n_t} \end{matrix} \right] \boldsymbol{b}' \right| \text{ (by Eq. (65))}$$

$$= \left| \sum_{i=1}^{n_t} \left(1 - \frac{\alpha_t}{n_t}\lambda_i^{t(1)^2}\right)^2 \boldsymbol{a}'_i\boldsymbol{b}'_i + \sum_{i=n_t+1}^{p} \boldsymbol{a}'_i\boldsymbol{b}'_i \right|$$

$$\leq \left( \max\left\{ \left|1 - \frac{\alpha_t}{n_t}\lambda_1^{t(1)^2}\right|, \left|1 - \frac{\alpha_t}{n_t}\lambda_{n_t}^{t(1)^2}\right| \right\} \right)^2 |\phi_{n_t}| + |\phi_{p-n_t}|$$

(by Eq. (65) and Eq. (53)). $\tag{67}$

Because of the rotational symmetry of normal distribution, we know that $\phi_{n_t}$ and $\phi_{p-n_t}$ have the same probability distribution if $\boldsymbol{a}'$ and $\boldsymbol{b}'$ follow *i.i.d.* $\mathcal{N}(\mathbf{0}, \mathbf{I}_p)$[6]. Define several events as follows:

$$\mathcal{A}_1 := \left\{ |\phi_{n_t}| > \sqrt{n_t \ln n_t} \right\},$$
$$\mathcal{A}_2 := \left\{ |\phi_{p-n_t}| > \sqrt{p}\ln p \right\},$$
$$\mathcal{A}_3 := \left\{ \lambda_{n_t}^{t(1)} > \sqrt{p} + \sqrt{n_t} + \ln\sqrt{n_t} \right\},$$
$$\mathcal{A}_4 := \left\{ \lambda_1^{t(1)} < \sqrt{p} - \sqrt{n_t} - \ln\sqrt{n_t} \right\}.$$

We have

$$\Pr_{\mathbf{X}^{t(1)},\boldsymbol{a},\boldsymbol{b}}\left\{ \bigcap_{i=1}^{4} \mathcal{A}_i^c \right\} = 1 - \Pr_{\mathbf{X}^{t(1)},\boldsymbol{a},\boldsymbol{b}}\left\{ \bigcup_{i=1}^{4} \mathcal{A}_i \right\}$$

$$\geq 1 - \Pr_{\boldsymbol{a},\boldsymbol{b}}\{\mathcal{A}_1\} - \Pr_{\boldsymbol{a},\boldsymbol{b}}\{\mathcal{A}_2\} - \Pr_{\mathbf{X}^{t(1)}}\{\mathcal{A}_3 \cup \mathcal{A}_4\} \text{ (by the union bound)}$$

$$\geq 1 - \frac{2}{n_t^{0.4}} - \frac{2}{p} - 2e^{-(\ln\sqrt{n_t})^2/2} \text{ (by Lemma 11 and Lemma 8)}$$

$$\geq 1 - \frac{3}{n_t^{0.4}} - \frac{2}{\exp\left(\frac{1}{2}\ln\sqrt{n_t} \cdot \ln\sqrt{n_t}\right)} \text{ (since } p \geq n_t \text{ and } n_t^{0.6} \geq 2 \text{ when } n_t \geq 256)$$

$$\geq 1 - \frac{3}{n_t^{0.4}} - \frac{2}{\exp\left(\ln\sqrt{n_t}\right)} \text{ (since } \ln\sqrt{n_t} \geq 2 \text{ when } n_t \geq 256)$$

$$\geq 1 - \frac{5}{n_t^{0.4}}. \tag{68}$$

Define the target event

$$\mathcal{A} := \left\{ \left| \boldsymbol{a}^T \left( \mathbf{I}_p - \frac{\alpha_t}{n_t}\mathbf{X}^{t(1)}\mathbf{X}^{t(1)^T} \right)^2 \boldsymbol{b} \right| \leq \bar{b}_2 \right\}.$$

---

[6]We can utilize the rotational symmetry because $\phi_{n_t}$ (or $\phi_{p-n_t}$) can be viewed as the result of the following steps: 1) project $\boldsymbol{a}'$ and $\boldsymbol{b}'$ into a fixed subspace that is spanned by the first $n_t$ (or last $p - n_t$) standard basis vectors in $\mathbb{R}^p$; 2) calculate the inner product of these two projected vectors.

It remains to show that $\bigcap_{i=1}^{4} \mathcal{A}_i^c \implies \mathcal{A}$. To that end, when $\bigcap_{i=1}^{4} \mathcal{A}_i^c$, we have

$$
\begin{aligned}
& \left| \boldsymbol{a}^T \left( \mathbf{I}_p - \frac{\alpha_t}{n_t} \mathbf{X}^{t(1)} \mathbf{X}^{t(1)T} \right)^2 \boldsymbol{b} \right| \\
& \leq \left( \max \left\{ \left| 1 - \frac{\alpha_t}{n_t} \lambda_1^{t(1)2} \right|, \left| 1 - \frac{\alpha_t}{n_t} \lambda_{n_t}^{t(1)2} \right| \right\} \right)^2 |\phi_{n_t}| + |\phi_{p-n_t}| \quad \text{(by Eq. (67))} \\
& \leq \left( \max \left\{ \left| 1 - \frac{\alpha_t}{n_t} (\sqrt{p} - \sqrt{n_t} - \ln \sqrt{n_t})^2 \right|, \left| 1 - \frac{\alpha_t}{n_t} (\sqrt{p} + \sqrt{n_t} + \ln \sqrt{n_t})^2 \right| \right\} \right)^2 \sqrt{n_t \ln n_t} \\
& \quad + \sqrt{p} \ln p \quad \text{(by } \mathcal{A}_1^c, \mathcal{A}_2^c, \mathcal{A}_3^c, \text{ and } \mathcal{A}_4^c) \\
& \leq \max\{\alpha_t', 1 - \alpha_t'\}^2 \sqrt{n_t \ln n_t} + \sqrt{p} \ln p. \quad \quad (69)
\end{aligned}
$$

By Eq. (69) (which implies $\bigcap_{i=1}^{4} \mathcal{A}_i^c \implies \mathcal{A}$) and Eq. (68), the result of this lemma thus follows. $\quad\square$

### L.3 PROOF OF LEMMA 19

Because all inputs are *i.i.d.*, Eq. (47) is the inner product of two independent vectors, where each vector follows the same distribution as the vector

$$
\boldsymbol{\rho} := \left( \mathbf{I}_p - \frac{\alpha_t}{n_t} \mathbf{X}^{t(1)} \mathbf{X}^{t(1)T} \right) \boldsymbol{a} \in \mathbb{R}^p, \quad\quad (70)
$$

where $\boldsymbol{a} \sim \mathcal{N}(\mathbf{0}, \mathbf{I}_p)$ and is independent of $\mathbf{X}^{t(1)}$. In other words, it is equivalent to estimate $\boldsymbol{\rho}_1^T \boldsymbol{\rho}_2$ where $\boldsymbol{\rho}_1$ and $\boldsymbol{\rho}_2$ follow the *i.i.d.* shown in (70). If we can characterize the probability distribution of $\boldsymbol{\rho}$, then we can estimate Eq. (47). The following lemma shows the rotational symmetry of Eq. (70).

**Lemma 21.** *The probability distribution of $\boldsymbol{\rho}$ has rotational symmetry. In other words, for any rotation $\mathbf{S} \in \mathsf{SO}(p)$ where $\mathsf{SO}(p) \subseteq \mathbb{R}^{p \times p}$ denotes the set of all rotations in $p$ dimension, the rotated random vector $\mathbf{S}\boldsymbol{\rho}$ shares the same probability distribution with the original vector $\boldsymbol{\rho}$.*

*Proof.* By Eq. (70), we have

$$
\begin{aligned}
\mathbf{S}\boldsymbol{\rho} =& \mathbf{S}\boldsymbol{a} - \frac{\alpha_t}{n_t} \mathbf{S} \mathbf{X}^{t(1)} \mathbf{X}^{t(1)T} \boldsymbol{a} \\
=& \mathbf{S}\boldsymbol{a} - \frac{\alpha_t}{n_t} \mathbf{S} \mathbf{X}^{t(1)} \mathbf{X}^{t(1)T} \mathbf{S}^{-1} \mathbf{S}\boldsymbol{a} \\
=& \mathbf{S}\boldsymbol{a} - \frac{\alpha_t}{n_t} \mathbf{S} \mathbf{X}^{t(1)} \mathbf{X}^{t(1)T} \mathbf{S}^T \mathbf{S}\boldsymbol{a} \\
& \text{(because } \mathbf{S}^{-1} = \mathbf{S}^T, \text{ as a rotation is an orthogonal matrix)} \\
=& \mathbf{S}\boldsymbol{a} - \frac{\alpha_t}{n_t} (\mathbf{S}\mathbf{X}^{t(1)})(\mathbf{S}\mathbf{X}^{t(1)})^T (\mathbf{S}\boldsymbol{a}).
\end{aligned}
$$

Notice that $\mathbf{S}\boldsymbol{a}$ and $\mathbf{S}\mathbf{X}^{t(1)}$ are the rotated vectors of $\boldsymbol{a}$ and $\mathbf{X}^{t(1)}$, respectively. Since $\boldsymbol{a}$ and $\mathbf{X}^{t(1)}$ are independent Gaussian vector/matrix, by rotational symmetry, we know their distribution and independence do not affected by a common rotation $\mathbf{S}$. Thus, $\mathbf{S}\boldsymbol{a} - \frac{\alpha_t}{n_t} (\mathbf{S}\mathbf{X}^{t(1)})(\mathbf{S}\mathbf{X}^{t(1)})^T (\mathbf{S}\boldsymbol{a})$ has the same probability distribution as $\boldsymbol{\rho}$. The result of this lemma thus follows. $\quad\square$

The following lemma characterize the distribution of the angle between two independent random vectors where both vectors have rotational symmetry.

**Lemma 22.** *Consider two* i.i.d. *random vector $\boldsymbol{c}_1, \boldsymbol{c}_2 \in \mathbb{R}^p$ that have rotational symmetry. We then have*

$$
\Pr \left\{ \frac{|\boldsymbol{c}_1^T \boldsymbol{c}_2|}{\|\boldsymbol{c}_1\|_2 \|\boldsymbol{c}_2\|_2} \geq \frac{\sqrt{p \ln p}}{p - 2\sqrt{p \ln p}} \right\} \leq \frac{2}{p} + \frac{2}{p^{0.4}}.
$$

*Proof.* Notice that $\frac{\left|\boldsymbol{c}_1^T \boldsymbol{c}_2\right|}{\|\boldsymbol{c}_1\|_2 \|\boldsymbol{c}_2\|_2}$ denotes the angle between $\boldsymbol{c}_1$ and $\boldsymbol{c}_2$. By rotational symmetry, it is equivalent to prove that the angle between two independent random vectors that have rotational symmetry. To that end, consider two *i.i.d.* random vectors $\boldsymbol{x}_1, \boldsymbol{x}_2 \sim \mathcal{N}(\boldsymbol{0}, \mathbf{I}_p)$. The distribution of the angle between $\boldsymbol{x}_1$ and $\boldsymbol{x}_2$ should be the same as the angle between $\boldsymbol{c}_1$ and $\boldsymbol{c}_2$. In other words, we have

$$\frac{\left|\boldsymbol{x}_1^T \boldsymbol{x}_2^T\right|}{\|\boldsymbol{x}_1\|_2 \|\boldsymbol{x}_2\|_2} \sim \frac{\left|\boldsymbol{c}_1^T \boldsymbol{c}_2\right|}{\|\boldsymbol{c}_1\|_2 \|\boldsymbol{c}_2\|_2}. \tag{71}$$

By Lemma 11, we have

$$\Pr\left\{\left|\boldsymbol{x}_1^T \boldsymbol{x}_2\right| > \sqrt{p \ln p}\right\} \leq \frac{2}{p^{0.4}}. \tag{72}$$

Noticing that $\|\boldsymbol{x}_1\|_2$ and $\|\boldsymbol{x}_2\|_2$ follow chi-square distribution with $p$ degrees of freedom, by Lemma 9, we have

$$\Pr\left\{\|\boldsymbol{x}_1\|_2^2 \leq p - 2\sqrt{p \ln p}\right\} \leq e^{-\ln p} = \frac{1}{p}, \tag{73}$$

$$\Pr\left\{\|\boldsymbol{x}_2\|_2^2 \leq p - 2\sqrt{p \ln p}\right\} \leq e^{-\ln p} = \frac{1}{p}. \tag{74}$$

By Eqs. (72)(73)(74) and the union bound, we thus have

$$\Pr\left\{\frac{\left|\boldsymbol{x}_1^T \boldsymbol{x}_2\right|}{\|\boldsymbol{x}_1\|_2 \|\boldsymbol{x}_2\|_2} \geq \frac{\sqrt{p \ln p}}{p - 2\sqrt{p \ln p}}\right\} \leq \frac{2}{p} + \frac{2}{p^{0.4}}.$$

By Eq. (71), the result of this lemma thus follows. $\qquad\square$

Now we are ready to prove Lemma 19.

*Proof of Lemma 19.* Since $p \geq n_t \geq 256$, by Eq. (60) of Lemma 20, we have

$$p - 2\sqrt{p \ln p} \geq \left(1 - 2\sqrt{\frac{1}{45}}\right) p \geq \left(1 - \sqrt{5}\sqrt{\frac{1}{45}}\right) p = \frac{2p}{3}. \tag{75}$$

We define some events as follows:

$$\mathcal{A}_1 := \left\{\|\boldsymbol{\rho}_1\|_2^2 \geq \bar{b}_1\right\},$$
$$\mathcal{A}_2 := \left\{\|\boldsymbol{\rho}_2\|_2^2 \geq \bar{b}_1\right\},$$
$$\mathcal{A}_3 := \left\{\frac{\left|\boldsymbol{\rho}_1^T \boldsymbol{\rho}_2\right|}{\|\boldsymbol{\rho}_1\|_2 \|\boldsymbol{\rho}_2\|_2} \geq \frac{\sqrt{p \ln p}}{p - 2\sqrt{p \ln p}}\right\},$$
$$\mathcal{A} := \left\{\left|\boldsymbol{\rho}_1^T \boldsymbol{\rho}_2\right| \leq \bar{b}_3\right\}.$$

By Lemma 17, we have

$$\Pr\{\mathcal{A}_1\} \leq \frac{5}{\sqrt{n_t}}, \ \Pr\{\mathcal{A}_2\} \leq \frac{5}{\sqrt{n_t}}. \tag{76}$$

By Lemma 22, we have

$$\Pr\{\mathcal{A}_3\} \leq \frac{2}{p} + \frac{2}{p^{0.4}}. \tag{77}$$

Since $n_t \geq 256$, by letting $x = n_t$ in Eq. (57) of Lemma 20, we have

$$n_t + \sqrt{2n_t \ln n_t} + \ln n_t \leq 2n_t. \tag{78}$$

We also have

$$p - n_t + \sqrt{2(p - n_t) \ln n_t} + \ln n_t$$

$$\leq p - n_t + \sqrt{2p \cdot \frac{1}{45} n_t} + \frac{1}{45} n_t \quad \text{(by Eq. (60) in Lemma 20)}$$

$$\leq p - n_t + \sqrt{\frac{2}{45} p} + \frac{1}{45} n_t \quad \text{(since } p \geq n_t\text{)}$$

$$= \left(1 + \sqrt{\frac{2}{45}}\right) p - \frac{44}{45} n_t. \tag{79}$$

Therefore, we have

$$\bar{b}_1 \leq \max\{\alpha'_t, 1 - \alpha'_t\}^2 \cdot 2n_t + \left(1 + \sqrt{\frac{2}{45}}\right) p - \frac{44}{45} n_t,$$

$$\leq \max\{\alpha'_t, 1\}^2 \cdot 4p \quad \text{(since } p \geq n_t\text{)}. \tag{80}$$

First, we want to show $\bigcap_{i=1}^3 \mathcal{A}_i^c \implies \mathcal{A}$. To that end, when $\bigcap_{i=1}^3 \mathcal{A}_i^c$, we have

$$\left|\boldsymbol{\rho}_1^T \boldsymbol{\rho}_2\right| = \|\boldsymbol{\rho}_1\|_2 \|\boldsymbol{\rho}_2\|_2 \frac{\left|\boldsymbol{\rho}_1^T \boldsymbol{\rho}_2\right|}{\|\boldsymbol{\rho}_1\|_2 \|\boldsymbol{\rho}_2\|_2}$$

$$\leq \bar{b}_1 \cdot \frac{\sqrt{p \ln p}}{p - 2\sqrt{p \ln p}} \quad \text{(by } \bigcap_{i=1}^3 \mathcal{A}_i^c \text{ and Lemma 17)}$$

$$\leq 4p \cdot \frac{\sqrt{p \ln p}}{2p/3} \quad \text{(by Eq. (75) and Eq. (80))}$$

$$= 6\sqrt{p \ln p}.$$

Thus, we have proven that $\bigcap_{i=1}^3 \mathcal{A}_i^c \implies \mathcal{A}$, which implies that

$$\Pr\{\mathcal{A}\} \geq \Pr\left\{\bigcap_{i=1}^3 \mathcal{A}_i^c\right\}$$

$$= 1 - \Pr\left\{\bigcup_{i=1}^3 \mathcal{A}_i\right\}$$

$$\geq 1 - \sum_{i=1}^3 \Pr\{\mathcal{A}_i\} \quad \text{(by the union bound)}$$

$$\geq 1 - \frac{10}{\sqrt{n_t}} - \frac{2}{p} - \frac{2}{p^{0.4}} \quad \text{(by Eq. (76) and Eq. (77))}$$

$$\geq 1 - \frac{13}{n_t^{0.4}} \quad \text{(by } p^{0.6} \geq 2 \text{ since } p \geq n_t \geq 256\text{)}.$$

The result of this lemma thus follows. $\qquad\square$

## M  PROOF OF PROPOSITION 5

**Lemma 23.** *For any $i \in \{1, 2, \cdots, m\}$ and any $j \in \{1, 2, \cdots, n_v\}$, when $n_t \geq p \geq 256$, we must have*

$$\Pr\left\{[\mathbf{BB}^T]_{(i-1)n_v+j, \ (i-1)n_v+j} \in [\underline{c}_1, \ \bar{c}_1]\right\} \geq 1 - \frac{2}{p} - \frac{2}{\sqrt{n_t}},$$

*where*

$$\underline{c}_1 := \max\{0, 1 - \alpha'_t\}^2 \cdot (p - 2\sqrt{p \ln p}),$$

$$\bar{c}_1 := \max\{\alpha'_t, 1 - \alpha'_t\}^2 \left(p + 2\sqrt{p \ln p} + 2 \ln p\right).$$

*Proof.* See Appendix M.1 ◻

**Lemma 24.** *For any $i \in \{1, 2, \cdots, m\}$ and any $j, k \in \{1, 2, \cdots, n_v\}$ that $j \neq k$, when $n_t \geq p \geq 256$ and $\alpha_t \leq 0.2$, we must have*

$$\Pr\left\{\left|[\mathbf{B}\mathbf{B}^T]_{(i-1)n_v+j,\ (i-1)n_v+k}\right| \leq \bar{c}_2\right\} \geq 1 - \frac{2}{p^{0.4}} - \frac{2}{\sqrt{n_t}},$$

*where*

$$\bar{c}_2 := \max\{\alpha_t', 1 - \alpha_t'\}^2 \cdot \sqrt{p \ln p}.$$

*Proof.* See Appendix M.2. ◻

**Lemma 25.** *For any $i, l \in \{1, 2, \cdots, m\}$ that $i \neq l$ and for any $j, k \in \{1, 2, \cdots, n_v\}$, when $n_t \geq p \geq 256$ and $\alpha_t \leq 0.2$, we must have*

$$\Pr\left\{\left|[\mathbf{B}\mathbf{B}^T]_{(i-1)n_v+j,\ (l-1)n_v+k}\right| \leq \bar{c}_3\right\} \geq 1 - \frac{6}{p} - \frac{4}{\sqrt{n_t}} - \frac{2}{p^{0.4}},$$

*where*

$$\bar{c}_3 := 2\max\{\alpha_t', 1 - \alpha_t'\}^2 \cdot \sqrt{p \ln p}.$$

*Proof.* See Appendix M.3. ◻

Now we are ready to prove Proposition 5

*Proof of Proposition 5.* Define a few events as follows:

$$\mathcal{A}_1 := \left\{\text{all type 1 elements of } \mathbf{B}\mathbf{B}^T \text{ are in } [\underline{c}_1,\ \bar{c}_1]\right\},$$
$$\mathcal{A}_2 := \left\{\text{all type 2 elements of } \mathbf{B}\mathbf{B}^T \text{ are in } [-\bar{c}_2,\ \bar{c}_2]\right\},$$
$$\mathcal{A}_3 := \left\{\text{all type 3 elements of } \mathbf{B}\mathbf{B}^T \text{ are in } [-\bar{c}_3,\ \bar{c}_3]\right\}.$$

Because $n_t \geq p \geq 256$, we have

$$\frac{1}{\sqrt{n_t}} \leq \frac{1}{\sqrt{p}} \leq \frac{1}{p^{0.4}}, \ \frac{2}{p} \leq \frac{1}{p^{0.4}}. \tag{81}$$

Since $p \geq 256$, by Eq. (60) in Lemma 20, we have

$$\ln p = \sqrt{\ln p \cdot \ln p} \leq \sqrt{\frac{1}{45} p \ln p}. \tag{82}$$

Notice that there are $mn_v$ type 1 elements, $mn_v(n_v - 1)$ type 2 elements, and $m(m-1)n_v^2$ type 3 elements. By the union bound, Lemmas 17, 18, 19, and Eq. (81), we have

$$\Pr[\mathcal{A}_1^c] \leq \frac{3mn_v}{p^{0.4}}, \ \Pr[\mathcal{A}_2^c] \leq \frac{4mn_v(n_v - 1)}{p^{0.4}}, \ \Pr[\mathcal{A}_3^c] \leq \frac{9m(m-1)n_v^2}{p^{0.4}}. \tag{83}$$

By $\bigcap_{i=1}^{3} \mathcal{A}_i$ and Lemma 7, we have all eigenvalues of $\mathbf{B}\mathbf{B}^T$ in

$$\left[\underline{c}_1 - ((n_v - 1)\bar{c}_2 + (m-1)n_v\bar{c}_3),\ \bar{c}_1 + ((n_v - 1)\bar{c}_2 + (m-1)n_v\bar{c}_3)\right].$$

Recalling the values of $\underline{c}_1$, $\bar{c}_2$, and $\bar{c}_3$ in Lemmas 23, 24, 25, we have

$$\begin{aligned}
&\underline{c}_1 - ((n_v - 1)\bar{c}_2 + (m-1)n_v\bar{c}_3) \\
&= \max\{0, 1 - \alpha_t'\}^2 p \\
&\quad - \left(2\max\{0, 1 - \alpha_t'\}^2 + (n_v - 1 + 2(m-1)n_v)\max\{\alpha_t', 1 - \alpha_t'\}^2\right)\sqrt{p \ln p} \\
&\geq \max\{0, 1 - \alpha_t'\}^2 p - 2mn_v\max\{\alpha_t', 1 - \alpha_t'\}^2\sqrt{p \ln p} \\
&\quad (\text{noticing that } \max\{\alpha_t', 1 - \alpha_t'\}^2 \geq \max\{0, 1 - \alpha_t'\}^2) \\
&= c_{\text{eig,min}}.
\end{aligned}$$

and

$$\bar{c}_1 + ((n_v - 1)\bar{c}_2 + (m - 1)n_v\bar{c}_3)$$
$$= \max\{\alpha_t', 1 - \alpha_t'\}^2 (p + 2\ln p)$$
$$\quad + (n_v + 1 + 2(m-1)n_v)\max\{\alpha_t', 1 - \alpha_t'\}^2\sqrt{p\ln p}$$
$$\leq \max\{\alpha_t', 1 - \alpha_t'\}^2\left(p + (2mn_v + 1)\sqrt{p\ln p}\right) \quad \text{(by Eq. (82))}$$
$$= c_{\text{eig,max}}.$$

Define the event

$$\mathcal{A}_{\text{target},1}^{(p \leq n_t)} := \left\{ c_{\text{eig,min}} \leq \lambda_{\min}(\mathbf{B}\mathbf{B}^T) \leq \lambda_{\max}(\mathbf{B}\mathbf{B}^T) \leq c_{\text{eig,max}} \right\},$$

Therefore, we have proven $\bigcap_{i=1}^{3} \mathcal{A}_i \implies \mathcal{A}_{\text{target},1}^{(p \leq n_t)}$. Thus, we have

$$\Pr[\mathcal{A}_{\text{target},1}^{(p \leq n_t)}] \geq \Pr\left\{ \bigcap_{i=1}^{3} \mathcal{A}_i \right\}$$
$$= 1 - \Pr\left\{ \bigcup_{i=1}^{3} \mathcal{A}_i^c \right\}$$
$$\geq 1 - \sum_{i=1}^{3} \Pr[\mathcal{A}_i] \quad \text{(by th union bound)}$$
$$\geq 1 - \frac{16m^2n_v^2}{p^{0.4}} \quad \text{(by Eq. (83))}.$$

The result of this proposition thus follows. $\qquad\qquad\qquad\qquad\qquad\qquad\qquad\square$

## M.1 PROOF OF LEMMA 23

*Proof.* When $p < n_t$, we define the singular values of $\mathbf{X}^{t(1)} \in \mathbb{R}^{p \times n_t}$ as

$$0 \leq \lambda_1^{t(1)} \leq \lambda_2^{t(1)} \leq \cdots \lambda_p^{t(1)}.$$

Define

$$\Lambda^{t(1)} := \text{diag}\left(\lambda_1^{t(1)}, \lambda_2^{t(1)}, \cdots, \lambda_p^{t(1)}\right) \in \mathbb{R}^{p \times p}.$$

We can still do the singular value decomposition as Eq. (54), but here $\mathbf{D}^{t(1)} = \begin{bmatrix} \Lambda^{t(1)} & \mathbf{0} \end{bmatrix} \in \mathbb{R}^{p \times n_t}$ since $p < n_t$. Using these notations, we thus have

$$\mathbf{I}_p - \frac{\alpha_t}{n_t}\mathbf{X}^{t(1)}\mathbf{X}^{t(1)T} = \mathbf{U}^{t(1)}\left(\mathbf{I}_p - \frac{\alpha_t}{n_t}\Lambda^{t(1)2}\right)\mathbf{U}^{t(1)T}.$$

Similar to Eq. (63), we have

$$\left\|\left(\mathbf{I}_p - \frac{\alpha_t}{n_t}\mathbf{X}^{t(1)}\mathbf{X}^{t(1)T}\right)\boldsymbol{a}\right\|_2^2 \in \left[\max\left\{0, 1 - \frac{\alpha_t}{n_t}\lambda_p^{t(1)2}\right\}^2 \chi_p^2, \right.$$
$$\left. \left(\max\left\{\left|1 - \frac{\alpha_t}{n_t}\lambda_p^{t(1)2}\right|, \left|1 - \frac{\alpha_t}{n_t}\lambda_1^{t(1)2}\right|\right\}\right)^2 \chi_p^2\right], \quad (84)$$

where $\chi_p^2 := \left\|\mathbf{U}^{t(1)T}\boldsymbol{a}\right\|_2^2 = \|\boldsymbol{a}\|_2^2$ follows $\chi^2$ distribution with $p$ degrees of freedom. We define several events as follows:

$$\mathcal{A}_1 := \left\{\chi_p^2 < p - 2\sqrt{p\ln p}\right\},$$
$$\mathcal{A}_2 := \left\{\chi_p^2 > p + 2\sqrt{p\ln p} + 2\ln p\right\},$$
$$\mathcal{A}_3 := \left\{\lambda_p^{t(1)} > \sqrt{n_t} + \sqrt{p} + \ln\sqrt{n_t}\right\},$$
$$\mathcal{A}_4 := \left\{\lambda_1^{t(1)} < \sqrt{n_t} - \sqrt{p} - \ln\sqrt{n_t}\right\}.$$

We have

$$\Pr_{\mathbf{X}^{t(1)}, a} \left\{ \bigcap_{i=1}^{4} \mathcal{A}_i^c \right\} = 1 - \Pr \left\{ \bigcup_{i=1}^{4} \mathcal{A}_i \right\}$$

$$\geq 1 - \sum_{i=1}^{2} \Pr_{a} \{\mathcal{A}_i\} - \Pr_{\mathbf{X}^{t(1)}} \{\mathcal{A}_3 \cup \mathcal{A}_4\} \text{ (by the union bound)}$$

$$\geq 1 - \frac{2}{p} - 2 \exp \left( -(\ln \sqrt{n_t})^2 / 2 \right) \text{ (by Lemma 9 and Lemma 8)}$$

$$\geq 1 - \frac{2}{p} - \frac{2}{\sqrt{n_t}} \text{ (since } \ln \sqrt{n_t} \geq 2 \text{ when } n_t \geq 256).$$

Define the target event

$$\mathcal{A} := \left\{ \left\| \left( \mathbf{I}_p - \frac{\alpha_t}{n_t} \mathbf{X}^{t(1)} \mathbf{X}^{t(1)T} \right) a \right\|_2^2 \in [\underline{c}_1, \overline{c}_1] \right\}.$$

It remains to prove that $\bigcap_{i=1}^{4} \mathcal{A}_i^c \implies \mathcal{A}$. To that end, when $\bigcap_{i=1}^{4} \mathcal{A}_i^c$, we have

$$\left\| \left( \mathbf{I}_p - \frac{\alpha_t}{n_t} \mathbf{X}^{t(1)} \mathbf{X}^{t(1)T} \right) a \right\|_2^2$$

$$\geq \max \left\{ 0, 1 - \frac{\alpha_t}{n_t} \lambda_p^{t(1)2} \right\}^2 \chi_p^2 \text{ (by Eq. (84))}$$

$$\geq \max \left\{ 0, 1 - \frac{\alpha_t}{n_t} (\sqrt{n_t} + \sqrt{p} + \ln \sqrt{n_t})^2 \right\}^2 \left( p - 2\sqrt{p \ln p} \right) \text{ (by } \mathcal{A}_1^c \text{ and } \mathcal{A}_3^c)$$

$$= \max \{ 0, 1 - \alpha_t' \}^2 \cdot (p - 2\sqrt{p \ln p}).$$

When $\bigcap_{i=1}^{4} \mathcal{A}_i^c$, we also have

$$\left\| \left( \mathbf{I}_p - \frac{\alpha_t}{n_t} \mathbf{X}^{t(1)} \mathbf{X}^{t(1)T} \right) a \right\|_2^2$$

$$\leq \left( \max \left\{ \left| 1 - \frac{\alpha_t}{n_t} \lambda_p^{t(1)2} \right|, \left| 1 - \frac{\alpha_t}{n_t} \lambda_1^{t(1)2} \right| \right\} \right)^2 \chi_p^2 \text{ (by Eq. (84))}$$

$$\leq \max \left\{ \left| 1 - \frac{\alpha_t}{n_t} (\sqrt{n_t} - \sqrt{p} - \ln \sqrt{n_t})^2 \right|, \left| 1 - \frac{\alpha_t}{n_t} (\sqrt{n_t} + \sqrt{p} + \ln \sqrt{n_t})^2 \right| \right\}^2$$

$$\cdot \left( p + 2\sqrt{p \ln p} + 2 \ln p \right) \text{ (by } \mathcal{A}_2^c, \mathcal{A}_3^c \text{ and } \mathcal{A}_4^c)$$

$$\leq \max \{ \alpha_t', 1 - \alpha_t' \}^2 \left( p + 2\sqrt{p \ln p} + 2 \ln p \right).$$

$$\square$$

## M.2   Proof of Lemma 24

*Proof.* Similar to Eq. (67), we have

$$\left| a^T \left( \mathbf{I}_p - \frac{\alpha_t}{n_t} \mathbf{X}^{t(1)} \mathbf{X}^{t(1)T} \right)^2 b \right|$$

$$\leq \left( \max \left\{ \left| 1 - \frac{\alpha_t}{n_t} \lambda_1^{t(1)2} \right|, \left| 1 - \frac{\alpha_t}{n_t} \lambda_p^{t(1)2} \right| \right\} \right)^2 |\phi_p|. \tag{85}$$

Define several events as follows:

$$\mathcal{A}_1 := \left\{ |\phi_p| > \sqrt{p \ln p} \right\},$$

$$\mathcal{A}_2 := \left\{ \lambda_p^{t(1)} > \sqrt{n_t} + \sqrt{p} + \ln \sqrt{n_t} \right\},$$

$$\mathcal{A}_3 := \left\{ \lambda_1^{t(1)} < \sqrt{n_t} - \sqrt{p} - \ln \sqrt{n_t} \right\}.$$

When $\bigcap_{i=1}^{3} \mathcal{A}_i^c$, we have

$$\left| \boldsymbol{a}^T \left( \mathbf{I}_p - \frac{\alpha_t}{n_t} \mathbf{X}^{t(1)} \mathbf{X}^{t(1)T} \right)^2 \boldsymbol{b} \right|$$

$$\leq \max \left\{ \left| 1 - \frac{\alpha_t}{n_t} \left( \sqrt{n_t} - \sqrt{p} - \ln \sqrt{n_t} \right)^2 \right|, \left| 1 - \frac{\alpha_t}{n_t} \left( \sqrt{n_t} + \sqrt{p} + \ln \sqrt{n_t} \right)^2 \right| \right\}^2$$

$$\cdot \sqrt{p \ln p} \quad \text{(by Eq. (85), } \mathcal{A}_1^c, \mathcal{A}_2^c, \text{ and } \mathcal{A}_3^c\text{)}$$

$$\leq \max\{\alpha_t', 1 - \alpha_t'\}^2 \cdot \sqrt{p \ln p}.$$

Thus, we have

$$\Pr \left\{ \left| \boldsymbol{a}^T \left( \mathbf{I}_p - \frac{\alpha_t}{n_t} \mathbf{X}^{t(1)} \mathbf{X}^{t(1)T} \right)^2 \boldsymbol{b} \right| \leq \sqrt{p \ln p} \right\}$$

$$\geq \Pr \left\{ \bigcap_{i=1}^{3} \mathcal{A}_i^c \right\}$$

$$= 1 - \Pr \left\{ \bigcup_{i=1}^{3} \mathcal{A}_i \right\}$$

$$\geq 1 - \Pr_{\boldsymbol{a}, \boldsymbol{b}}[\mathcal{A}_1] - \Pr_{\mathbf{X}^{t(1)}}[\mathcal{A}_2 \cup \mathcal{A}_3] \quad \text{(by the union bound)}$$

$$\geq 1 - \frac{2}{p^{0.4}} - 2 \exp\left( -(\ln \sqrt{n_t})^2/2 \right) \quad \text{(by Lemma 11 and Lemma 8)}$$

$$\geq 1 - \frac{2}{p^{0.4}} - \frac{2}{\sqrt{n_t}} \quad \text{(since } \ln \sqrt{n_t} \geq 2 \text{ when } n_t \geq 256\text{)}.$$

$\square$

## M.3 PROOF OF LEMMA 25

*Proof.* Define several events as follows:

$$\mathcal{A}_1 := \left\{ \|\boldsymbol{\rho}_1\|_2^2 \geq \bar{c}_1 \right\},$$

$$\mathcal{A}_2 := \left\{ \|\boldsymbol{\rho}_2\|_2^2 \geq \bar{c}_1 \right\},$$

$$\mathcal{A}_3 := \left\{ \frac{|\boldsymbol{\rho}_1^T \boldsymbol{\rho}_2|}{\|\boldsymbol{\rho}_1\|_2 \|\boldsymbol{\rho}_2\|_2} \geq \frac{\sqrt{p \ln p}}{p - 2\sqrt{p \ln p}} \right\},$$

$$\mathcal{A} := \left\{ |\boldsymbol{\rho}_1^T \boldsymbol{\rho}_2| \leq \bar{c}_3 \right\}.$$

By Lemma 23, we have

$$\Pr[\mathcal{A}_1] \leq \frac{2}{p} + \frac{2}{\sqrt{n_t}}, \ \Pr[\mathcal{A}_2] \leq \frac{2}{p} + \frac{2}{\sqrt{n_t}}. \tag{86}$$

By Lemma 22, we have

$$\Pr[\mathcal{A}_3] \leq \frac{2}{p} + \frac{2}{p^{0.4}}. \tag{87}$$

When $p \geq 256$, by Eq. (60) of Lemma 20, we have

$$\ln p \leq \frac{1}{45} p, \ \sqrt{p \ln p} \leq \sqrt{\frac{1}{45}} p. \tag{88}$$

First, we want to show $\bigcap_{i=1}^{3} \mathcal{A}_i^c \implies \mathcal{A}$. To that end, when $\bigcap_{i=1}^{3} \mathcal{A}_i^c$, we have

$$
\begin{aligned}
\left|\boldsymbol{\rho}_1^T \boldsymbol{\rho}_2\right| &= \|\boldsymbol{\rho}_1\|_2 \|\boldsymbol{\rho}_2\|_2 \frac{\left|\boldsymbol{\rho}_1^T \boldsymbol{\rho}_2\right|}{\|\boldsymbol{\rho}_1\|_2 \|\boldsymbol{\rho}_2\|_2} \\
&\leq \max\{\alpha_t', 1 - \alpha_t'\}^2 \cdot (p + 2\sqrt{p \ln p} + 2 \ln p) \cdot \frac{\sqrt{p \ln p}}{p - 2\sqrt{p \ln p}} \ \text{(by} \bigcap_{i=1}^{3} \mathcal{A}_i^c) \\
&\leq \max\{\alpha_t', 1 - \alpha_t'\}^2 \cdot \frac{1 + 2\sqrt{\frac{1}{45}} + \frac{2}{45}}{1 - 2\sqrt{\frac{1}{45}}} \sqrt{p \ln p} \ \text{(by Eq. (88))} \\
&\leq 2 \max\{\alpha_t', 1 - \alpha_t'\}^2 \cdot \sqrt{p \ln p} \ \text{(because} \ \frac{1 + 2\sqrt{\frac{1}{45}} + \frac{2}{45}}{1 - 2\sqrt{\frac{1}{45}}} \approx 1.91 \leq 2\text{).}
\end{aligned}
$$

Thus, we have proven $\bigcap_{i=1}^{3} \mathcal{A}_i^c \implies \mathcal{A}$, which implies that

$$
\begin{aligned}
\Pr[\mathcal{A}] &\geq \Pr\left\{\bigcap_{i=1}^{3} \mathcal{A}_i^c\right\} \\
&= 1 - \Pr\left\{\bigcup_{i=1}^{3} \mathcal{A}_i\right\} \\
&\geq 1 - \sum_{i=1}^{3} \Pr[\mathcal{A}_i] \ \text{(by the union bound)} \\
&\geq 1 - \frac{6}{p} - \frac{4}{\sqrt{n_t}} - \frac{2}{p^{0.4}} \ \text{(by Eq. (86) and Eq. (87)).}
\end{aligned}
$$

The result of this lemma thus follows. $\qquad\square$

## N   PROOF OF PROPOSITION 6

Plugging Eq. (1) into Eq. (7), we have

$$
\gamma = \begin{bmatrix}
\mathbf{X}^{v(1)^T}\left(\mathbf{I}_p - \frac{\alpha_t}{n_t}\mathbf{X}^{t(1)}\mathbf{X}^{t(1)^T}\right)\boldsymbol{w}^{(1)} - \frac{\alpha_t}{n_t}\mathbf{X}^{v(1)^T}\mathbf{X}^{t(1)}\boldsymbol{\epsilon}^{t(1)} + \boldsymbol{\epsilon}^{v(1)} \\
\mathbf{X}^{v(2)^T}\left(\mathbf{I}_p - \frac{\alpha_t}{n_t}\mathbf{X}^{t(2)}\mathbf{X}^{t(2)^T}\right)\boldsymbol{w}^{(2)} - \frac{\alpha_t}{n_t}\mathbf{X}^{v(2)^T}\mathbf{X}^{t(2)}\boldsymbol{\epsilon}^{t(2)} + \boldsymbol{\epsilon}^{v(2)} \\
\vdots \\
\mathbf{X}^{v(m)^T}\left(\mathbf{I}_p - \frac{\alpha_t}{n_t}\mathbf{X}^{t(m)}\mathbf{X}^{t(m)^T}\right)\boldsymbol{w}^{(m)} - \frac{\alpha_t}{n_t}\mathbf{X}^{v(m)^T}\mathbf{X}^{t(m)}\boldsymbol{\epsilon}^{t(m)} + \boldsymbol{\epsilon}^{v(m)}
\end{bmatrix}.
$$

By Eq. (11), we thus have

$$
\delta\gamma = \begin{bmatrix}
\mathbf{X}^{v(1)^T}\left(\mathbf{I}_p - \frac{\alpha_t}{n_t}\mathbf{X}^{t(1)}\mathbf{X}^{t(1)^T}\right)(\boldsymbol{w}^{(1)} - \boldsymbol{w}_0) - \frac{\alpha_t}{n_t}\mathbf{X}^{v(1)^T}\mathbf{X}^{t(1)}\boldsymbol{\epsilon}^{t(1)} + \boldsymbol{\epsilon}^{v(1)} \\
\mathbf{X}^{v(2)^T}\left(\mathbf{I}_p - \frac{\alpha_t}{n_t}\mathbf{X}^{t(2)}\mathbf{X}^{t(2)^T}\right)(\boldsymbol{w}^{(2)} - \boldsymbol{w}_0) - \frac{\alpha_t}{n_t}\mathbf{X}^{v(2)^T}\mathbf{X}^{t(2)}\boldsymbol{\epsilon}^{t(2)} + \boldsymbol{\epsilon}^{v(2)} \\
\vdots \\
\mathbf{X}^{v(m)^T}\left(\mathbf{I}_p - \frac{\alpha_t}{n_t}\mathbf{X}^{t(m)}\mathbf{X}^{t(m)^T}\right)(\boldsymbol{w}^{(m)} - \boldsymbol{w}_0) - \frac{\alpha_t}{n_t}\mathbf{X}^{v(m)^T}\mathbf{X}^{t(m)}\boldsymbol{\epsilon}^{t(m)} + \boldsymbol{\epsilon}^{v(m)}
\end{bmatrix}.
$$

$$\tag{89}$$

In Eq. (89), since terms $\boldsymbol{\epsilon}^{t(1:m)}$ and $\boldsymbol{\epsilon}^{v(1:m)}$ have zero mean and are independent of each other, we have

$$
\mathop{\mathbb{E}}_{\boldsymbol{\epsilon}^{t(1:m)},\boldsymbol{\epsilon}^{v(1:m)}} \|\delta\gamma\|_2^2
$$

$$
= \mathop{\mathbb{E}}_{\boldsymbol{\epsilon}^{t(1:m)},\boldsymbol{\epsilon}^{v(1:m)}} \sum_{i=1}^m \left( \left\| \mathbf{X}^{v(i)T} \left( \mathbf{I}_p - \frac{\alpha_t}{n_t} \mathbf{X}^{t(i)} \mathbf{X}^{t(i)T} \right) (\boldsymbol{w}^{(i)} - \boldsymbol{w}_0) \right\|_2^2 \right.
$$

$$
\left. + \left\| \frac{\alpha_t}{n_t} \mathbf{X}^{v(i)T} \mathbf{X}^{t(i)} \boldsymbol{\epsilon}^{t(i)} \right\|_2^2 + \left\| \boldsymbol{\epsilon}^{v(i)} \right\|_2^2 \right)
$$

$$
= m n_v \sigma^2 + \sum_{i=1}^m \left\| \mathbf{X}^{v(i)T} \left( \mathbf{I}_p - \frac{\alpha_t}{n_t} \mathbf{X}^{t(i)} \mathbf{X}^{t(i)T} \right) (\boldsymbol{w}^{(i)} - \boldsymbol{w}_0) \right\|_2^2
$$

$$
+ \sum_{i=1}^m \mathop{\mathbb{E}}_{\boldsymbol{\epsilon}^{t(i)}} \left\| \frac{\alpha_t}{n_t} \mathbf{X}^{v(i)T} \mathbf{X}^{t(i)} \boldsymbol{\epsilon}^{t(i)} \right\|_2^2. \tag{90}
$$

Notice that

$$
\mathop{\mathbb{E}}_{\boldsymbol{\epsilon}^{t(i)}} \left\| \frac{\alpha_t}{n_t} \mathbf{X}^{v(i)T} \mathbf{X}^{t(i)} \boldsymbol{\epsilon}^{t(i)} \right\|_2^2
$$

$$
= \frac{\alpha_t^2}{n_t^2} \mathop{\mathbb{E}}_{\boldsymbol{\epsilon}^{t(i)}} \left( \mathbf{X}^{v(i)T} \mathbf{X}^{t(i)} \boldsymbol{\epsilon}^{t(i)} \right)^T \left( \mathbf{X}^{v(i)T} \mathbf{X}^{t(i)} \boldsymbol{\epsilon}^{t(i)} \right)
$$

$$
= \frac{\alpha_t^2}{n_t^2} \mathop{\mathbb{E}}_{\boldsymbol{\epsilon}^{t(i)}} \mathrm{Tr} \left[ \mathbf{X}^{v(i)T} \mathbf{X}^{t(i)} \boldsymbol{\epsilon}^{t(i)} \boldsymbol{\epsilon}^{t(i)T} \mathbf{X}^{t(i)T} \mathbf{X}^{v(i)} \right] \quad \text{(by trace trick } \mathrm{Tr}[WZ] = \mathrm{Tr}[ZW])
$$

$$
= \frac{\alpha_t^2 \sigma^2}{n_t^2} \mathrm{Tr} \left[ \mathbf{X}^{v(i)T} \mathbf{X}^{t(i)} \mathbf{X}^{t(i)T} \mathbf{X}^{v(i)} \right] \quad \text{(since } \mathop{\mathbb{E}}_{\boldsymbol{\epsilon}^{t(i)}} [\boldsymbol{\epsilon}^{t(i)} \boldsymbol{\epsilon}^{t(i)T}] = \sigma^2 \mathbf{I}_{n_t}). \tag{91}
$$

Plugging Eq. (91) into Eq. (90), we thus have

$$
\mathop{\mathbb{E}}_{\boldsymbol{\epsilon}^{t(1:m)},\boldsymbol{\epsilon}^{v(1:m)}} \|\delta\gamma\|_2^2
$$

$$
= m n_v \sigma^2 + \underbrace{\sum_{i=1}^m \left\| \mathbf{X}^{v(i)T} \left( \mathbf{I}_p - \frac{\alpha_t}{n_t} \mathbf{X}^{t(i)} \mathbf{X}^{t(i)T} \right) (\boldsymbol{w}^{(i)} - \boldsymbol{w}_0) \right\|_2^2}_{\text{Term A}}
$$

$$
+ \underbrace{\sum_{i=1}^m \frac{\alpha_t^2 \sigma^2}{n_t^2} \mathrm{Tr} \left[ \mathbf{X}^{v(i)T} \mathbf{X}^{t(i)} \mathbf{X}^{t(i)T} \mathbf{X}^{v(i)} \right]}_{\text{Term B}} \quad \text{(by Eq. (91))}. \tag{92}
$$

The following two lemmas estimate Term A and Term B.

**Lemma 26.** *When $n_t \geq 16$, we have*

$$
\mathop{\mathrm{Pr}}_{\mathbf{X}^{t(1:m)},\mathbf{X}^{v(1:m)}} \left\{ \mathop{\mathbb{E}}_{\boldsymbol{w}^{(i)}} [\textit{Term A of Eq. (92)}] \leq m n_v \nu^2 \cdot 2\ln(sn_t) \right.
$$

$$
\left. \cdot \left( D(\alpha_t, n_t, s) + \frac{\alpha_t^2(p-1)}{n_t} \cdot 6.25(\ln(spn_t))^2 \right) \right\} \geq 1 - \frac{5 m n_v}{n_t},
$$

*and*

$$
\mathop{\mathbb{E}}_{\mathbf{X}^{t(1:m)},\mathbf{X}^{v(1:m)},\boldsymbol{w}^{(1:m)}} [\textit{Term A of Eq. (92)}] = \nu^2 m n_v \left( (1-\alpha_t)^2 + \frac{\alpha_t^2(p+1)}{n_t} \right).
$$

*Proof.* See Appendix N.1. □

**Lemma 27.** *We have*

$$\Pr_{\mathbf{X}^{v(1:n_t)},\mathbf{X}^{t(1:n_t)}} \left\{ \textit{Term B in Eq. (92)} \leq \frac{m\alpha_t^2\sigma^2 n_v p}{n_t} \ln p \cdot (\ln n_t)^2 \right\} \geq 1 - \frac{2mn_v}{p^{0.4}}, \qquad (93)$$

$$\mathbb{E}_{\mathbf{X}^{v(1:n_t)},\mathbf{X}^{t(1:n_t)}} \left[ \textit{Term B in Eq. (92)} \right] = \frac{m\alpha_t^2\sigma^2 n_v p}{n_t}. \qquad (94)$$

*Proof.* See Appendix N.2. □

Now we are ready to prove Proposition 6.

*Proof of Proposition 6.* By Eq. (92), Lemma 26, and Lemma 27, the result of Proposition 6 thus follows. Notice that the probability is estimated by the union bound. □

### N.1 PROOF OF LEMMA 26

*Proof.* We have

$$\mathbb{E}_{\boldsymbol{w}^{(i)}} \left\| \mathbf{X}^{v(i)T} \left( \mathbf{I}_p - \frac{\alpha_t}{n_t}\mathbf{X}^{t(i)}\mathbf{X}^{t(i)T} \right) (\boldsymbol{w}^{(i)} - \boldsymbol{w}_0) \right\|_2^2$$

$$= \mathbb{E} \left[ \left( \mathbf{X}^{v(i)T} \left( \mathbf{I}_p - \frac{\alpha_t}{n_t}\mathbf{X}^{t(i)}\mathbf{X}^{t(i)T} \right) (\boldsymbol{w}^{(i)} - \boldsymbol{w}_0) \right)^T \right.$$

$$\left. \left( \mathbf{X}^{v(i)T} \left( \mathbf{I}_p - \frac{\alpha_t}{n_t}\mathbf{X}^{t(i)}\mathbf{X}^{t(i)T} \right) (\boldsymbol{w}^{(i)} - \boldsymbol{w}_0) \right) \right]$$

$$= \mathbb{E} \operatorname{Tr} \left( \mathbf{X}^{v(i)T} \left( \mathbf{I}_p - \frac{\alpha_t}{n_t}\mathbf{X}^{t(i)}\mathbf{X}^{t(i)T} \right) (\boldsymbol{w}^{(i)} - \boldsymbol{w}_0) \right.$$

$$\left. (\boldsymbol{w}^{(i)} - \boldsymbol{w}_0)^T \left( \mathbf{I}_p - \frac{\alpha_t}{n_t}\mathbf{X}^{t(i)}\mathbf{X}^{t(i)T} \right) \mathbf{X}^{v(i)} \right) \quad \text{(by the trace trick)}$$

$$= \operatorname{Tr} \left( \mathbf{X}^{v(i)T} \left( \mathbf{I}_p - \frac{\alpha_t}{n_t}\mathbf{X}^{t(i)}\mathbf{X}^{t(i)T} \right) \mathbb{E} \left[ (\boldsymbol{w}^{(i)} - \boldsymbol{w}_0)(\boldsymbol{w}^{(i)} - \boldsymbol{w}_0)^T \right] \right.$$

$$\left. \left( \mathbf{I}_p - \frac{\alpha_t}{n_t}\mathbf{X}^{t(i)}\mathbf{X}^{t(i)T} \right) \mathbf{X}^{v(i)} \right)$$

$$= \operatorname{Tr} \left( \mathbf{X}^{v(i)T} \left( \mathbf{I}_p - \frac{\alpha_t}{n_t}\mathbf{X}^{t(i)}\mathbf{X}^{t(i)T} \right) \begin{bmatrix} \boldsymbol{\Lambda} & \mathbf{0} \\ \mathbf{0} & \mathbf{0} \end{bmatrix} \left( \mathbf{I}_p - \frac{\alpha_t}{n_t}\mathbf{X}^{t(i)}\mathbf{X}^{t(i)T} \right) \mathbf{X}^{v(i)} \right)$$

$$\text{(by Assumption 2).} \qquad (95)$$

Define

$$\mathbf{A}_{(i)} := \begin{bmatrix} \sqrt{\boldsymbol{\Lambda}} & \mathbf{0} \\ \mathbf{0} & \mathbf{0} \end{bmatrix} \left( \mathbf{I}_p - \frac{\alpha_t}{n_t}\mathbf{X}^{t(i)}\mathbf{X}^{t(i)T} \right) \mathbf{X}^{v(i)} \in \mathbb{R}^{p \times n_v}.$$

Plugging $\mathbf{A}_{(i)}$ into Eq. (95), we thus have

$$\mathbb{E}_{\boldsymbol{w}^{(i)}} \left\| \mathbf{X}^{v(i)T} \left( \mathbf{I}_p - \frac{\alpha_t}{n_t}\mathbf{X}^{t(i)}\mathbf{X}^{t(i)T} \right) (\boldsymbol{w}^{(i)} - \boldsymbol{w}_0) \right\|_2^2 = \operatorname{Tr} \left( \mathbf{A}_{(i)}^T \mathbf{A}_{(i)} \right). \qquad (96)$$

Here $[\cdot]_{j,k}$ denotes the element at the $j$-th row, $k$-th column of the matrix, $[\cdot]_{l,:}$ denotes the $l$-th row (vector) of the matrix, $[\cdot]_{:,k}$ denotes the $k$-th column (vector) of the matrix. Notice that only the first $s$ rows of $\mathbf{A}_{(i)}$ is non-zero. Define

$$Q_{i,j,k} := \mathbf{X}^{v(i)}{}_{j,k} \left( 1 - \frac{\alpha_t}{n_t} \left\| \mathbf{X}^{t(i)}{}_{j,:} \right\|_2^2 \right) + \sum_{l=\{1,2,\cdots,p\}\setminus\{j\}} -\mathbf{X}^{v(i)}{}_{l,k} \cdot \frac{\alpha_t}{n_t} \langle \mathbf{X}^{t(i)}{}_{j,:}, \mathbf{X}^{t(i)}{}_{l,:} \rangle.$$

We thus have

$$[\mathbf{A}_{(i)}]_{j,k} = \begin{cases} \nu_{(i),j} \left\langle \left( \mathbf{I}_p - \frac{\alpha_t}{n_t} \mathbf{X}^{t(i)} \mathbf{X}^{t(i)T} \right)_{j,:}, \mathbf{X}^{v(i)}_{:,k} \right\rangle, & \text{when } j = 1, \cdots, s, \\ 0, & \text{when } j = s+1, \cdots, p, \end{cases}$$

$$= \begin{cases} \nu_{(i),j} Q_{i,j,k}, & \text{when } j = 1, \cdots, s, \\ 0, & \text{when } j = s+1, \cdots, p. \end{cases}$$

Therefore, for any $k \in \{1, 2, \cdots, n_v\}$, we have

$$[\mathbf{A}_{(i)}^T \mathbf{A}_{(i)}]_{k,k} = [\mathbf{A}_{(i)}]_{:,k}^T \cdot [\mathbf{A}_{(i)}]_{:,k} = \sum_{j=1}^{p} [\mathbf{A}_{(i)}]_{j,k}^2 = \sum_{j=1}^{s} \nu_{(i),j}{}^2 Q_{i,j,k}^2. \tag{97}$$

By Eq. (96) and Eq. (97), we thus have

$$\text{Term A in Eq. (92)} = \sum_{i=1}^{m} \sum_{k=1}^{n_v} \sum_{j=1}^{s} \nu_{(i),j}{}^2 Q_{i,j,k}^2. \tag{98}$$

**Part 1: calculate the expected value of $Q_{i,j,k}^2$**

By Assumption 1 and Lemma 13, we have

$$\mathbb{E} \left\| \mathbf{X}^{t(t)}_{j,:} \right\|_2^2 = n_t, \text{ and } \mathbb{E} \left\| \mathbf{X}^{t(t)}_{j,:} \right\|_2^4 = n_t(n_t + 2). \tag{99}$$

We also have

$$\mathbb{E}\langle \mathbf{X}^{t(i)}_{j,:}, \mathbf{X}^{t(i)}_{l,:} \rangle^2 = \mathbb{E} \left( \sum_{q=1}^{n_t} \mathbf{X}^{t(i)}_{j,q} \mathbf{X}^{t(i)}_{l,q} \right)^2$$

$$= \sum_{q=1}^{n_t} \mathbb{E}(\mathbf{X}^{t(i)}_{j,q} \mathbf{X}^{t(i)}_{l,q})^2 \text{ (by Assumption 1)}$$

$$= \sum_{q=1}^{n_t} \mathbb{E}[\mathbf{X}^{t(i)}_{j,q}^2] \mathbb{E}[\mathbf{X}^{t(i)}_{l,q}^2]$$

$$= n_t. \tag{100}$$

If we fix $\mathbf{X}^{t(i)}$ and only consider the randomness in $\mathbf{X}^{v(i)}$, since each element of $\mathbf{X}^{v(i)}_{:,k}$ are *i.i.d.* standard Gaussian random variables, then we have

$$Q_{i,j,k} \sim \mathcal{N}(0, \sigma_{Q_{i,j,k}}^2),$$

$$\text{where } \sigma_{Q_{i,j,k}}^2 = \left( 1 - \frac{\alpha_t}{n_t} \left\| \mathbf{X}^{t(i)}_{j,:} \right\|_2^2 \right)^2 + \sum_{l=\{1,2,\cdots,p\}\setminus\{j\}} \left( \frac{\alpha_t}{n_t} \langle \mathbf{X}^{t(i)}_{j,:}, \mathbf{X}^{t(i)}_{l,:} \rangle \right)^2. \tag{101}$$

Thus, we have

$$\mathbb{E}_{\mathbf{X}^{v(i)},\mathbf{X}^{t(i)}} Q_{i,j,k}^2$$

$$= \mathbb{E}_{\mathbf{X}^{t(i)}} \mathbb{E}_{\mathbf{X}^{v(i)}} Q_{i,j,k}^2$$

$$= \mathbb{E}_{\mathbf{X}^{t(i)}} \sigma_{Q_{i,j,k}}^2 \text{ (by Eq. (101))}$$

$$= 1 - 2\frac{\alpha_t}{n_t} \mathbb{E}_{\mathbf{X}^{t(i)}} \left\| \mathbf{X}^{t(i)}_{j,:} \right\|_2^2 + \frac{\alpha_t^2}{n_t^2} \mathbb{E}_{\mathbf{X}^{t(i)}} \left\| \mathbf{X}^{t(i)}_{j,:} \right\|_2^4 + \frac{\alpha_t^2}{n_t^2} \sum_{l=\{1,2,\cdots,p\}\setminus\{j\}} \mathbb{E}\langle \mathbf{X}^{t(i)}_{j,:}, \mathbf{X}^{t(i)}_{l,:} \rangle^2$$

$$= 1 - 2\alpha_t + \frac{\alpha_t^2(n_t + 2)}{n_t} + \frac{\alpha_t^2(p-1)}{n_t} \text{ (by Eq. (99) and Eq. (100))}$$

$$= (1 - \alpha_t)^2 + \frac{\alpha_t^2(p+1)}{n_t}. \tag{102}$$

By Eq. (102) and Eq. (97), we thus have

$$\mathbb{E}[\mathrm{Tr}(\mathbf{A}_{(i)}^T \mathbf{A}_{(i)})] = n_v \left( \sum_{j=1}^{s} \nu_{(i),j}^2 \right) \left( (1 - \alpha_t)^2 + \frac{\alpha_t^2 (p+1)}{n_t} \right).$$

Thus, we have

$$\mathbb{E}[\text{Term A}] = \nu^2 m n_v \left( (1 - \alpha_t)^2 + \frac{\alpha_t^2 (p+1)}{n_t} \right).$$

Notice that we use the definition of $\nu$ and $\nu_{(i)}$ in Assumption 2 that $\nu^2 = \frac{\sum_{i=1}^{m} \nu_{(i)}^2}{m}$ and $\nu_{(i)}^2 = \sum_{j=1}^{s} \nu_{(i),j}^2$.

**Part 2: derive high probability upper bound**

By Assumption 1 and Lemma 11 (where $c = 2.5 \ln(spn_t)$ and $q = e$), for any given $i \in \{1, 2, \cdots, m\}$, $j \in \{1, 2, \cdots, s\}$, and $l \in \{1, 2, \cdots, p\} \setminus \{j\}$, we must have

$$\Pr \underbrace{\left\{ \left| \langle \mathbf{X}^{t(i)}_{j,:}, \mathbf{X}^{t(i)}_{l,:} \rangle \right| \geq 2.5 \ln(spn_t) \sqrt{n_t} \right\}}_{\mathcal{A}_{1,(i,j,l)}} \leq \frac{2}{spn_t}. \tag{103}$$

By Assumption 1 and Lemma 9, for any given $i \in \{1, 2, \cdots, m\}$ and $j \in \{1, 2, \cdots, s\}$, we have

$$\Pr \underbrace{\left\{ \left\| \mathbf{X}^{t(i)}_{j,:} \right\|_2^2 \notin \left[ n_t - 2\sqrt{n_t \ln(sn_t)}, \ n_t + 2\sqrt{n_t \ln(sn_t)} + 2\ln(sn_t) \right] \right\}}_{\mathcal{A}_{2,(i,j)}} \leq \frac{2}{sn_t}. \tag{104}$$

By Eq. (101) and Lemma 14, for any $\mathbf{X}^{t(i)}$, we have

$$\Pr_{\mathbf{X}^{v(i)}} \left\{ \left| \frac{Q_{i,j,k}}{\sigma_{Q_{i,j,k}}} \right| \geq \sqrt{2 \ln(sn_t)} \right\} \leq \frac{1}{sn_t}.$$

Thus, we have

$$\Pr_{\mathbf{X}^{v(i)}, \mathbf{X}^{t(i)}} \underbrace{\left\{ \left| \frac{Q_{i,j,k}}{\sigma_{Q_{i,j,k}}} \right| \geq \sqrt{2 \ln(sn_t)} \right\}}_{\mathcal{A}_{3,(i,j,k)}} \leq \frac{1}{sn_t}. \tag{105}$$

We define $\mathcal{A}_{1,(i,j,l)}$, $\mathcal{A}_{2,(i,j)}$, and $\mathcal{A}_{3,(i,j,k)}$ as shown in Eq. (103), Eq. (104), and Eq. (105), respectively. Then, we define the event $\mathcal{A}$ as

$$\mathcal{A} := \left\{ \begin{array}{l} \mathcal{A}_{1,(i,j,l)}^c, \mathcal{A}_{2,(i,j)}^c, \mathcal{A}_{3,(i,j,k)}^c \text{ hold for all } i \in \{1, 2, \cdots, m\}, \ j \in \{1, 2, \cdots, s\}, \\ l \in \{1, 2, \cdots, p\} \setminus \{j\}, \text{ and } k \in \{1, 2, \cdots, n_v\} \end{array} \right\},$$

Applying the union bound, we then have

$$\Pr[\mathcal{A}] \geq 1 - \frac{2m}{n_t} - \frac{2m}{n_t} - \frac{mn_v}{n_t} \geq 1 - \frac{5mn_v}{n_t}.$$

When $\mathcal{A}$ happens, we have

$$\sigma_{Q_{i,j,k}}^2 = \left( 1 - \frac{\alpha_t}{n_t} \left\| \mathbf{X}^{t(i)}_{j,:} \right\|_2^2 \right)^2 + \sum_{l=\{1,2,\cdots,p\}\setminus\{j\}} \left( \frac{\alpha_t}{n_t} \langle \mathbf{X}^{t(i)}_{j,:}, \mathbf{X}^{t(i)}_{l,:} \rangle \right)^2 \text{ (by Eq. (101))}$$

$$\leq D(\alpha_t, n_t, s) + \frac{\alpha_t^2 (p-1)}{n_t} \cdot 6.25 (\ln(spn_t))^2 \text{ (by } \mathcal{A}_{1,(i,j,l)}^c \text{ and } \mathcal{A}_{2,(i,j)}^c \text{ for all } i, j, l).$$
$$\tag{106}$$

Thus, we have

$$
\begin{aligned}
\text{Term A} &= \sum_{i=1}^{m}\sum_{k=1}^{n_v}\sum_{j=1}^{s}\nu_{(i),j}{}^2 Q_{i,j,k}^2 \text{ (by Eq. (98))}\\
&\leq \sum_{i=1}^{m}\sum_{k=1}^{n_v}\sum_{j=1}^{s}\nu_{(i),j}{}^2 \sigma_{Q_{i,j,k}}^2 \cdot 2\ln(sn_t) \text{ (by } \mathcal{A}^c_{3,(i,j,k)} \text{ for all } i,j,k)\\
&\leq mn_v\nu^2 \cdot 2\ln(sn_t) \cdot \left( D(\alpha_t,n_t,s) + \frac{\alpha_t^2(p-1)}{n_t}\cdot 6.25(\ln(spn_t))^2 \right)
\end{aligned}
$$

(by Eq. (106) and the definition of $\nu^2$ in Assumption 2).

The result of this lemma thus follows by combining Part 1 and Part 2. $\qquad\square$

### N.2 PROOF OF LEMMA 27

*Proof.* In order to show Eq. (93), it suffices to show that

$$
\Pr\left\{ \max_{i\in\{1,2,\cdots,m\}} \text{Tr}\left[ \mathbf{X}^{v(i)^T}\mathbf{X}^{t(i)}\mathbf{X}^{t(i)^T}\mathbf{X}^{v(i)} \right] \leq n_v n_t (\ln n_t)^3 p \right\} \geq 1 - \frac{2mn_v}{n_t^{0.4}}, \tag{107}
$$

and in order to show Eq. (94), it suffices to show that for any $i \in \{1,2,\cdots,m\}$,

$$
\mathbb{E}\,\text{Tr}\left[ \mathbf{X}^{v(i)^T}\mathbf{X}^{t(i)}\mathbf{X}^{t(i)^T}\mathbf{X}^{v(i)} \right] = n_v n_t p. \tag{108}
$$

We first prove Eq. (107). To that end, we notice that $\mathbf{X}^{v(i)^T}\mathbf{X}^{t(i)}$ is a $n_v \times n_t$ matrix. For any $i = 1,2,\cdots,m$ and $j = 1,2,\cdots,n_v$, we have

$$
\begin{aligned}
\left[ \mathbf{X}^{v(i)^T}\mathbf{X}^{t(i)}\mathbf{X}^{t(i)^T}\mathbf{X}^{v(i)} \right]_{j,j} &= \left\| [\mathbf{X}^{v(i)^T}\mathbf{X}^{t(i)}]_{j,:} \right\|_2^2\\
&= \sum_{k=1}^{n_t} \left[ \mathbf{X}^{v(i)^T}\mathbf{X}^{t(i)} \right]_{j,k}^2\\
&= \sum_{k=1}^{n_t} \langle [\mathbf{X}^{v(i)}]_{:,j},\ [\mathbf{X}^{t(i)}]_{:,k} \rangle^2\\
&= \sum_{k=1}^{n_t} \left( \sum_{l=1}^{p} [\mathbf{X}^{v(i)}]_{l,j}[\mathbf{X}^{t(i)}]_{l,k} \right)^2. 
\end{aligned} \tag{109}
$$

Thus, we have

$$
\begin{aligned}
&\max_{i\in\{1,2,\cdots,m\}} \text{Tr}\left[ \mathbf{X}^{v(i)^T}\mathbf{X}^{t(i)}\mathbf{X}^{t(i)^T}\mathbf{X}^{v(i)} \right]\\
&= \max_{i\in\{1,2,\cdots,m\}} \sum_{j=1}^{n_v} \left[ \mathbf{X}^{v(i)^T}\mathbf{X}^{t(i)}\mathbf{X}^{t(i)^T}\mathbf{X}^{v(i)} \right]_{j,j}\\
&= \max_{i\in\{1,2,\cdots,m\}} \sum_{j=1}^{n_v}\sum_{k=1}^{n_t} \left| \sum_{l=1}^{p} [\mathbf{X}^{v(i)}]_{l,j}[\mathbf{X}^{t(i)}]_{l,k} \right|^2 \text{ (by Eq. (109))}\\
&\leq n_v n_t \left( \max_{\substack{i\in\{1,2,\cdots,m\}\\ j\in\{1,2,\cdots,n_v\}\\ k\in\{1,2,\cdots,n_t\}}} \left| \sum_{l=1}^{p} [\mathbf{X}^{v(i)}]_{l,j}[\mathbf{X}^{t(i)}]_{l,k} \right| \right)^2. 
\end{aligned} \tag{110}
$$

Notice that training input $\mathbf{X}^{t(i)}$ and validation input $\mathbf{X}^{v(i)}$ are independence with each other and each element follows *i.i.d.* standard Gaussian distribution. Therefore, by applying Lemma 11 (where $c = \ln n_t$, $k = p$, and $q = p$), for any given $i$, $j$, and $k$, we have

$$
\Pr\left\{ \left| \sum_{l=1}^{p} [\mathbf{X}^{v(i)}]_{l,j}[\mathbf{X}^{t(i)}]_{l,k} \right| > \ln n_t \sqrt{p\ln p} \right\} \leq \frac{2}{\exp(\ln n_t \cdot 0.4\ln p)} \leq \frac{2}{n_t \cdot p^{0.4}}.
$$

The last inequality we use the fact that $\ln n_t \cdot 0.4 \ln p \geq \ln n_t + 0.4 \ln p$ when $\min\{n_t, p\} \geq 256.$[7] By the union bound, we thus have

$$\Pr\left\{\max_{\substack{i \in \{1,2,\cdots,m\} \\ j \in \{1,2,\cdots,n_v\} \\ k \in \{1,2,\cdots,n_t\}}} \left| \sum_{l=1}^p [\mathbf{X}^{v(i)}]_{l,j}[\mathbf{X}^{t(i)}]_{l,k} \right| > \ln n_t \sqrt{p \ln p} \right\} \leq \frac{2mn_v}{p^{0.4}}. \tag{111}$$

By Eq. (110) and Eq. (111), we have proven Eq. (107). The result Eq. (93) of this lemma thus follows.

It remains to prove Eq. (108). To that end, by Eq. (109), we have

$$\mathrm{Tr}\left[\mathbf{X}^{v(i)^T}\mathbf{X}^{t(i)}\mathbf{X}^{t(i)^T}\mathbf{X}^{v(i)}\right] = \sum_{j=1}^{n_v}\sum_{k=1}^{n_t}\left(\sum_{l=1}^p [\mathbf{X}^{v(i)}]_{l,j}[\mathbf{X}^{t(i)}]_{l,k}\right)^2.$$

Thus, by Assumption 1, we have

$$\mathbb{E}\sum_{j=1}^{n_v}\sum_{k=1}^{n_t}\left(\sum_{l=1}^p [\mathbf{X}^{v(i)}]_{l,j}[\mathbf{X}^{t(i)}]_{l,k}\right)^2 = \sum_{j=1}^{n_v}\sum_{k=1}^{n_t}\mathbb{E}\left(\sum_{l=1}^p [\mathbf{X}^{v(i)}]_{l,j}[\mathbf{X}^{t(i)}]_{l,k}\right)^2$$

$$= \sum_{j=1}^{n_v}\sum_{k=1}^{n_t}\sum_{l=1}^p \mathbb{E}\left([\mathbf{X}^{v(i)}]_{l,j}^2[\mathbf{X}^{t(i)}]_{l,k}^2\right)$$

$$= \sum_{j=1}^{n_v}\sum_{k=1}^{n_t}\sum_{l=1}^p \mathbb{E}[\mathbf{X}^{v(i)}]_{l,j}^2\,\mathbb{E}[\mathbf{X}^{t(i)}]_{l,k}^2$$

$$= n_v n_t p,$$

i.e., we have proven Eq. (108) (and therefore Eq. (94) holds). The result of this lemma thus follows. $\square$

## O    UNDERPARAMETERIZED SITUATION

In this case, we have $p \leq mn_v$. The solution that minimize the meta loss is

$$\hat{\boldsymbol{w}}_{\ell_2} := \arg\min_{\hat{\boldsymbol{w}}} \mathcal{L}^{\mathrm{meta}}.$$

When $\mathbf{B}$ is full column-rank, we have

$$\hat{\boldsymbol{w}}_{\ell_2} = (\mathbf{B}^T\mathbf{B})^{-1}\mathbf{B}^T\gamma.$$

Thus, we have

$$\|\boldsymbol{w}_0 - \hat{\boldsymbol{w}}_{\ell_2}\|_2^2 = \|\boldsymbol{w}_0 - (\mathbf{B}^T\mathbf{B})^{-1}\mathbf{B}^T\gamma\|_2^2$$

$$= \|(\mathbf{B}^T\mathbf{B})^{-1}\mathbf{B}^T\delta\gamma\|_2^2 \text{ (by Eq. (11)).} \tag{112}$$

Define

$$b_w^{(p=1)}(a_1, a_2, a_3, a_4) := \frac{\nu^2 a_3}{m}\left(\frac{1 - \frac{\alpha_t}{n_t}a_2}{1 - \frac{\alpha_t}{n_t}a_1}\right)^4 + \frac{\sigma^2 a_3}{m}\left(\frac{\alpha_t a_1}{n_t}\right)^2\frac{\left(1 - \frac{\alpha_t}{n_t}a_2\right)^2}{\left(1 - \frac{\alpha_t}{n_t}a_1\right)^4}$$

$$+ \frac{\sigma^2}{\left(1 - \frac{\alpha_t}{n_t}a_1\right)^2 a_4}.$$

---

[7]Notice that $\ln n_t \geq \ln 256 \approx 5.5 \geq 2$ and $0.4\ln p \geq 0.4\ln 256 \approx 0.4 \times 5.5 \geq 2$. Thus, we have $(\ln n_t - 1)(0.4\ln p - 1) \geq (2-1) \times (2-1) = 1$, which implies that $\ln n_t \cdot 0.4\ln p \geq \ln n_t + 0.4\ln p$.

**Lemma 28.** *Consider the case $p = s = 1$. If there exist $\underline{g}, \overline{g}, \overline{h} \in \mathbb{R}$ such that*

$$\left\|\mathbf{X}^{t(i)^T}\right\|_2^2 \in [\underline{g}, \overline{g}] \text{ for all } i \in \{1, 2, \cdots, m\}, \tag{113}$$

$$\sum_{i=1}^{m} \sum_{j=1}^{n_v} [\mathbf{X}^{v(i)}]_j^2 \leq \overline{h},$$

$$1 - \frac{\alpha_t}{n_t}\overline{g} \geq 0, \tag{114}$$

*then we must have*

$$\underset{\boldsymbol{w}^{(1:m)}, \boldsymbol{\epsilon}^{t(1:m)}, \boldsymbol{\epsilon}^{v(1:m)}}{\mathbb{E}} \|\boldsymbol{w}_0 - \hat{\boldsymbol{w}}_{\ell_2}\|_2^2 \geq b_w^{(p=1)}(\underline{g}, \overline{g}, 1, \overline{h}).$$

*Further, if*

$$\sum_{j=1}^{n_v} [\mathbf{X}^{v(i)}]_j^2 \in [\underline{r}, \ \overline{r}] \text{ for all } i \in \{1, 2, \cdots, m\}, \tag{115}$$

*we then have*

$$\underset{\boldsymbol{w}^{(1:m)}, \boldsymbol{\epsilon}^{t(1:m)}, \boldsymbol{\epsilon}^{v(1:m)}}{\mathbb{E}} \|\boldsymbol{w}_0 - \hat{\boldsymbol{w}}_{\ell_2}\|_2^2 \leq b_w^{(p=1)}(\overline{g}, \underline{g}, (\overline{r}/\underline{r})^2, m\underline{r}).$$

*Proof.* Since $p = s = 1$, $\mathbf{B}$ becomes a vector and $\mathbf{B}^T\mathbf{B}$ is a scalar that equals to

$$\mathbf{B}^T\mathbf{B} = \sum_{i=1}^{m}\left(1 - \frac{\alpha_t}{n_t}\left\|\mathbf{X}^{t(i)^T}\right\|_2^2\right)^2 \sum_{j=1}^{n_v}[\mathbf{X}^{v(i)}]_j^2. \tag{116}$$

By Eq. (112), we have

$$\|\boldsymbol{w}_0 - \hat{\boldsymbol{w}}_{\ell_2}\|_2^2 = \frac{(\mathbf{B}^T\delta\gamma)}{(\mathbf{B}^T\mathbf{B})^2}. \tag{117}$$

In this case, $\mathbf{B}^T\delta\gamma$ is also a scalar that equals to

$$\mathbf{B}^T\delta\gamma = \sum_{i=1}^{m}\sum_{j=1}^{n_v}\left\{\left(1 - \frac{\alpha_t}{n_t}\left\|\mathbf{X}^{t(i)^T}\right\|_2^2\right)^2 \cdot [\mathbf{X}^{v(i)}]_j^2 \cdot (\boldsymbol{w}^{(i)} - \boldsymbol{w}_0)\right.$$

$$- \frac{\alpha_t}{n_t}\left(1 - \frac{\alpha_t}{n_t}\left\|\mathbf{X}^{t(i)^T}\right\|_2^2\right) \cdot [\mathbf{X}^{v(i)}]_j^2 \cdot (\mathbf{X}^{t(i)}\boldsymbol{\epsilon}^{t(i)})$$

$$\left. + \left(1 - \frac{\alpha_t}{n_t}\left\|\mathbf{X}^{t(i)^T}\right\|_2^2\right) \cdot [\mathbf{X}^{v(i)}]_j[\boldsymbol{\epsilon}^{v(i)}]_j\right\}.$$

By the independence and zero mean of $\boldsymbol{w}^{(i)} - \boldsymbol{w}_0$, $\boldsymbol{\epsilon}^{t(i)}$, and $\boldsymbol{\epsilon}^{v(i)}$, we have

$$\underset{\boldsymbol{w}^{(1:m)}, \boldsymbol{\epsilon}^{t(1:m)}, \boldsymbol{\epsilon}^{v(1:m)}}{\mathbb{E}} \left(\mathbf{B}^T\delta\gamma\right)^2$$

$$= \underbrace{\nu^2 \sum_{i=1}^{m}\left(1 - \frac{\alpha_t}{n_t}\left\|\mathbf{X}^{t(i)^T}\right\|_2^2\right)^4 \cdot \left(\sum_{j=1}^{n_v}[\mathbf{X}^{v(i)}]_j^2\right)^2}_{\text{Term 1}}$$

$$+ \underbrace{\left(\frac{\sigma\alpha_t}{n_t}\right)^2 \sum_{i=1}^{m}\left(1 - \frac{\alpha_t}{n_t}\left\|\mathbf{X}^{t(i)^T}\right\|_2^2\right)^2 \left\|\mathbf{X}^{t(i)^T}\right\|_2^2 \left(\sum_{j=1}^{n_v}[\mathbf{X}^{v(i)}]_j^2\right)^2}_{\text{Term 2}}$$

$$+ \underbrace{\sigma^2 \mathbf{B}^T\mathbf{B}}_{\text{Term 3}}. \tag{118}$$

By Cauchy-Schwarz inequality, we have

$$m \sum_{i=1}^{m} \left( \sum_{j=1}^{n_v} [\mathbf{X}^{v(i)}]_j^2 \right)^2 \geq \left( \sum_{i=1}^{m} \sum_{j=1}^{n_v} [\mathbf{X}^{v(i)}]_j^2 \right)^2. \tag{119}$$

We then have

Term 1 of Eq. (118)

$$\in \left[ \nu^2 \left( 1 - \frac{\alpha_t}{n_t} \overline{g} \right)^4 \sum_{i=1}^{m} \left( \sum_{j=1}^{n_v} [\mathbf{X}^{v(i)}]_j^2 \right)^2, \ \nu^2 \left( 1 - \frac{\alpha_t}{n_t} \underline{g} \right)^4 \sum_{i=1}^{m} \left( \sum_{j=1}^{n_v} [\mathbf{X}^{v(i)}]_j^2 \right)^2 \right]$$

(by Eq. (113) and Eq. (114))

$$\in \left[ \nu^2 \left( 1 - \frac{\alpha_t}{n_t} \overline{g} \right)^4 \frac{1}{m} \left( \sum_{i=1}^{m} \sum_{j=1}^{n_v} [\mathbf{X}^{v(i)}]_j^2 \right)^2, \ \nu^2 \left( 1 - \frac{\alpha_t}{n_t} \underline{g} \right)^4 \sum_{i=1}^{m} \left( \sum_{j=1}^{n_v} [\mathbf{X}^{v(i)}]_j^2 \right)^2 \right]$$

(by Eq. (119))

$$\in \left[ \nu^2 \left( 1 - \frac{\alpha_t}{n_t} \overline{g} \right)^4 \frac{1}{m} \left( \sum_{i=1}^{m} \sum_{j=1}^{n_v} [\mathbf{X}^{v(i)}]_j^2 \right)^2, \ \nu^2 \left( 1 - \frac{\alpha_t}{n_t} \underline{g} \right)^4 m \overline{r}^2 \right] \text{ (by Eq. (115))},$$

and

Term 2 of Eq. (118)

$$\in \left[ \left( \frac{\sigma \alpha_t \underline{g}}{n_t} \right)^2 \left( 1 - \frac{\alpha_t}{n_t} \overline{g} \right)^2 \sum_{i=1}^{m} \left( \sum_{j=1}^{n_v} [\mathbf{X}^{v(i)}]_j^2 \right)^2, \ \left( \frac{\sigma \alpha_t \overline{g}}{n_t} \right)^2 \left( 1 - \frac{\alpha_t}{n_t} \underline{g} \right)^2 \sum_{i=1}^{m} \left( \sum_{j=1}^{n_v} [\mathbf{X}^{v(i)}]_j^2 \right)^2 \right]$$

(by Eq. (113) and Eq. (114))

$$\in \left[ \left( \frac{\sigma \alpha_t \underline{g}}{n_t} \right)^2 \left( 1 - \frac{\alpha_t}{n_t} \overline{g} \right)^2 \frac{1}{m} \left( \sum_{i=1}^{m} \sum_{j=1}^{n_v} [\mathbf{X}^{v(i)}]_j^2 \right)^2, \ \left( \frac{\sigma \alpha_t \overline{g}}{n_t} \right)^2 \left( 1 - \frac{\alpha_t}{n_t} \underline{g} \right)^2 \sum_{i=1}^{m} \left( \sum_{j=1}^{n_v} [\mathbf{X}^{v(i)}]_j^2 \right)^2 \right]$$

(by Eq. (119))

$$\in \left[ \left( \frac{\sigma \alpha_t \underline{g}}{n_t} \right)^2 \left( 1 - \frac{\alpha_t}{n_t} \overline{g} \right)^2 \frac{1}{m} \left( \sum_{i=1}^{m} \sum_{j=1}^{n_v} [\mathbf{X}^{v(i)}]_j^2 \right)^2, \ \left( \frac{\sigma \alpha_t \overline{g}}{n_t} \right)^2 \left( 1 - \frac{\alpha_t}{n_t} \underline{g} \right)^2 m \overline{r}^2 \right] \text{ (by Eq. (115))}.$$

Similarly, we have

$$(\mathbf{B}^T \mathbf{B})^2 = \left( \sum_{i=1}^{m} \left( 1 - \frac{\alpha_t}{n_t} \left\| \mathbf{X}^{t(i)T} \right\|_2^2 \right)^2 \sum_{j=1}^{n_v} [\mathbf{X}^{v(i)}]_j^2 \right)^2 \text{ (by Eq. (116))}$$

$$\in \left[ \left( 1 - \frac{\alpha_t}{n_t} \overline{g} \right)^4 \left( \sum_{i=1}^{m} \sum_{j=1}^{n_v} [\mathbf{X}^{v(i)}]_j^2 \right)^2, \ \left( 1 - \frac{\alpha_t}{n_t} \underline{g} \right)^4 \left( \sum_{i=1}^{m} \sum_{j=1}^{n_v} [\mathbf{X}^{v(i)}]_j^2 \right)^2 \right] \text{ (by Eq. (113))}$$

$$\in \left[ \left( 1 - \frac{\alpha_t}{n_t} \overline{g} \right)^4 (m \underline{r})^2, \ \left( 1 - \frac{\alpha_t}{n_t} \underline{g} \right)^4 \left( \sum_{i=1}^{m} \sum_{j=1}^{n_v} [\mathbf{X}^{v(i)}]_j^2 \right)^2 \right] \text{ (by Eq. (115))}.$$

Plugging the above equations into Eq. (117), the result of this lemma thus follows. $\square$

**Proposition 7.** *Define*

$$\underline{b}_{single} = b_w^{(p=1)}(\underline{g}, \overline{g}, 1, \overline{h}),$$

$$\overline{b}_{single} = b_w^{(p=1)}(\overline{g}, \underline{g}, (\overline{r}/\underline{r})^2, m\underline{r}),$$

$$\overline{g} = n_t + 2\sqrt{n_t \log n_t} + 2\log n_t,$$

$$\underline{g} = n_t - 2\sqrt{n_t \log n_t},$$

$$\overline{h} = mn_v + 2\sqrt{mn_v \log(mn_v)} + 2\log(mn_v),$$

$$\overline{r} = n_v + 2\sqrt{n_v \log n_v} + 2\log n_v,$$

$$\underline{r} = n_v - 2\sqrt{n_v \log n_v}.$$

*When $p = s = 1$ and $\alpha_t$ is relatively small such that $1 - \frac{\alpha_t}{n_t}\overline{g} \geq 0$, we must have*

$$\Pr_{\mathbf{X}^{t(1:m)}, \mathbf{X}^{v(1:m)}} \left\{ \mathbb{E}_{\boldsymbol{w}^{(1:m)}, \boldsymbol{\epsilon}^{t(1:m)}, \boldsymbol{\epsilon}^{v(1:m)}} \left\| \hat{\boldsymbol{w}}_{\ell_2}^{(p=1)} - \boldsymbol{w}_0 \right\|_2^2 \geq \underline{b}_{single} \right\} \geq 1 - \frac{2m}{n_t} - \frac{1}{mn_v},$$

$$\Pr_{\mathbf{X}^{t(1:m)}, \mathbf{X}^{v(1:m)}} \left\{ \mathbb{E}_{\boldsymbol{w}^{(1:m)}, \boldsymbol{\epsilon}^{t(1:m)}, \boldsymbol{\epsilon}^{v(1:m)}} \left\| \hat{\boldsymbol{w}}_{\ell_2}^{(p=1)} - \boldsymbol{w}_0 \right\|_2^2 \leq \overline{b}_{single} \right\} \geq 1 - \frac{2m}{n_t} - \frac{2m}{n_v},$$

*Proof.* Notice that $\left\| \mathbf{X}^{t(i)} \right\|_2^2$ follows $\chi^2$ distribution with $n_t$ degrees of freedom, and $\sum_{i=1}^{m}\sum_{j=1}^{n_v}[\mathbf{X}^{v(i)}]_j^2$ follows $\chi^2$ distribution with $mn_v$ degrees of freedom. Given any fixed $i \in \{1, 2, \cdots, m\}$, by Lemma 9 (letting $x = \log n_t$), we have

$$\Pr\left\{ \left\| \mathbf{X}^{t(i)} \right\|_2^2 \geq \overline{g} \right\} \leq \frac{1}{n_t},$$

$$\Pr\left\{ \left\| \mathbf{X}^{t(i)} \right\|_2^2 \leq \underline{g} \right\} \leq \frac{1}{n_t}.$$

By Lemma 9 (letting $x = \log(mn_v)$), we have

$$\Pr\left\{ \sum_{i=1}^{m}\sum_{j=1}^{n_v}[\mathbf{X}^{v(i)}]_j^2 \geq \overline{h} \right\} \leq \frac{1}{mn_v}.$$

By the union bound, we thus have

$$\Pr\left\{ \left\| \mathbf{X}^{t(i)} \right\|_2^2 \in [\underline{g}, \overline{g}], \text{ for all } i \in \{1, 2, \cdots, m\} \right\} \leq \frac{2m}{n_t}.$$

Similarly, we have

$$\Pr\left\{ \left\| \mathbf{X}^{v(i)} \right\|_2^2 \in [\underline{r}, \overline{r}], \text{ for all } i \in \{1, 2, \cdots, m\} \right\} \leq \frac{2m}{n_v}.$$

The result thus follows by Lemma 28. □

We interpret the meaning of Proposition 7 as follows. We first approximate each part by the highest order term. Then we have $\overline{g} \approx \underline{g} \approx n_t, \overline{r} \approx \underline{r} \approx n_v$, and $\overline{h} \approx mn_v \approx m\underline{r}$. Thus, we have

$$b_w^{(p=1)}(\underline{g}, \overline{g}, 1, \overline{h}) \approx b_w^{(p=1)}(\overline{g}, \underline{g}, (\overline{r}/\underline{r})^2, m\underline{r}).$$

Therefore, we can conclude that our estimation on the model error $\left\| \hat{\boldsymbol{w}}_{\ell_2} - \boldsymbol{w}_0 \right\|_2^2$ in this case is relatively tight. In other words, we know that (with high probability when $n_v$ and $n_t$ is relatively large, and $\alpha_t$ is relatively small)

$$\left\| \hat{\boldsymbol{w}}_{\ell_2}^{(p=1)} - \boldsymbol{w}_0 \right\|_2^2 \approx \frac{\nu^2}{m} + \frac{\sigma^2\alpha_t^2}{m} + \frac{\sigma^2}{(1-\alpha_t)^2 mn_v}.$$

