# OpenReview forum: "Theoretical Characterization of the Generalization Performance of Overfitted Meta-Learning"
_ICLR.cc/2023/Conference — ICLR 2023 poster_

### Official Review · Reviewer_uCi7 · 2022-10-23

**Confidence:** 4
**Correctness:** 4
**Technical Novelty And Significance:** 4
**Empirical Novelty And Significance:** 4
**Recommendation:** 6

**Clarity, Quality, Novelty And Reproducibility:**

The results are clearly presented and further explained. I believe the results are novel. For clarity, I suggest
- Add a notation table since there are three stages (training - validation - test) and involve a lot of super/sub-scripts.
- Add the experiments that show experimental values and curves on theoretical errors, and check if they match, like Figure 3 in ​​”Provable Benefits of Overparameterization in Model Compression: From Double Descent to Pruning Neural Networks”.
- Intro: I suggest not using bulk citation like “Belkin et al., 2018; 2019; Bartlett et al., 2020; Hastie et al., 2019; Muthukumar et al., 2019; Ju et al., 2020; Mei & Montanari, 2019”. Please give a brief review of each paper individually so that one can get the message and be convinced that the authors are really aware of what those papers do.

**Strength And Weaknesses:**

Although the model is only linear case, I think the result is important for a first step of understanding the MAML. The proof is solid. Questions in detail:
- In Theorem 1, if $n_t$ is small, $m$ or $n_v$ is large, then the RHS is smaller than zero. Does it mean, if $b_{w_0}$ and $b_w^{ideal}$ are both positive, then RHS of Thm. 1 is always positive? It is still a bit confusing. Intuitively, we need a few data for test phase, but typically as long as it’s larger than $s$ the number of features, the error should be small if you can identify the useful features in previous stages. But I did not see the explicit dependence on $s$. More importantly, why do more training tasks and validation samples hurt the probability, shouldn’t more data mean better learning performance?
- I think the above issue also appears in Prop. 4-6.
- Eq(2), should it be $X^{r^T}$?
- Eq(4) and the text below: $\hat w$ was defined in Eq(3), is it switched to the parameter that minimizes validation error? If so, it would be better to use another notation.
- Could you have a corollary for the case $w_s^0 = 0$ and a discussion? I'm interested in separating the role of mean and covariance in the bound.


**Summary Of The Paper:**

This paper proposes the sample complexity and error bound for MAML applied on linear meta-learning problem. While the features are iid and the tasks apply a subset of features, one can learn the useful features and apply them for downstream learning. The overfitting would not hurt generalization.

**Summary Of The Review:**

I believe the paper is solid in theory and justifies an important problem. However I have a few confusions regarding the insight of the results, and whether they are too restrictive in the scenario that the RHS of the probability bound is larger than zero. I would like to see authors' comment and consider adding on the score.

=============== Update ===============

Score changed to 6 after first round response.

---

> ### Author Response · Authors · 2022-11-18
> **Response to Reviewer uCi7 (Part 2)**
>
> **Continued from "Response to Reviewer uCi7 (Part 1)"**
>
> > *Eq(4) and the text below: $\hat{\\boldsymbol{w}}$ was defined in Eq(3), is it switched to the parameter that minimizes validation error? If so, it would be better to use another notation.*
>
> Eq. (3) and Eq. (4) describe how to calculate different loss/error functions from $\hat{\\boldsymbol{w}}$ and are not the definition of $\hat{\\boldsymbol{w}}$. The definition of $\hat{\\boldsymbol{w}}$ is “a common meta parameter” (mentioned at the beginning of the paragraph before Eq.(3)). The value of  $\hat{\\boldsymbol{w}}$ is indeed determined by some kind of training process (not necessarily minimizes the validation loss). One possible training process/method (also our focus) is defined in Eq.(8) which minimizes the validation loss $\mathcal{L}^{meta}$ and the $l_2$ norm of itself, where we use a different notation $\hat{\\boldsymbol{w}}\_{\ell_2}$. In the remaining part of the paper, we only use $\hat{\\boldsymbol{w}}$ when we want to state some results for the general training process (such as Lemma 1 and Proposition 1), otherwise some other notation is used (e.g., $\hat{\\boldsymbol{w}}\_{\ell_2}$ in Theorem 1 and $\hat{\\boldsymbol{w}}\_{ideal}$ in Proposition 2.)
>
> > *Could you have a corollary for the case $\\boldsymbol{w}_s^0=\\boldsymbol{0}$ and a discussion? I'm interested in separating the role of mean and covariance in the bound.*
>
> Yes, we already have such a result as Proposition 2. Actually, when $\\boldsymbol{w}_s^0=\\boldsymbol{0}$, we have $\\boldsymbol{w}^0=\\boldsymbol{0}$. Therefore, Term 1 of Eq. (12) becomes zero and the error is only Term 2, which is characterized by Eq. (13) and Proposition 2.
>
> > *Add a notation table since there are three stages (training - validation - test) and involve a lot of super/sub-scripts.*
>
> Thanks for the suggestion. In the revision, we have added such a notation table in Appendix B.
>
> > *Add the experiments that show experimental values and curves on theoretical errors, and check if they match, like Figure 3 in ”Provable Benefits of Overparameterization in Model Compression: From Double Descent to PruningNeural Networks”.*
>
> Thanks for the suggestion. In our revision, in Appendix E.1, we have added simulation results as in Figure 2, which shows that our theoretical bound closely matches the experimental value of the ground truth for all the cases we consider.
>
> > *Intro: I suggest not using bulk citations like “Belkin et al., 2018; 2019; Bartlett et al., 2020; Hastie et al., 2019; Muthukumar et al., 2019; Ju et al., 2020; Mei & Montanari, 2019”. Please give a brief review of each paper individually so that one can get the message and be convinced that the authors are really aware of what those papers do.*
>
> Thanks for your suggestion. In the first paragraph of Appendix A, we provided an extended review of those references. To refer readers to this part, we added a sentence in the Introduction that points to Appendix A.
>
> Finally, we thank the reviewer again for the helpful comments and suggestions for our work. If our response resolves your concerns to a satisfactory level, we kindly ask the reviewer to consider raising the rating of our work. Certainly, we are more than happy to address any further questions that you may have during the discussion period.

---

> ### Author Response · Authors · 2022-11-18
> **Response to Reviewer uCi7 (Part 1)**
>
> We thank the reviewer for providing the helpful review! We have addressed the reviewer’s helpful comments and modified the paper accordingly. Please note that in the revised paper, we highlighted our changes by magenta-colored texts in both the main body and the appendices of the paper.
>
> > *In Theorem 1, if $n_t$ is small, $m$ or $n_v$ is large, then the RHS is smaller than zero. Does it mean, if $b_{w_0}$ and $b_w^{ideal}$ are both positive, then RHS of Thm. 1 is always positive? It is still a bit confusing. Intuitively, we need a few data for test phase, but typically as long as it’s larger than $s$ the number of features, the error should be small if you can identify the useful features in previous stages. But I did not see the explicit dependence on $s$. More importantly, why do more training tasks and validation samples hurt the probability, shouldn’t more data mean better learning performance? I think the above issue also appears in Prop. 4-6.*
>
> For linear regression, in the under-parameterized region, we agree that it is true that more data means better learning performance. However, in the overparameterized region, this is not always true. In other words, more data can harm the generalization performance. An intuitive explanation is that more data brings more noise, and the overfitted solution needs to completely fit all noise, which harms the generalization performance. This has been confirmed even in the single-task setting, e.g., shown by Belkin et al. (2020) as we mentioned in Point 4) of Section 3.2 on Page 7, the expected test error of overfitted single task regression $\frac{p-n}{p}||w_0||^2+\frac{n}{p-n-1}\sigma^2$. For a fixed $p$, where $p>n$ as it is in the overparameterized region, by increasing n, the denominator $p-n-1$ becomes smaller, which makes the term larger. A direct consequence is that the generalization curve will have a peak when $p$ is close to $n$.
>
> The dependence on $s$ of our result can be found in some log terms in Proposition 6, which is negligible compared to other terms. Compared to the min $l_1$-norm solution that emphasizes sparsity $s$, it is not surprising that the min $l_2$-norm solution has an insensitive relationship with respect to $s$ (because the min $l_2$-norm solution is not a spare solution that only a few elements are non-zero).
>
> > *Regarding the insight of the results, whether they are too restrictive in the scenario that the RHS of the probability bound is larger than zero.*
>
> Although our current bound only makes sense when the probability (i.e., RHS of the bound) is positive and close to 1, those insights are not restrictive and can still hold even if this requirement is not strictly satisfied. One piece of evidence is that our current simulation results in Fig. 1 are consistent with all our insights, while such a requirement is not always satisfied. For example, for the purple curve in Fig. 1(b), the descent floor appears at $p=31$ when $m=10$, $n_v=3$, and the RHS of the probability bound is negative $1-\\eta=1-26 * 10^2*3^3/(50)^{0.4}\\approx -4892.6<0$.
>
> > *In Eq. (2), should it be ${X^r}^T$?*
>
> Yes. Thanks for pointing this out. We have corrected this typo in the revision.
>
> **Followed by "Response to Reviewer uCi7 (Part 2)"**

---

> > ### Comment · Reviewer_uCi7 · 2022-11-23
> > **Follow-up**
> >
> > Thanks for your comment in both Part A & B.
> >
> > Regarding the first question, is the intuition that when the sample size is roughly same as the degree of freedom, i.e. the number of variables to be learned, then it's close to the "peak" of a typical double descent curve (in the curve the x-axis is number of variables and y-axis is error, like your Fig 1)? In your case, can I interpret the following way: in over-parameterized case, when you fix the sample complexity, the higher the variable dimension is, the better the error is; when you fix the variable dimension $p$, there is a "best" choice of sample complexity (perhaps discuss how to define "best") between $1$ and $p$?
> >
> > Regarding your response to the question "Regarding the insight of the results...", could you discuss how your experiment, especially when it succeed at the point $1-\eta<0$, is compatible with your tightness discussion (Sec 4)? I believe it is important to place a more thorough discussion in Sec 4, such as,
> > - When the success probability $1-\eta<0$, we do not know whether it overfits or generalizes (it perhaps works well, or even better than the result that Thm 1 covers), and we cannot upper/lower bound the generalization error with high probability, is it correct?
> > - Is there another "non-overlapping" scenario that is covered by Theorem 1 but not by the Sec 4 and you can see the discrepancy?
> > - In which cases the tightness is not tight, for example, there are possibly minor differences between the probabilities (RHS) of Thm 1, Prop 3, Prop 4, etc. There are upper and lower uncertanties/gaps in Prop 4, is there a case when $b_{eig, max} - b_{eig, min}$ comparable with/larger than $b_{eig, max}$ or $b_{eig, min}$, can $b_{eig, min}<0$?
> >
> > Will lift my score to 6 for now.

---

> > > ### Author Response · Authors · 2022-11-25
> > > **Response to the follow-up comments of Reviewer uCi7**
> > >
> > > Thank you for the follow-up questions/comments.
> > >
> > > ---
> > >
> > > > *Is the intuition that when the sample size is roughly same as the degree of freedom, then it's close to the "peak" of a typical double descent curve?*
> > >
> > > Yes. In the classical single-task linear regression, the peak usually appears $p\approx n$. In our meta-learning setup, the peak appears around $p\approx m n_v$, where $m n_v$ means the total number of validation samples of all tasks.
> > >
> > > > *Can I interpret the following way: in overparameterized case, when you fix the sample complexity, the higher the variable dimension is, the better the error is.*
> > >
> > > Thank you. While this is a good observation, it is true only when noise and/or task diversity is large (e.g., blue and orange curves in Fig 1(a)). In other cases, the error decreases to a floor then increases (e.g., green, red, and purple curves in Fig1(a)(b)). This is explained in detail in the 3rd insight on Page 6: **"3) The descent curve is easier to observe under large noise and task diversity, and the curve sometimes has a descent floor."**
> > >
> > > > *For fixed $p$, is there a "best" choice of sample complexity between $1$ and $p$?*
> > >
> > > Great question! The “best” choice means the lowest test error. The test error is bounded by $b_{w_0}+b_{ideal}$ in Thm. 1, where $b_{w_0}$ and $b_{ideal}$ are estimated more explicitly in Eq. (9). In Eq. (9), we can see that $b_{w_0}$ will decrease when $n_v$ increases, while $b_{ideal}$ will increase when $n_v$ increases. By letting the derivative of the sum of the two terms with respect to $n_v$ to zero and examining the solution within the boundary of $n_v$ between $1$ and $\frac{p}{m}$, we can precisely obtain the optimal value of $n_v$.
> > >
> > > > *Discuss how your experiment, especially when it succeed at the point $1-\eta<0$, is compatible with your tightness discussion (Sec 4)?*
> > >
> > > Yes, we will add more discussions as you have suggested. There are three parts of our paper to characterize the generalization performance: the upper bound in Thm. 1, the theoretical tightness discussion in Sec. 4, and the experiment. The first two parts are consistent (i.e., they have similar conditions/requirements and work in the same regime) since Thm. 1 is derived by combining all components estimated by propositions in Sec. 4. The experiment is examined regardless of those conditions/requirements (as we show in the previous response, part of the curves in Fig. 1 are outside of those conditions), which shows that the insights from Thm. 1 are not limited to the regime for the theory to hold.
> > >
> > > We also want to clarify that the tightness is on the expression of the error under certain conditions (e.g, $p$ is sufficiently large), not the expression of the probability $1-\eta$. It is possible that the expression of $1-\eta$ can be further improved (e.g., by using finer/stronger inequalities to estimate large-deviation/tail probability).
> > >
> > > > *When $1-\eta<0$, we do not know whether it overfits or generalizes, and we cannot upper/lower bound the generalization error with high probability, is it correct?*
> > >
> > > The current Theorem 1 becomes trivial when $1-\eta<0$ since it is no longer a high probability. However, by minor changes in derivation (by enlarging the logarithm terms and constants terms in the bound), we can make $1-\eta$ larger, which still leads to a high probability bound (although it may not be very sharp in terms of those logarithm terms and constants).
> > >
> > > From a math point of view, the high-probability estimation of the test error always needs quantities such as $p$ and $n_t$ to be large to overcome the randomness. The expression of $1-\eta$ is from the large-deviation/tail probability estimation on certain random variables/matrices by some lemmas shown in Appendix G.3. Usually, the requirement on sufficiently many samples to achieve high probability in those lemmas is conservative and more than enough to observe a phenomenon in the experiment. Consequently, the probability $1-\eta$ in Thm. 1 is also conservative, which explains why the experiment still matches the bound even when $1-\eta<0$.
> > >
> > > > *Is there another "non-overlapping" scenario that is covered by Thm. 1 but not by Sec 4?*
> > >
> > > No. Thm. 1 is derived by combining all components estimated by propositions in Sec. 4, so Theorem 1's scenario/requirement is indeed the intersection of all propositions in Sec. 4.
> > >
> > > > *In which cases the tightness is not tight ... can $b_{eig, min}<0$?*
> > >
> > > Yes, when $p$ is relatively small (i.e., $p$ and $\sqrt{p}\ln p$ are close), then $b_{eig, min}<0$, which makes the tightness not tight. In this case, the result in Theorem 1 also becomes trivial because $b_{ideal}$ in Theorem 1 becomes infinite (the precise expression of $b_{ideal}$ is given in the last paragraph of Sec 4, which is $b_{ideal}=\frac{b_{\delta}}{\max\\{b_{eig,min},\ 0\\}}$ when $p>n_t$).
> > >
> > > ---
> > > ---
> > > We will add the above discussions to our future revision of the paper  (currently we are not allowed to update the paper anymore before the final decision).

---

> > > > ### Comment · Reviewer_uCi7 · 2022-11-27
> > > > **Follow-up**
> > > >
> > > > First, thank you for the detailed response. Many points are clearer to me.
> > > >
> > > > The explanation about the "tightness" is still vague to me. In a nutshell, could I say there are two types of tightnesses: the tightness in *error*, and in *probability*. The first one is
> > > >
> > > > $ P(x<A) > B \ \Rightarrow \ P(x\approx A) > B$ (1)
> > > >
> > > > and the second one is
> > > >
> > > > $ P(x<A) > B \ \Rightarrow \ P(x < A) \approx B$ (2)
> > > >
> > > > Ideally we want $P(x \approx A) \approx B$. Your tightness meant (1). Did I interpret it correctly?

---

> > > > > ### Author Response · Authors · 2022-11-27
> > > > > **Response to the follow-up question of Reviewer uCi7**
> > > > >
> > > > > Yes, we fully agree with your interpretation of the two types of errors, and our tightness is type (1). When $B$ is close to 1, type (1) already gives sufficient confidence on the tightness of $x\approx A$. Type (2) is more ideal but generally very hard to achieve, e.g., even if the objective is simply to estimate the eigenvalues of a Gaussian matrix in high probability (as far as we know, the best result in the literature for this simple objective is Lemma 8 in Appendix G.3 and is also type (1)). From a math point of view, our result is type (1) because the standard/existing concentration inequalities used in our paper (in Appendix G.3) are type (1).
> > > > >
> > > > > Thanks for pointing this out. We will clarify this part in the future revision of the paper.

---

> > > > > > ### Comment · Reviewer_uCi7 · 2022-11-27
> > > > > > **Follow up**
> > > > > >
> > > > > > Thanks for your response. Yes I see your point that (2) is generally hard. Indeed, most of time the right hand side $B$ is really close to 1 in high dimensional case, so that it looks like $P(x<A) \approx 1$, with the tightness (1), it's $P(x \approx A) \approx 1$. Sometimes, it can also be $E(x) = A$ plus a concentration bound $P(x<A-\epsilon), P(x>A+\epsilon) < \delta$. For failures, it should look like $P(x>A') \approx 1$ or $E(x) = A'$ where $A'$ is a big number that makes the error too large.
> > > > > >
> > > > > > So *I think it would be important to discuss the scenarios when $B$ is far from $1$ even $B<0$, in which case your theorem does not provide a insight*.
> > > > > >
> > > > > > The point does not necessarily make your result weak (although your result would have been stronger with that). The reason is that, concentration usually happens asymptotically, or in high dimensions, but you have a few dimensions or degree of freedoms there (typically the dimensions are analyzed when they go to infinity with a fixed proportion, like the paper "On the Optimal Weighted l2 Regularization in
> > > > > > Overparameterized Linear Regression", so it does not concentrate trivially). Especially, as an example, when the sample complexity is close to the dimension of learned parameters, i.e. at the peak of the double descent curve, it is unlikely to obtain a high probability bound.

---

> > > > > > > ### Author Response · Authors · 2022-11-28
> > > > > > > **Response to the follow-up comment of Reviewer uCi7**
> > > > > > >
> > > > > > > We thank the reviewer’s kind and helpful comments and will add more discussion in the paper correspondingly. While our theorem does not cover the near-field case (i.e., around the peak, no high probability), our experiment suggests that the expression revealed by the theorem in the far field (i.e., away from the peak so the probability is close to 1) still works in the near field. We totally agree that it would be an important and interesting future direction to theoretically study the near-field case so that we can better understand this consistency between the far field and the near field. As the reviewer pointed out, some different techniques (e.g., asymptotic analysis with fixed ratio) might be needed because the current non-asymptotic high-probability estimation around the peak is unlikely to be obtained.

---

### Official Review · Reviewer_Akh7 · 2022-10-24

**Confidence:** 3
**Clarity, Quality, Novelty And Reproducibility:** See above
**Correctness:** 4
**Technical Novelty And Significance:** 3
**Empirical Novelty And Significance:** Not applicable
**Recommendation:** 8

**Strength And Weaknesses:**


Overall, the results proved in this paper are significant for demonstrating the benefits of overparameterized meta-learners, though the typical asterisks associated with the mixed-linear regression task and Gaussian features persist. The paper feels complete, but I find two factors that leave me wanting:

1. Bernacchia(2021) leverages NTK theory to extend their analysis to non-gaussian and non-linear settings. Could similar techniques be used to extend your results to infinite-width settings? A road map highlighting results and propositions required to bridge this gap could contribute greatly to a discussion section.

2. In showing the tightness of the bound in theorem 1, can you provide some numerical examples under some concrete sets of system parameters, and present it through visualizations? The tightness of the bounds in propositions 4 and 5 are quite difficult to contextualize as is. Space can certainly be found to present such figures, if not in at least in the appendix?

3. In your setting, an assumption seems to be that the true features are a subset of the chosen features? Can you comment on what would happen if you incorporate a “misspecification” term into the analysis, i.e. where some true features are omitted?


**Summary Of The Paper:**

This work examines the generalization error and model error of single-step MAML in the mixed linear regression task. Unlike prior works, this work explicitly considers optimization in the over-parameterized regime, and derives model-error bounds in the non-asymptotic regime. Through theoretical derivations, the authors suggest that overparameterization tends to be beneficial, especially when task diversity and difficulty is high.

**Summary Of The Review:**

While some additional experimental verifications (or rather illustrations) would improve the absorbability of this work, I believe that this work makes solid theoretical contributions to our understanding of meta-learning, and therefore I would recommend acceptance.

---

> ### Author Response · Authors · 2022-11-18
> **Response to Reviewer Akh7**
>
> We thank the reviewer for providing the helpful review! We have addressed the reviewer’s helpful comments and modified the paper accordingly. Please note that in the revised paper, we highlighted our changes by magenta-colored texts in both the main body and the appendices of the paper.
>
> > *Bernacchia(2021) leverages NTK theory to extend their analysis to non-gaussian and non-linear settings. Could similar techniques be used to extend your results to infinite-width settings? A road map highlighting results and propositions required to bridge this gap could contribute greatly to a discussion section.*
>
> Yes, we expect that our methods to characterize the double-descent shape can be extended to NTK that approximates a shallow NN setting where $p$ denotes the number of neurons (i.e., width), by leveraging some existing works (mentioned in Appendix A) in the single task regression that utilize the NTK to explain the double-descent curve shape together with the techniques we develop here for meta-learning.
>
> Here are some high-level steps to figure out the shape of the descent curve with respect to the width of NN under the NTK setting: 1) determine the features from NTK assumption, i.e., small changes from the initial point such that the activation pattern of neurons does not change much; 2) construct the overfitted solution by the meta-learning process and those NTK features; 3) derive the set of functions corresponding to the overfitted solution when the width $p$ goes to infinity (i.e., the potential learnable function set); 4) figure out the test error under the ideal situation (infinite width, no noise or task diversity); 5) bound the gap of the performance caused by finite width, noise, and task diversity.
>
> Notice that the shape of the descent curve for NTK  could be very different because of the critical difference between the Gaussian features and NTK features. Notice that even in the single-task scenario, the double descent phenomenon in the NTK setting can be very different from the one with Gaussian features (see, e.g., [Ju et al., 2021)]).
>
> > *In showing the tightness of the bound in theorem 1, can you provide some numerical examples under some concrete sets of system parameters, and present them through visualizations? The tightness of the bounds in propositions 4 and 5 are quite difficult to contextualize as is.*
>
> Thanks for your suggestion. In our revision, in Appendix E.1, We have also added simulation results in Figure 2, which shows that our theoretical bound closely matches the experimental value of the ground truth for all the cases we consider.
>
> > *In your setting, an assumption seems to be that the true features are a subset of the chosen features? Can you comment on what would happen if you incorporate a "misspecification" term into the analysis, i.e. where some true features are omitted?*
>
> Yes, we assume all true features are a subset of the chosen featrues.
> If some true features are omitted, it is equivalent that there always exists a portion in the output that cannot be fitted by the chosen features. Since we assume the i.i.d. standard Gaussian features, this portion is also a Gaussian vector with zero mean and variance being equal to the power of corresponding missing true parameters, which can be viewed as a part of the noise. In other words, the only difference is that $\sigma^2$ becomes $\sigma^2+||w_{missing}||_2^2$.

---

### Official Review · Reviewer_LapN · 2022-10-24

**Confidence:** 2
**Correctness:** 3
**Technical Novelty And Significance:** 3
**Empirical Novelty And Significance:** Not applicable
**Recommendation:** 6

**Clarity, Quality, Novelty And Reproducibility:**

The paper is clearly written, especially considering that it is math-heavy, and is of high technical quality. The novelty of the problem considered and the proof technique is somewhat limited, but this should not be a reason for rejection.

**Strength And Weaknesses:**

__Strength__
- The technical quality of the result is clear, and the proof seems to be correct (but I could be wrong, as did not have enough time to go over the full derivation line-by-line, especially Appendix G).
- The writing is clear overall, stating the assumptions and the setting in a rigorous manner.
- The paper gives a very detailed discussion on the implication of the bound.

__Weakness__
- An explicit head-to-head comparison with the prior work is much needed. Although the paper already gives some discussions regarding the prior work in the introduction (and against Bernacchia (2021) in p6), it is still not super clear to me why Theorem 1 is an improvement over the previous results other than Bernacchia's. I am curious how the bound compares with Chen et al. (2022), especially because the criticism is that their bound is "not in an explicit form (due to unknown quantities)" and "tightness of the bound is unclear." Regarding the former criticism, I think the same comment can be given to Theorem 1, as it contains the terms regarding the true model parameter $\mathbf{w}_0$ and other constants from assumptions (which are usually not explicitly known but should be estimated). Perhaps including a big table for comparison, similar to that appearing in Chen et al., would be a good idea.

- Related to the previous point, this paper considers a slightly different framework than previous works. Namely, the framework has a additional parameter $s$ to denote the true degree of freedom as opposed to the nominal dimensionality. I am curious how the provided result compares to the prior results when $s = p$ (please correct me if I missed something).

- Tightness of the bound is part quite difficult to follow. Exactly how good is the approximation (e.g., the ratio between the upper and lower bound)? I recommend using cruder and simpler approximations to the weights regarding $p, m, n_v$ for the sake of clarity.


**Summary Of The Paper:**

The paper gives the test risk guarantee on the minimum $\ell_2$ norm zero-loss solution (via combination of Lemma 1 and Theorem 1), which is where overparameterized model converges to via SGD. The bound, when simplified, leads to several interesting observations such as double descent via task diversity. Authors also provide results that support the relative tightness of the provided bound.

**Summary Of The Review:**

The contribution of the paper is not very clear in comparison with the prior work; a head-to-head comparison is much needed.

---

> ### Author Response · Authors · 2022-11-18
> **Response to Reviewer LapN**
>
> We thank the reviewer for providing the helpful review! We have addressed the reviewer’s helpful comments and modified the paper accordingly. Please note that in the revised paper, we highlighted our changes by magenta-colored texts in both the main body and the appendices of the paper.
>
> > *It is not super clear to me why Theorem 1 is an improvement over the previous results other than Bernacchia's. I am curious how the bound compares with Chen et al. (2022), especially because the criticism is that their bound is "not in an explicit form (due to unknown quantities)" and the "tightness of the bound is unclear." Regarding the former criticism, I think the same comment can be given to Theorem 1, as it contains the terms regarding the true model parameter and other constants from assumptions (which are usually not explicitly known but should be estimated). Perhaps including a big table for comparison, similar to that appearing in Chen et al., would be a good idea.*
>
> > *The contribution of the paper is not very clear in comparison with the prior work; a head-to-head comparison is much needed.*
>
> Thanks for the comments. We clarify the comparison with Chen et. (2022) as follows.
>
> * By “unknown quantities”, we mean that the eigenvalues of weight matrices in Chen et al. (2022)’s bound are coupled with the system parameters such as the number of tasks, sample size, and feature numbers, etc, so that their bound is not fully expressed in terms of the scaling orders of those system parameters. In contrast, our bound is expressed fully in the scaling orders of those system parameters, which allows us to explicitly characterize the shape of the double-descent curve as well as other phenomena in a quantitative way (in Section 3.2). Nevertheless, we understand that such a difference is reasonable because we consider a more specific setup (such as Gaussian features) than the setup in Chen et al. (2022), which makes it possible for us to get a sharper estimation.
>
> * By “tightness of the bound”, we mean that we have justified the tightness of our bound in Section 4, whereas Chen et al. (2022) does not have such a discussion. We also understand that it may not be easy to analyze the tightness with their more general data model.
>
> In the revision, we have rephrased our comments compared with Chen et al. (2022), and provided a table that positions our work in Table 1 of Chen et al. (2022) as suggested. We have also improved our discussion on tightness by providing more explanation and an additional simulation figure (in Appendix E.1) to verify the tightness.
>
>
> > *Related to the previous point, this paper considers a slightly different framework than previous works. Namely, the framework has an additional parameter $s$ to denote the true degree of freedom as opposed to the nominal dimensionality. I am curious how the provided result compares to the prior results when $s=p$ (please correct me if I missed something).*
>
> By letting $s=p$ and adding additional assumptions that used in Bernacchia (2021) (such as some asymptotic order of system parameters), our upper bound is consistent with Bernacchia (2021)’s mean value estimation (only differs by some constant factors). However, we do want to emphasize that differentiating between $s$ and $p$ makes the entire derivation much more complex. Further, not forcing $s=p$ is essential to investigate how the generalization performance is affected by different $p$ (numbers of features), since we have to keep the ground truth unchanged to isolate the effect of $p$. If we always force $s=p$, then by increasing $p$, the ground truth also changes, which makes it impossible to analyze how increasing the number of parameters affects the generalization performance.
>
> > *Tightness of the bound is the part quite difficult to follow. Exactly how good is the approximation (e.g., the ratio between the upper and lower bound)? I recommend using cruder and simpler approximations to the weights regarding $p,m,n_v$ for the sake of clarity.*
>
> Thanks for your suggestion. In the revision, we added additional discussion on the tightness for each proposition in Section 4 to make it easier to understand. Specifically, for every component of the bound, our discussion illustrates how the estimates from below, estimates from above, and/or mean values approximately match each other by ignoring terms with lower-order magnitudes. For example, we add the following explanation for Proposition 3:
>
> *Proposition 3 gives three estimates on Term 1 of Eq. (12): the upper bound $b_{w_0}$, lower bound $\tilde{b}_{w_0}$, and the mean value. If we ignore all logarithm terms, then these three estimates are the same, which implies that our estimation on Term 1 of Eq. (12) is fairly precise.*
>
> In our revision, in Appendix E.1, We have also added simulation results in Figure 2, which shows that our theoretical bound closely matches the experimental value of the ground truth for all the cases we consider.

---

> ### Author Response · Authors · 2022-12-01
> **Your feedback is important to us**
>
> Dear Reviewer LapN,
>
> This is a friendly reminder that we have submitted our response to your review comments and uploaded a revision of the paper two weeks ago, and we will appreciate very much if you could give us any feedback. In particular, our response explained in detail about the comparison of our result to Chen et al. (2022), which we hope will resolve your main concern. Our response also provided answers to other questions you asked in your review. If our response resolves your concerns, we kindly ask you to consider raising the rating of our work. We are also more than happy to answer your further questions. Thank you very much for your time and efforts!

---

> > ### Comment · Reviewer_LapN · 2022-12-02
> > **Thank you for the response.**
> >
> > Dear authors,
> >
> > sorry for the late response, and thank you for providing a detailed one.
> > My concerns have been addressed well. Raised a score.
> >
> > Best,
> > reviewer.

---

> > > ### Author Response · Authors · 2022-12-02
> > > **Thank you**
> > >
> > > Many thanks for the positive feedback and raising the score!

---

### Official Review · Reviewer_3kzv · 2022-10-24

**Confidence:** 2
**Correctness:** 3
**Technical Novelty And Significance:** 4
**Empirical Novelty And Significance:** 3
**Recommendation:** 8

**Clarity, Quality, Novelty And Reproducibility:**

This paper is well-ordered, and the message delivered is clear. I didn't see the supplementary material, so I am not sure whether the experiments are reproducible.

**Strength And Weaknesses:**

It is interesting to check whether benign overfitting and double descent phenomena exist in Meta-learning. The theoretical result is supported by comprehensive and technically strong proof. However, more evidence is needed to see whether such a finding also exists in a real application.

**Summary Of The Paper:**

This paper studied benign overfitting and double descent phenomenon in Meta-learning. Their analysis is based on linear regression with Gaussian features. In particular, they characterize the descent curve of the model error for the overfitted min l2-norm solution and show the phenomenon would be quite different compared to the single task (non-meta learning) setting.

**Summary Of The Review:**

This paper studied benign overfitting and double descent phenomenon in Meta-learning. I think this paper is well written, but currently, I can only give a weak acceptance because I have the following concerns,

I am very interested in the messages delivered in section 3.2. I am convinced that such phenomena exist in the synthetic data distribution, but do they also exist for NNs in real applications?  Some experiments for real data or relevant literature may help a lot.

One of the most significant double descent phenomena for a single task is for epochs (iterations). What will happen for Meta-learning? The authors may want to add some comments.

In this paper, the authors studied the train-validation split method. What will happen for the train-train split method? Will the main results change, or remain the same?

---

> ### Author Response · Authors · 2022-11-18
> **Response to Reviewer 3kzv**
>
> We thank the reviewer for providing the helpful review! We have addressed the reviewer’s helpful comments and modified the paper accordingly. Please note that in the revised paper, we highlighted our changes by magenta-colored texts in both the main body and the appendices of the paper.
>
> > *I didn't see the supplementary material, so I am not sure whether the experiments are reproducible.*
>
> Thanks for pointing this out. In the revision, we have provided the code of our experiments in the supplementary material.
>
> > *I am very interested in the messages delivered in section 3.2. I am convinced that such phenomena exist in the synthetic data distribution, but do they also exist for NNs in real applications? Some experiments for real data or relevant literature may help a lot.*
>
> Thanks for your interest in our findings in Section 3.2. Following your suggestion on NNs with real data, we have run an additional meta-learning experiment with a two-layer fully connected NN on the MNIST data set. In this experiment, we do observe that the descent floor also appears in this situation. We have included our result for this experiment in Section E.1.
>
> > *One of the most significant double descent phenomena for a single task is for epochs (iterations). What will happen for Meta-learning? The authors may want to add some comments.*
>
> Many thanks for this suggestion! Regarding the double descent phenomena over epochs (iterations), there have been two research directions as we discuss below.
>
> * Direction 1 is double descent still with respect to model complexity, i.e., the number of model parameters (like our study here), but the model is given as an output of SGD training (e.g.,[1][2][3]), rather than an exact min $\ell_2$ norm solution. Here, we conjecture that meta-learning will have a similar phenomenon due to the intrinsic similarity between single-task and multi-task (e.g., the concept of overfitting and underfitting). However, we also expect that due to task diversity in meta-learning, meta-learning may show some distinct properties in such double descent for epochs.
>
>
> * Direction 2 is double descent with respect to epochs/iterations (e.g., [4] and [5]), i.e., the generalization performance is monitored as the training time goes, and it first improves, then decreases, and finally improves again. This happens because there are some slow-training features, which start to improve the generalization only in the later stage of training. We note that such phenomena are typically observed when there are intermediate embedding layers (which capture unknown features to be learned) so that slow-training features can cause the second descent of generalization. Then in meta-learning, we expect such phenomena will occur if we consider training embedding layers together with linear weights.
>
> [1] Fanghui Liu, Johan A.K. Suykens, Volkan Cevher, “On the Double Descent of Random Features Models Trained with SGD”.
>
> [2] Difan Zou, Jingfeng Wu, Vladimir Braverman, Quanquan Gu, Sham M. Kakade, “Benign Overfitting of Constant-Stepsize SGD for Linear Regression”.
>
> [3] Difan Zou, Jingfeng Wu, Vladimir Braverman, Quanquan Gu, Sham M. Kakade, “Risk Bounds of Multi-Pass SGD for Least Squares in the Interpolation Regime”.
>
> [4] Cory Stephenson, Tyler Lee, “When and how epochwise double descent happens”.
>
> [5] Mohammad Pezeshki, Amartya Mitra, Yoshua Bengio, Guillaume Lajoie, “Multi-scale Feature Learning Dynamics: Insights for Double Descent”.
>
>
> > *In this paper, the authors studied the train-validation split method. What will happen for the train-train split method? Will the main results change, or remain the same?*
>
> When using train-train split (i.e., train-train outer algorithms in Eq. (3) of [1]), both the inner loop and the outer loop use all data. A direct consequence is that the expression of $\mathbf{B}$ in Eq.(7) will be different and thus affects the estimation of different types of the elements in $\mathbf{B}\mathbf{B}^T$ (three types are illustrated in Fig. 4 in Appendix L), which will affect the expression of the bound of the generalization error. Whether the current insights exist in the train-train split situation is unclear without conducting a detailed derivation (we suspect some of the insights may persist, e.g., heavier overparameterization reduces the negative effects of noise and task diversity) and can be an interesting future direction.
>
> [1] Nikunj Saunshi, Arushi Gupta, Wei Hu, “A Representation Learning Perspective on the Importance of Train-Validation Splitting in Meta-Learning”.
>
> Finally, we thank the reviewer again for the helpful comments and suggestions for our work. If our response resolves your concerns to a satisfactory level, we kindly ask the reviewer to consider raising the rating of our work. Certainly, we are more than happy to address any further questions that you may have during the discussion period.

---

> > ### Comment · Reviewer_3kzv · 2022-12-03
> > **Thanks for the clarification**
> >
> > Thanks for providing a detailed response that addressed my concerns. I have raised the score.

---

> > > ### Author Response · Authors · 2022-12-03
> > > **Thank you**
> > >
> > > Many thanks for your positive feedback and raising the score!

---

### Decision · Program_Chairs · 2023-01-20

**Decision:**

Accept: poster

**Justification For Why Not Higher Score:**

The paper is restricted to linear regression with Gaussian features. While this allows for detailed analysis, it is a bit far from state-of-the-art theory (e.g. random feature models are probably within scope) and a bit from from practice.

**Justification For Why Not Lower Score:**

The topic is salient and the reviewers agree that the results are sound and interesting, and they seem sufficiently novel to merit publication.

**Metareview: Summary, Strengths And Weaknesses:**

This paper examines double descent/overparameterization in meta-learning through an analysis of single-step MAML to conclude that benign overfitting can occur and is more prominent when the noise and the diversity of tasks are large. While there were some questions about the comparison to prior work and about the practicality of the assumptions, the reviewers agreed that the theoretical contributions were sound and that the results would be interesting to the community.

**Note From Pc:**

if the above contains the word "oral" or "spotlight" please see: "oral" presentation means -> notable-top-5% and "spotlight" means -> notable-top-25%. As stated in our emails, we are disassociating presentation type from AC recommendations